# Adaptogens in Long-Lasting Brain Fatigue: An Insight from Systems Biology and Network Pharmacology

**DOI:** 10.3390/ph18020261

**Published:** 2025-02-15

**Authors:** Alexander Panossian, Terrence Lemerond, Thomas Efferth

**Affiliations:** 1Phytomed AB, Sjöstadsvägen 6, 59344 Västervik, Sweden; 2EuroPharma USA Inc., Green Bay, WI 54311, USA; terryl@europharmausa.com; 3Department of Pharmaceutical Biology, Institute of Pharmaceutical and Biomedical Sciences, Johannes Gutenberg University, 55128 Mainz, Germany

**Keywords:** stress, mental fatigue, stroke, neurogenesis, adaptogens, network pharmacology, polyvalence, botanical hybrid preparations, synergy

## Abstract

Long-lasting brain fatigue is a consequence of stroke or traumatic brain injury associated with emotional, psychological, and physical overload, distress in hypertension, atherosclerosis, viral infection, and aging-related chronic low-grade inflammatory disorders. The pathogenesis of brain fatigue is linked to disrupted neurotransmission, the glutamate-glutamine cycle imbalance, glucose metabolism, and ATP energy supply, which are associated with multiple molecular targets and signaling pathways in neuroendocrine-immune and blood circulation systems. Regeneration of damaged brain tissue is a long-lasting multistage process, including spontaneously regulating hypothalamus-pituitary (HPA) axis-controlled anabolic–catabolic homeostasis to recover harmonized sympathoadrenal system (SAS)-mediated function, brain energy supply, and deregulated gene expression in rehabilitation. The driving mechanism of spontaneous recovery and regeneration of brain tissue is a cross-talk of mediators of neuronal, microglia, immunocompetent, and endothelial cells collectively involved in neurogenesis and angiogenesis, which plant adaptogens can target. Adaptogens are small molecules of plant origin that increase the adaptability of cells and organisms to stress by interaction with the HPA axis and SAS of the stress system (neuroendocrine-immune and cardiovascular complex), targeting multiple mediators of adaptive GPCR signaling pathways. Two major groups of adaptogens comprise (i) phenolic phenethyl and phenylpropanoid derivatives and (ii) tetracyclic and pentacyclic glycosides, whose chemical structure can be distinguished as related correspondingly to (i) monoamine neurotransmitters of SAS (epinephrine, norepinephrine, and dopamine) and (ii) steroid hormones (cortisol, testosterone, and estradiol). In this narrative review, we discuss (i) the multitarget mechanism of integrated pharmacological activity of botanical adaptogens in stress overload, ischemic stroke, and long-lasting brain fatigue; (ii) the time-dependent dual response of physiological regulatory systems to adaptogens to support homeostasis in chronic stress and overload; and (iii) the dual dose-dependent reversal (hormetic) effect of botanical adaptogens. This narrative review shows that the adaptogenic concept cannot be reduced and rectified to the various effects of adaptogens on selected molecular targets or specific modes of action without estimating their interactions within the networks of mediators of the neuroendocrine-immune complex that, in turn, regulates other pharmacological systems (cardiovascular, gastrointestinal, reproductive systems) due to numerous intra- and extracellular communications and feedback regulations. These interactions result in polyvalent action and the pleiotropic pharmacological activity of adaptogens, which is essential for characterizing adaptogens as distinct types of botanicals. They trigger the defense adaptive stress response that leads to the extension of the limits of resilience to overload, inducing brain fatigue and mental disorders. For the first time, this review justifies the neurogenesis potential of adaptogens, particularly the botanical hybrid preparation (BHP) of Arctic Root and Ashwagandha, providing a rationale for potential use in individuals experiencing long-lasting brain fatigue. The review provided insight into future research on the network pharmacology of adaptogens in preventing and rehabilitating long-lasting brain fatigue following stroke, trauma, and viral infections.

## 1. Introduction

The human organism has a self-regulated integrated cell and organ defense system that can maintain harmonized and synchronized interactions to recover normal physiological functions and maintain health in response to stress. Fatigue is a symptom of overload strain and an adaptive alarming signal to attenuate strain and normalize or upgrade homeostasis. Fatigue, also known as weariness, tiredness, and exhaustion, is a common health complaint generally defined as a lack of energy. Physical fatigue is the inability to continue functioning at a level adequate with normal ability, while mental fatigue may manifest as decreased attention, reduced ability to concentrate, or somnolence. Brain fatigue is a long-lasting pathological mental/cognitive fatigue related to brain injury characterized by an extreme lack of energy and associated with prolonged mental effort, stress, or neurological conditions, which interferes with patients’ ability to perform routine daily activities and negatively affects overall well-being, quality of life, and social activities.

Technological progress and rapid development of innovative technologies have a crucial, stressful impact on the health of many individuals experiencing mental overload, a shortage of mental energy, and fatigue. Modern humans live differently from our ancestors; however, their biological functions remain in a stress-induced state of maximal work capacity without energy-consuming muscle activity. Over the past decades, work has changed from demanding physical effort to demanding mental exertion, resulting in a substantial increase in stress and mental fatigue complaints. During the day, they face complex tasks by organizing, planning, and managing at work and dealing with household and personal responsibilities, often accompanied by unexpected interruptions that lead to mental/brain fatigue.

Stress can enhance fatigue when an individual performs a highly cognitive and attention-demanding task, e.g., long hours with saturated information [1]. Furthermore, stress affects brain areas that control working memory, increasing the risk of cognitive overload and learning; stress enhances mental fatigue, causing individuals to spiral into a state of anxiety, depression, and other medical conditions, e.g., stroke or traumatic injury [2].

Stress-induced mental fatigue can have a profound and negative impact on people’s social and occupational life. Thus, Euronews reported on 28 September 2024 the results of European surveys to see which countries have the most stressed employees; see “Are these the most stressful jobs in Europe?”. Research conducted by Lepaya, a training company, across the Netherlands, Belgium, the United Kingdom, and Germany highlighted that two-thirds of European employees were stressed. In Germany and the Netherlands, education, healthcare, public services, finance, and the car maker sector were some of the most stressful sectors to work in. In the UK, nursing jobs are among the highest-stress ones, besides police, social and welfare, and housing jobs. Working in education also fell into this sector, affecting primary, secondary, and higher education teaching professionals. Human resources jobs and national government administrative ones also had high-stress levels. What is common for all of them is that they are permanently busy, sitting all day long in front of their computers and conducting stressful cognitive tasks that induce mental fatigue. Consequences and the cost of this lifestyle are hypertension, obesity, metabolic syndrome, and stroke after emotional distress—extreme anxiety, emotional response, or sorrow. The harmful effects of mental fatigue include loss of productivity at work due to medical disability, occupational hazards, deaths from medication errors, and suicidal ideation [3]. Moreover, mental fatigue is a common symptom in many other chronic medical conditions such as cancer, human immunodeficiency virus, multiple sclerosis, chronic fatigue syndrome, Alzheimer’s disease, Huntington’s disease, and Parkinson’s disease [4].

Brain fatigue is a long-lasting symptom of stress-induced neurological disorders, including hemorrhagic stroke. It limits the ability to perform routine daily activities, including work, education, and social activities, negatively impacting overall well-being and quality of life [5,6,7].

Research suggests a neurobiological origin for brain fatigue [5,6]. Thus, a dopamine imbalance, reduced glutamate handling by astrocytes, unspecific signaling of cell networks, and reduced energy supply in the brain have been suggested to cause brain fatigue. Temporal relief fatigue by rest may presumably normalize the imbalance of glutamate and dopamine-mediated neurotransmission and energy supply in neurons and astrocytes [5,6]. Nonetheless, effective recovery of brain and cognitive functions suggests an increase in the development of new neurons (neurogenesis), characterized as a long-lasting process [8,9,10,11,12]. Neurogenesis activation is associated with the upregulation of the expression of genes involved in the proliferation and differentiation of the brain stem into neurons and glial cells. This supports brain homeostasis and harmonizes brain cell communications involved in cross-talks within brain cell networks [13,14,15,16]. Plant secondary metabolites, polyphenols, and terpenoids play protective and defense roles in plant physiology in response to harmful and damaging impacts of UV, heat, cold, insects, microorganisms, etc. Some phytochemicals, particularly adaptogens, might be toxic for insects and microorganisms but not harmful for humans due to comparatively low doses consumed with nutrients. However, adaptogens act as mild stressors and trigger adaptive stress responses in organisms that bolster the organism’s resilience. Numerous studies have aimed to alleviate fatigue by improving adaptability and tolerance to stress through botanical preparations, particularly adaptogens that trigger the organism’s defense response to stressors, extending the limits of resilience to overload in mental fatigue and stress-induced disorders.

Meanwhile, a growing body of recent studies suggests the potential benefits of botanical preparations, including adaptogens in fatigue and stress-induced disorders [17,18,19,20,21,22,23,24,25,26,27,28,29,30,31,32,33,34,35,36,37,38,39,40,41,42,43,44,45,46,47,48,49,50,51,52,53,54,55,56]. Numerous studies have aimed to alleviate fatigue by improving adaptability and tolerance to stress through adaptogens [17,18,19,20,21,22,23,24,25,26,27,28,29,30,31,32,33,34,35,36,37,38,39,40,41,42,43,44,45,46,47,48,49,50,51,52,53,54,55,56]. Early and recent studies show that adaptogens may improve cognitive functions and exhibit antifatigue effects in human subjects under stressful conditions [18,31,32,37,39,41,43,44,56,57,58,59,60]. At the same time, network analysis of the systems pharmacology provided other theoretical models that propose using botanical adaptogens to treat stress-induced disorders, including brain fatigue and related neurological diseases [61,62,63,64,65,66,67,68]. Recent studies of gene expression in neuroglia and neuronal cells suggest that adaptogens exhibit multitarget and pleiotropic effects, unveiling the polyvalent and synergistic actions of botanical hybrid preparations in stress-induced disorders, including neuronal development, suggesting speedy recovery of long-lasting neurological disorders [66,69]. Furthermore, some botanical hybrid preparations used in traditional Chinese medicines resolved stroke-induced brain damage, resulting in a faster recovery of patients, attenuating symptoms of mental fatigue, and normalizing their cognitive functions [70,71].

In this narrative review, we summarize the knowledge gained on this topic, aiming to understand better the neurobiological mechanisms of adaptogens in individuals affected by brain fatigue and provide insight into future research for the prevention and rehabilitation of post-stroke fatigue.

## 2. Stress, Mental Fatigue, and the Effect of Adaptogens

The terms stress and fatigue were introduced in the scientific lexicon, explicitly in materials science, by a German railway engineer, August Wöhler, in 1855 to describe the relationship between applied stress (S) and component life or the number of cycles to failure (N) to determine a material’s fatigue life [72] (Appendix A in the Appendix A). In mechanics and materials science, stress refers to a physical quantity that expresses the internal forces that neighboring particles of a continuous material exert on each other. Stress (S) is measured by the equation S = F/A, where F is the force (F) or the load per unit area applied per cross-sectional area (A). At the same time, fatigue refers to the weakening of a material caused by cyclic loading, resulting in progressive, brittle, and localized structural damage. Once a crack has initiated, each loading cycle will grow the crack a small amount, even if repeated alternating or cyclic stresses are of an intensity below the normal strength. The stresses could be due to vibration or thermal cycling, while the simultaneous action of cyclic stress, tensile stress, and plastic strain causes fatigue damage.

The American Society for Testing and Materials defines fatigue life, N, as the number of stress cycles of a specified character that a specimen sustains before failure of a specified nature occurs. Fatigue life is affected by cyclic stresses, residual stresses, material properties, internal defects, grain size, temperature, etc. The theoretical value for stress amplitude below which the material will not fail for any number of cycles is called a fatigue limit, endurance limit, or fatigue strength [72]. The classic stress life curve drawn by August Wöhler was limited to N = 10^6^ cycles and assumed the stress limit for infinite fatigue life at this point [73].

It is not difficult to find similarities between the impact of stress on brain fatigue in humans, their longevity, and stress-induced damages to the brain, e.g., stroke, traumatic injury, and some other medical conditions. One more analogy is between the stress-protective effects of anticorrosive treatments or lubricants on a material’s fatigue life and the effects of adaptogens on extending productive and disease-free human lifespans.

In medicine, the WHO defines stress as “a state of worry or mental tension caused by a difficult situation. Stress is a natural human response that prompts us to address challenges and threats in our lives” [74].

According to the Canadian Centre for Occupational Health and Safety, 2017, “fatigue is feeling very tired, weary, or sleepy resulting from insufficient sleep, prolonged mental or physical work, or extended periods of stress or anxiety. Boring or repetitive tasks can intensify feelings of fatigue. Fatigue can be described as either acute or chronic” [75].

In life science, stress is defined as threatened homeostasis—a complex dynamic equilibrium/steady state maintained by coordinated physiological processes in the organism [76]. Depending on severity and duration, stress can have quite the following different impacts on the organism—from beneficial to harmful:-Eustress (good, mild, and optimal stress) that initiates a beneficial adaptive stress response;-Distress, when acute distress increases beyond a certain level or chronic stress (burnout) leads to harmful health effects and can cause numerous diseases [77,78,79,80,81,82,83,84].

The general adaptation syndrome has three stages: the alarm reaction, at which the body detects the external stimulus; adaptation, during which the body engages defensive countermeasures against the stressor; and exhaustion, where the body begins to run out of defenses [85,86]. Stress may result from negative or positive events, including both eustress and distress (roughly meaning challenge and overload, respectively). While eustress is essential to life (in the sense, for example, that exercise is required to avoid muscular atrophy), distress can cause disease (Figure 1).

In this context, adaptogens act as chronic eustress, activating adaptive stress response, resilience, and overall survival. Adaptogens trigger the defense adaptive stress response of the organism to stressors, leading to the extension of the limits of resilience to overload, brain fatigue, and mental and aging disorders [25,87] (Figure 1).

The repeated administration of adaptogens gives rise to an adaptogenic, or stress-protective, effect analogous to repeated physical exercise. This leads to a prolonged resistance phase and increased endurance and cognitive functions (Figure 1). A characteristic feature of adaptogens is that they act as eustressors [88].

Homeostasis is a harmonized interaction of all biomolecules involved in regulating metabolic and signaling pathways, maintaining normal cellular and organismal functions and processes in response to external stressors—disruption and dysfunction of homeostasis result in disorders and diseases. Meanwhile, living organisms can spontaneously restore homeostasis due to their ability (adaptability) to recover normal physiological functions over time [25,87]. The challenge is to accelerate the recovery process through pharmaceutical and nutraceutical intervention. This review suggests that adaptogens may adjust recovery time by promoting neurogenesis and supporting adaptability.

**Figure 1 pharmaceuticals-18-00261-f001:**
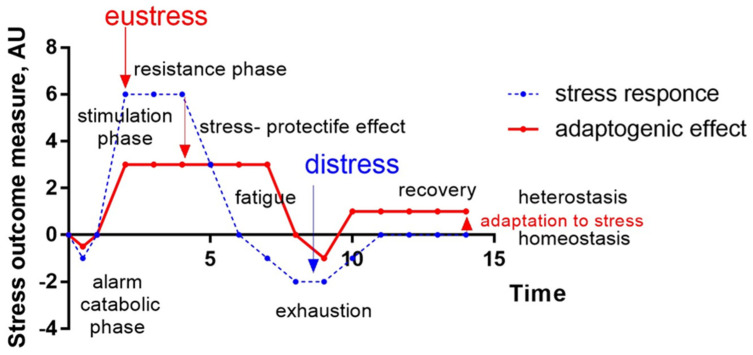
Simplified pattern of adaptive stress response over time (blue color dot line) and stress-protective effect of an adaptogen (red color line) increasing resistance and decreasing sensitivity to a conditional stressor, extending the duration of the resistance anabolic phase to fatigue, and preventing the onset of the exhaustion phase (distress), leading to adaptation to stress and increasing the level of homeostasis. AU—an arbitrary unit of an outcome measure of a stress response, e.g., increasing the number of errors over the time detected in Conner’s computerized cognitive performance test for attention and impulsivity, biochemical markers (cortisol, nitric oxide, etc.). It is modified from [31,32,88,89].

Mental and physical fatigue results from different conditions and has different symptoms [90]. Physical fatigue is a form of tiredness caused by repeated muscle movements that occur after physical exertion, and rest helps [91].

In contrast, mental fatigue is a psychobiological state of tiredness caused by prolonged periods of performing demanding, cognitive-load-inducing activities, which reduces efficiency in cognitive and behavioral performance [92,93,94,95]. Mental fatigue occurs when the brain becomes overworked, tired, and exhausted when rest helps but does not cure it. It leads to an aversion to continuing with the present activity, decreased commitment to the tasks, and difficulties in concentration, memory, and overall mental performance [92].

Mental fatigue decreases an individual’s ability to inhibit responses, process information, and concentrate [2]. The feeling of fatigue provides a signal to terminate action to conserve energy and increase the chances of survival, making the fatigue phenomenon highly adaptive. The adaptive signal encourages the organism to lower effort alternative strategies, e.g., alterations of dopamine and acetylcholine influx into the prefrontal cortex that occur with prolonged task performance is adaptive in the sense that it signals the need to abandon the ongoing behavior in such a way as to promote energy conservation. Efficient spending and energy conservation result in superb chances of survival, making fatigue highly adaptive [92].

Fatigue is a debilitating symptom that impacts healthy individuals and patients with acute and chronic disorders and diseases [3]. The common signs and symptoms of fatigue in the workplace [96] include rubbing the eyes, head nodding, forgetting instructions, long eye blinks, yawning, fidgeting and squirming in the seat, not talking to co-workers, inability to solve routine work problems, irritability, sleepiness (including involuntary sleep onset), inability to concentrate or memorize, lack of motivation, depression, giddiness, and headaches [96]. Fatigue can result from inadequate sleep (e.g., in-flight crews, nurses, truck drivers, the railway industry, nuclear power, medical residents) or intense or monotonous cognitive activities or physical demands [96].

Negative effects of fatigue include reduced decision-making ability, reduced ability to do complex planning, reduced communication skills, reduced productivity or performance, decreased attention and vigilance, reduced ability to handle stress on the job, increased reaction time in speed and thought, loss of memory or the ability to recall details, failure to respond to changes in surroundings or information provided, inability to stay awake, increased tendency for risk-taking, increased forgetfulness, increased errors in judgment, increased sick time, absenteeism, rate of turnover, increased medical costs, and increased rates of adverse incidents [96].

Fatigue is common in various diseases. It is usually defined within the context of specific medical conditions and clinical symptoms, while definitions and characteristics of mental fatigue are different in multiple medical conditions such as Chronic Fatigue Syndrome (CFS), Chronic Obstructive Pulmonary Disorder (COPD), Autoimmune Rheumatic Disease, Cushing’s Syndrome, Epilepsy, Fibromyalgia, Inflammatory Bowel Disease, Multiple Sclerosis, Myasthenia Gravis, Parkinson’s Disease, Rheumatoid Arthritis, Sjogren’s Syndrome and Traumatic Brain Injury [3], e.g., “… one of the most challenging and distressing long-term symptoms, interfering considerably with their ability to work and lead a normal life, including social activities with family and friends” [97].

## 3. Brain Fatigue

The terminologies brain fatigue and mental fatigue are often used interchangeably, but they have distinct nuances. In essence, brain fatigue highlights a more pathophysiological state of the brain’s weariness, while mental fatigue focuses on the psychological aspect of feeling mentally exhausted. However, they frequently overlap in practice. To some extent, the difference between brain and mental fatigue reflects its association with the material of brain tissue, analogous to Wöhler’s mechanical fatigue of various inanimate objects, which lack adaptability. 

The term “brain fatigue” (Hjärntrötthet, in Swedish) was introduced by Rönnbäck and Johansson in 2022 in Sweden [5,6,7] and is often described as mental or cognitive fatigue or long-lasting pathological mental fatigue after physically recovered skull trauma, stroke, infection, brain injury, or inflammation in the central nervous system. Unlike physical fatigue, brain fatigue primarily affects cognitive functions rather than physical stamina [5,6,7,98,99,100].

Brain fatigue is a pathological and disabling symptom with diminished mental energy related to the damaging consequences of traumatic head injuries, strokes, and other diseases that affect the brain. It is characterized by an extreme lack of energy, even after moderate mental activity. Extensive fatigue can set in very quickly, and continuing the activity is impossible. Also typical is a remarkably long-lasting recovery to regain mental energy. It is difficult to return to work and participate in familiar social activities, as typically, the person can only remain mentally active for short periods, and if mentally exhausted, the recovery time will be disproportionally long [5,6,7,99,100].

Brain fatigue is associated with prolonged mental effort, stress, or neurological conditions, including stroke, multiple sclerosis (MS), Parkinson’s disease, dementia, brain tumors, neuroinflammation (encephalitis, meningitis), head injuries, exhaustion syndrome, and neuropsychiatric conditions: ADHD/ADD, autism, cerebral hypoxia, endocrine diseases that affect the brain, and myalgic encephalopathy [5,6].

Most people recover within months after a mild traumatic brain injury or concussion. However, some will suffer from long-term fatigue, reduced quality of life, and the inability to maintain employment status or education [5]. Those who suffer from brain fatigue commonly experience a “stiffness in thinking”, and mental energy can suddenly run out. They usually are overloaded with unsorted, strong, and not attenuated impressions, like the sound of a low-frequency fan lingering and disturbing. They are typically sensitive to stress and experience problems with memory. Everyday activities such as working, participating in conversations, reading, watching TV, shopping, etc., become problematic and can be completely drained of energy, and the recovery time is disproportionately long [5,7,99,100].

In most people, brain fatigue is alleviated or ceases when the injury or illness heals, but in some patients, it remains for months or even years [4,5,101,102] (Figure 2). Brain fatigue is not related to the severity of the injury/disease. Even a mild concussion can have severe and long-lasting consequences [103,104]. Long-term brain fatigue after a minor head injury occurs in 20–25 percent of the 20,000 people who seek medical care in Sweden each year after head trauma [105]. The exact relationship is seen after stroke and meningitis, treated brain tumors and endocrine diseases with brain involvement, as well as in stress-related conditions and myalgic encephalopathy [106]. Brain fatigue also appears to be a lingering symptom in some people affected by COVID-19 [5,40,107,108,109,110,111]. Brain fatigue occurs at all ages, even among children [112].

Today, there is no convincing explanation of how fatigue becomes long-lasting even after the brain injury or disease has healed. Since the extent of the brain injury, location, age, or gender of the person do not seem to have a bearing on the risk of long-term brain fatigue, it may be wondered whether there are risk factors already before the onset or injury in those with long-term brain fatigue. Several studies indicate that people who suffer from depression and anxiety before the injury are more likely to suffer from long-term problems after the injury. If there are problems before the onset of the disease, they often become more evident and bothersome after the onset of the illness, and thus, it increases the burden if brain fatigue is present. There may also be many other factors that are important for brain fatigue to be long-lasting. It also appears that people who have experienced multiple head injuries are more likely to develop brain fatigue [5,6].

### 3.1. Symptoms and Methods of Assessment of Brain/Mental Fatigue and Brain Energy

Brain fatigue cannot be adequately diagnosed with blood tests, cognitive tests, or brain imaging techniques since the underlying cause of brain fatigue in the individual patient should be clarified and a direct temporal relationship with illness or damage to the nervous system [5,95]. To distinguish brain fatigue, Johansson and Rönnbäck, based on their long healthcare experience, have developed a self-assessment scale for mental fatigue (MFS) [113,114,115], including questions related to symptoms of mental fatigue. They lacked a scale that could be used for different patient groups and distinguish mental fatigue. Symptoms included in the scale have been selected based on clinical experience and what has been reported as common symptoms in neurological diseases and injuries [116]. A score of over 10 points on the scale indicates a suspicion of brain fatigue problems crucial to in-depth analysis [114]. MFS is also related to work ability [97]. Objective tests include measuring mental speed/information processing speed and attention using sensitive and robust computerized tests [5,7].

Brain fatigue symptoms can vary in intensity and daytime. The most common signs include an abnormal loss of mental energy, prolonged recovery time, difficulty concentrating, memory problems, mental fog, slowness of thinking, reduced cognitive performance, increasing degree of mental fatigue over the day, difficulty in starting activities, emotional lability and irritability, stress sensitivity, etc. (see Table 1).

Symptoms of brain and mental fatigue can overlap but have key differences, often reflecting each condition’s physical versus psychological nature. Below is a breakdown of the differences.

Symptoms of brain fatigue (physical exhaustion of the brain) include the following:

Cognitive slowness: slower reaction times or delayed thinking processes.Mental fog: difficulty processing information or understanding concepts.Headaches or pressure: sensations in the head after cognitive tasks.Memory issues: forgetfulness or inability to recall recent information.Visual disturbances: blurred vision, light sensitivity, or difficulty focusing the eyes.Physical tiredness: feeling physically drained after mental exertion, even without significant physical activity.Coordination problems: in severe cases, mild issues with hand-eye coordination or balance.

These symptoms are often related to overexertion of brain activity or neurological conditions and can worsen with lack of rest.

Symptoms of mental fatigue (psychological exhaustion) include the following:Lack of motivation: feeling demotivated to start or continue tasks, even simple ones.Irritability: becoming easily frustrated, annoyed, or emotionally reactive.Difficulty concentrating: struggling to focus on tasks, especially those requiring sustained attention.Indecisiveness: making decisions, even about trivial matters, is more demanding.Emotional numbness: feeling emotionally detached, disinterested, or overwhelmed by small things.Sleep disturbances: insomnia or restless sleep due to racing thoughts or stress despite feeling tired.Burnout-like symptoms: feeling mentally and emotionally exhausted, often linked to chronic stress or overload.

While brain fatigue often presents with physical symptoms such as slowed cognitive function, mental fatigue tends to be more tied to emotional and motivational issues such as irritability, lack of focus, and emotional exhaustion. However, the two can often coexist, especially in high-stress situations.

Neuropsychological examination often yields results within normal limits in cases of brain fatigue, even if the patient describes significant problems in everyday life [117].

A wide variety of techniques, methods, and tools were implemented to detect and assess mental fatigue, such as self-reporting questionnaires, attention network tests, heart rate variability, salivary cortisol levels, electroencephalogram, brain imaging, and saccadic eye movements [2,95,113,114,115,118].

Various self-assessments and diverse types of questionnaires were developed to detect and assess mental fatigue, including the following:Occupation fatigue exhaustion recovery (OFER) scale;Visual analog scale to evaluate fatigue severity (VAS-F);Fatigue assessment scale (FAS);Multidimensional fatigue inventory (MFI);Fatigue severity inventory (FSI);Mental fatigue scale (MFS).

Overall, the pros of mental fatigue detection methods include multiple brief questionnaires, which are reliable, easy to administer, and accessible. Cons are related to the inability to provide moment-to-moment fluctuations of mental fatigue, subjective and not immune to response biases, and may not accurately reflect humans’ cognitive state at that particular moment. The validity of these questionnaires is uncertain, as they are highly subjective measurement tools and are not immune to response bias detection techniques [2,113,114].

Another method to detect mental fatigue is the Attention Network Test, which is related to three areas: alerting, orienting, and executive control [119], particularly the following:The changes in reaction times insinuated by a warning signal (alerting).The changes in reaction times result from cues indicating where the target will occur (orienting).The subject examined the efficiency of the executive network in responding by pressing a key indicating the direction of a central arrow surrounded by congruent, incongruent, or neutral flankers.

The pros of mental fatigue detection methods cover three fundamental areas of attention: alertness, orientation, and executive control. The cons are related to a long testing period (25–30 min), practice effects, mainly cognitive and for inferring purposes, and an attention network too complex to be assessed using a single task [2].

Mental fatigue was also measured using EEG spectral changes in the theta and alpha waves in the brain’s frontal and occipital cortical regions, where slow wave activity increases in mentally fatigued individuals [120,121,122], suggesting that as mental load increases, the brain loses its capacity to cope with it, thus slowing down brain activity. Also, individuals who were mentally fatigued from performing cognitive tasks produced decreased α-frequency band power in the brain area, which is responsible for language and semantic memory processing, visual perception, and multimodal sensory integration [123].

Overall, the theta wave was a reliable primary biomarker for mental fatigue, while the alpha wave was a reliable secondary biomarker. These methods require a laboratory setting to reduce signal noise, and they cannot detect mental fatigue while performing cognitively demanding tasks accurately [2].

Heart rate variability measurement is easily integrated into smart devices; however, there are too many confounding variables, and it is of low reliability.

Cortisol level was used alongside health monitoring technologies; however, there is a lack of evidence to support a link between salivary cortisol and mental fatigue; furthermore, saliva samples are easily contaminated.

The testosterone/cortisol ratio, the anabolic index, was used as a biological marker of exercise that indicates chronic overload and overtraining of athletes, suggesting that it reflects the anabolic and catabolic hormonal balance [124,125,126,127]. A prolonged increase in catabolism and a decrease in anabolism may reduce athletic performance and result in overtraining [128,129].

The primary symptoms and signs of chronic overload and overtraining syndrome were classified into four categories [129,130] as follows:Psychological: feelings of depression, general apathy, difficulty concentrating, emotional instability, fear of competition, excitation, and restlessness.Immunological: increased susceptibility to infection, increased severity of minor ailments, decreased functional activity of neutrophils, decreased total lymphocyte counts, reduced response.Biochemical: decreased hemoglobin, increased lactate, elevated cortisol levels, decreased free testosterone levels, and a decrease of more than 30% in the ratio of free testosterone to cortisol.Physiological: decreased performance, body fat, decreased muscular strength, increased CO_2_ at submaximal loads, muscle soreness, changes in the heart rate, prolonged recovery periods, loss of appetite, and chronic fatigue.

Among these symptoms and signs, the testosterone-to-cortisol ratio (T/C) is a reliable marker for monitoring an athlete’s training and possibly diagnosing overtraining syndrome [124,125,126,127,131] and assessing the efficacy of adaptogens used for recovery of athletes after exercising and prevention of the symptoms of overtraining [21]. The results of that study show that supplementation with adaptogenic preparations increases the anabolic index, physical performance, and recovery of athletes after heavy physical and emotional loads, improving the adaptation to physical and emotional stress and significantly decreasing inattention, impulsivity, the perception of stress, and reducing fatigue [21].

Mental energy is associated with mood and is defined as the ability or willingness to engage in cognitive work. Methods used to assess mental energy include tests of cognitive performance, mood questionnaires, electrophysiological techniques, brain scanning technologies, and ambulatory monitoring.

The studies suggest that cognitive tests that assess vigilance, the ability to sustain attention, and choice reaction time are optimal for assessing mental energy. Specific tests recommended include the psychomotor vigilance task, Wilkinson’s choice visual reaction time, the scanning visual vigilance test, and the Wilkinson auditory vigilance test. These tests are sensitive to factors that increase and decrease mental energy [132,133]. Cognitive tests assessing vigilance and choice reaction time appear to be the optimal performance tests to evaluate mental energy [132]. Mental energy was assessed using mood questionnaires and tests of cognitive performance in studies of drugs and food constituents in various disease states where consistent changes in both mood and performance were observed. These tests are usually administered on laboratory microcomputers but can be administered on personal digital assistants or automated ambulatory monitoring devices. Critical methodological issues include using double-masked, placebo-controlled designs, ensuring adequate statistical power, including controls for practice effects, and using appropriate statistical analysis procedures [133].

### 3.2. Treatment of Long-Lasting Brain Fatigue in Rehabilitation of Stroke and Post-Traumatic Brain Injury

Overall, there is no curative treatment for brain fatigue or treatment guidelines apart from what is summarized below [5]. Brain fatigue can be alleviated with pharmacological and non-pharmacological methods to improve patients’ quality of life. Palliative treatment can be implemented in healthcare within a foreseeable time. However, it should be administered by healthcare personnel with knowledge and experience of patients with brain fatigue and an understanding of what the treatment entails. Fatigue is a central parameter to consider if work ability assessments are made or if rehabilitation is planned [5].

Management and treatment of brain fatigue focus on addressing the underlying causes and providing symptomatic relief by the following:-Taking regular breaks during mentally intensive tasks helps to prevent the brain from becoming fatigued.-Improving sleep quality (e.g., through good sleep hygiene or treating sleep disorders like insomnia) is essential. Adequate sleep is critical for brain repair and recovery.-Cognitive Behavioral Therapy can help manage mental fatigue, especially in individuals with underlying stress or mood disorders.-Regular exercise improves blood flow to the brain and enhances overall mental and physical energy levels. It also promotes the release of neurochemicals like endorphins, which can counter fatigue.-Maintaining balanced blood sugar levels through a diet rich in complex carbohydrates, proteins, and healthy fats is crucial.-Techniques such as mindfulness meditation, deep breathing, and yoga can help reduce mental fatigue by lowering stress levels and promoting relaxation.-Modifying workloads, reducing multitasking, and providing flexible scheduling can be helpful for individuals experiencing work-related brain fatigue.-Cognitive brain training exercises or games may help enhance cognitive flexibility and resilience, improving overall cognitive function and reducing fatigue.-Proper management of neurological conditions (e.g., multiple sclerosis, post-concussion syndrome) or psychiatric conditions (e.g., depression, anxiety) can alleviate brain fatigue as a secondary symptom.-Pharmacological interventions such as the following:Stimulants such as methylphenidate (Ritalin) may be prescribed for conditions like Alzheimer’s and Huntington’s disease, where mental fatigue is common.Antidepressants for those with mood disorders contribute to cognitive fatigue.Cognitive enhancers such as modafinil may be off-label in some chronic fatigue cases.

By combining these approaches, individuals can mitigate brain fatigue, improve their cognitive endurance, and improve their overall quality of life.

There is no cure for brain fatigue; however, brain fatigue seems to be alleviated. One approach to alleviating brain fatigue is to stimulate dopamine norepinephrine signaling, which is decreased during neuroinflammation. In several studies, a CNS stimulant, methylphenidate, increases the amount of dopamine and norepinephrine in synaptic areas in the frontal lobes [134,135] and reduces brain fatigue in subjects with mild head injury or stroke [136,137,138].

A dopamine stabilizer OSU6162 produced a milder effect; fewer showed a beneficial effect and fewer side effects [139,140,141]. The participants who had a favorable effect reported that they became clearer in their heads, had more energy, and had an easier time getting started with activities. They rated lower mental fatigue according to the Mental Fatigue Score, and with methylphenidate, they also performed better on neuropsychological speed and attention tests. However, the treatment effects appear reversible, i.e., the positive impact on brain fatigue is only seen during ongoing treatment with methylphenidate [138].

Non-medical treatment for brain fatigue has also been tried to a limited extent. Mindfulness programs can alleviate depression, anxiety, and brain fatigue in multiple sclerosis and after an acquired brain injury [142,143]. The mindfulness program “mindfulness-based stress reduction” can alleviate brain fatigue after stroke and traumatic brain injury, improve cognition (attention), and alleviate symptoms related to depression and anxiety [144,145]. Blue light treatment and cognitive behavioral therapy after a traumatic head injury can also reduce brain fatigue, sleepiness, and depression symptoms [146]. Physical activity is generally described as beneficial for the brain, but when brain fatigue engages in physical activity, it does not work either. Instead, many take leisurely walks and perceive it as beneficial [147].

A better understanding of the underlying pathophysiological and molecular mechanisms can help to develop a more specific treatment for brain fatigue.

### 3.3. Pathophysiology and Biochemistry of Brain Fatigue and Brain Energy

The exact mechanisms behind brain fatigue are not entirely clear, but several factors are believed to contribute, including the following:Chronic low-grade neuroinflammation and oxidative stress: Chronic stress, mental overload, and cardiovascular or neurological disorders can cause inflammation in the brain and oxidative damage to neurons and astrocytes, further contributing to brain fatigue and impaired cognitive function.Impaired brain regions: Mental fatigue is often linked to dysfunction in the prefrontal cortex (responsible for decision-making and attention), the anterior cingulate cortex (which helps in error detection and attention), and other areas involved in executive functioning.Neurotransmitter imbalance: Brain fatigue is often associated with imbalances in neurotransmitters. Overusing certain neural pathways can deplete these critical chemicals, reducing the brain’s efficiency. Excessive excitatory neurotransmitter glutamate in neuronal synapses, disruption of homeostasis in the glutamate-glutamine cycle in astrocytes, and dysfunction of ATP-dependent energy supply by anaerobic glycolysis of glucose.Glucose depletion: The brain consumes significant amounts of glucose (its primary source of brain energy and ATP), especially during prolonged mental activity. As glucose levels decrease, cognitive performance can suffer, leading to fatigue.Sleep Deprivation: Lack of restorative sleep impairs brain functioning, leading to slower processing speeds, reduced memory consolidation, and overall cognitive decline.

#### 3.3.1. Glutamate Neurotransmission Imbalance, Glucose Depletion, and Energy Shortage Hypothesis

Brain injury-derived long-lasting fatigue negatively correlates with information processing speed, brain energy supply, and specific neural signaling [148,149,150], likely due to the disruption of the mitochondria’s function [115,151]. Brain damage results in the activation of microglial cells, which release pro-inflammatory cytokines, affecting astrocyte functions and leading to altered activity in the nerve cell network [149,152,153]. The neuroinflammatory response to brain injury induces excessive excitatory neurotransmitter glutamate in neuronal synapses, disruption of homeostasis in the glutamate-glutamine cycle, and dysfunction of ATP-dependent energy supply by anaerobic glycolysis of glucose [5,6]. The availability of glucose as a primary energy source and ATP production is limited in the cerebral cortex, which is responsible for energy-demanding cognitive functions, decreases. This reduces dopamine, GABA, and serotonin neurotransmission in several brain areas, while glutamate signaling becomes more non-specific [154]. It has been demonstrated that dopamine and norepinephrine signaling decreases in the frontal cortex after skull damage and stroke, which is associated with a reduced ability to concentrate, while serotonin signaling decreases, leading to emotional instability [154]. Upon signaling, dopamine and serotonin are released from the presynaptic terminal and taken back. In contrast, after signaling between neurons is completed, glutamate is taken up in surrounding astrocytes [149,155]. With limited energy availability, the astrocytes cannot take care of all the released glutamate, leading to some glutamate remaining in and around the synaptic cleft, causing the signaling to become less distinct.

Glutamate is the major excitatory neurotransmitter, mediating nervous system plasticity when released at the proper concentrations, whereas inhibitory signals are carried by γ-amino butyric acid (GABA).

Glutamine is synthesized from glutamate in the astrocytes to return the glutamate removed from the synaptic cleft after release from the presynaptic neuron. The neuron will readily convert the astrocyte-derived glutamine to glutamate via glutaminase to complete the glutamine/glutamate cycle in the CNS (Figure 3).

The cycle is energy-dependent since ATP is consumed in the synthesis of glutamine from glutamate: glutamate + ATP + NH3 → glutamine + ADP + phosphate + 17 kJ/mol. In the human cortex, the cycle accounts for 80% of the energy derived from glucose oxidation [161]. After glutamate is released from nerve cell terminals, it affects glutamate receptors of receiving neurons, and the signal can be passed on within the nerve cell networks. The astrocytes uptake glutamate as soon as it has exerted its effect on the receiving neuron. In this way, the sensitivity to glutamate of the nerve cells is maintained. Under normal physiological conditions, there is no limit to how many signals can pass per unit of time. Low extracellular glutamate levels in synapses between neurons and astrocytes are necessary to maintain high precision in information processing, learning, and memory processes. Astroglia cells support optimal glutamate transmission, but it is distorted by an imbalanced energy supply in response to neuroinflammation in brain pathology (Figure 3) [6].

The astroglia regulates the extracellular glutamate concentration maintained at approximately 1–3 mM to maintain a sufficient signal-to-noise ratio in glutamate transmission [162] and to minimize the risk of excitotoxic actions of glutamate on neurons [163]. After a presynaptic release and interaction with the post-synaptic glutamate receptors, they clear the extracellular space from excessive glutamate [164]. Recipient neuronal cells should be sensitive to glutamate, have enough energy to withstand normal stimulation, and remove glutamate at the appropriate rates from the right locations. Glutamate excites nerve cells but can excite cells to death, resulting in excitotoxicity due to glutamate receptors on the surface of brain cells. Both too much glutamate and too little glutamate are harmful. Robust glutamate uptake systems comprising glutamate transporters prevent excessive activation of these receptors by continuously removing glutamate from the extracellular fluid in the brain. Further, the blood–brain barrier shields the brain from glutamate in the blood. The highest glutamate concentrations are in synaptic vesicles in nerve terminals, from which exocytosis can release it [165].

During an acute insult, astrocytes can prevent excitotoxicity by removing extracellular glutamate with high-affinity sodium-dependent excitatory amino acid transporters (EAAT). Furthermore, astrocytes maintain glutamate homeostasis by sustaining its synthesis, uptake, and release via the glutamate-glutamine cycle [156,157,158]. Through this cycle, synaptically released glutamate is predominantly taken up into astrocytes, where it is amidated to glutamine by the astrocyte-specific enzyme glutamine synthetase [166]. Glutamine is then released to the synapse and taken up by adjacent neurons, where it is converted to glutamate and γ-aminobutyric acid (GABA), which are then repackaged into vesicles and again released in the synapse as neurotransmitters [167,168].

The glutamate transporters are responsible for the glutamate removal. Glutamate is transported into synaptic vesicles through a transporter, and glutamatergic neurons release glutamate to synapses through synaptic vesicles. The postsynaptic glutamate receptors are activated by glutamate, inducing synaptic signal transmission. Various transporters in cell and mitochondrial membranes, such as the solute carriers (SLCs) superfamily, maintain the balance of glutamate and glutamine in the synaptic cleft and within cells [169]. Most of the glutamate in synapses is transported into astrocytes by excitatory amino acid transporters (EAATs/GLT and glutamate transporters GLSAT) and converted into glutamine in cells by glutamine synthetase (GS, glutamate ammonia ligase). Glutamine in astrocytes is then secreted into synapses through transporters (SNAT3, SNAT5, or HM). Glutamine is retaken through transporters (SNAT1 or SNAT2) by neurons, where it is converted into glutamate by glutaminase (GLS, glutamine aminohydrolase). Various transporters in cell and mitochondrial membranes, such as the solute carriers (SLCs) superfamily, maintain the balance of glutamate and glutamine in the synaptic cleft and within cells [169].

Glutamate and glutamine are the most abundant amino acids in the blood and play a crucial role in cell survival in the nervous system [168]. In the forebrain, approximately 20% of synoptically released glutamate reaches the postsynaptic glutamate receptors from the synaptic cleft during synaptic transmission, and the remainder can reach the astrocytes for glutamate re-uptake [170]. The impairment of astrocytic glutamate transporters leaves neurons highly susceptible to excitotoxicity [171]. In astrocytes, approximately 80% of the glutamate is converted into glutamine by two exergonic steps: (i) the γ-phosphate residue is transferred from ATP to glutamate, which gives rise to an “energy-rich” mixed acid anhydride, and (ii) NH3 substitutes the phosphate residue from the intermediate, and glutamine and free phosphate are produced. The energy balance of these two reactions (glutamate + ATP + NH3 → glutamine + ADP + phosphate + 17 kJ/mol) is the sum of the changes in free enthalpy of direct glutamine synthesis (∆G 0n = 14 kJ/mol) plus ATP hydrolysis (∆G 0n = −31 kJ/mol), although ATP has not been hydrolyzed.

In glutamatergic neurons, glutamine is primarily deaminated in mitochondria by glutaminase (GLS, GA, or glutamine aminohydrolase) to form glutamate, which enters synaptic vesicles to complete the glutamate-glutamine cycle. The reaction process for glutamate formation is glutamine + H_2_O → glutamate + NH_3_ [169].

Glutamine is the most abundant amino acid in blood plasma, supplying nitrogen from tissue (skeletal muscle, liver, or lung) to sites of utilization, including the brain. This suggests that dietary supplementation or parenteral nutrition can improve the outcome for critically ill patients, postsurgical patients, or those recovering from injury [159]. In many physiological circumstances, glutamine provides glutamate, which appears to promote a wider array of metabolic functions compared to glutamine. Glutamine and glutamate metabolism are as crucial as glucose metabolism in the cell due to their wide variety of metabolic roles critical for cell function [160].

An important open question is whether the alterations in glutamate uptake, recycling, and metabolism in neurodegenerative diseases are a consequence of disease progression or if they could be causative mechanisms driving the neurodegeneration and potentially targets for treatment [172,173].

Astrocytes obtain energy primarily via glycolysis, while neurons do this mainly through mitochondrial oxidative metabolism. Energy utilization in neurons is related to the activity of ion pumps, which establish electrical gradients essential for efficient neuronal activation and information transfer. Astrocytes are thus vital cells in coupling synaptic activity and energy metabolism transfer of lactate to neurons [148]. Both astrocytes and neurons are metabolically upregulated in response to increased neurotransmission [174]. Glucose is used both as an energy source for the brain and a precursor for neurotransmitters (e.g., acetylcholine, glutamate, GABA) and neuromodulators [175].

#### 3.3.2. Brain Energy Resources and Utilization in Brain Fatigue

The biochemistry of brain energy is essential to understanding how the brain functions at a cellular and molecular level. The brain’s energy metabolism is a highly coordinated process that involves glucose and alternative fuels such as lactate and ketones and complex interactions between neurons, astrocytes, and mitochondria. Disruptions in this energy balance, whether due to age, diet, or disease, can impair brain function and contribute to neurodegenerative conditions. The field is expanding, with new research showing the importance of metabolic flexibility and the potential for targeting energy metabolism to improve brain health and treat neurological disorders.

##### Energy Sources and ATP Generation

Glucose is the primary source of energy for brain functions, ATP production, oxidative stress management, and synthesis of neurotransmitters, neuromodulators, and structural components [150,176]. Inadequate energy resources give rise to pathological brain function [150,173]. The human brain consumes approximately 20% of the total energy budget and needs continuous glucose delivery [176]. Under normal physiological conditions, these cellular glucose uptake rates into neurons are controlled by brain activation [150]. Synaptic transmission needs a constant energy supply [176]; therefore, glucose metabolism is critical for information transfer and processing in the brain.

Once in the brain, glucose is metabolized in neurons and astrocytes to produce ATP, the main cellular energy currency. This process occurs through glycolysis (anaerobic) in the cytoplasm and oxidative phosphorylation (aerobic) in the mitochondria, producing ATP more efficiently. Oxidative phosphorylation generates most of the ATP needed for brain activity.

Different forms of energy, including chemical and metabolic energy, are stored/“conserved” in the form of “energy-rich” chemical bonds, measured in joules or calories (1 Cal = 4.187 J). So-called “energy currency” [177] refers to the molecule of nucleotide coenzyme adenosine triphosphate (ATP), which is the most exergonic source of chemical energy for use and storage in all cells, particularly in brain cells, to drive numerous endergonic processes, including biosynthesis, movement, ion transport, muscle contraction, nerve impulse propagation, intracellular signaling, and substrate phosphorylation [177]. The physiological energy yield of ATP hydrolytic cleavage to ADP and phosphate is about 50 kJ/mol, or Gibbs-free energy of −7.3 Cal/mol [178].

Brain cells store energy in the form of “energy-rich” molecules. Adenosine triphosphate (ATP) is the most essential chemical energy storage molecule, which drives many energy-dependent reactions via energetic coupling. The brain is the highest consumer of ATP in the body, utilizing almost twenty-five percent of the total energy available [179]. Brain cells constantly require organic and inorganic nutrients and chemical energy, mainly from adenosine triphosphate (ATP), to function and survive. The cleavage of ATP into ADP and phosphate releases energy. The physiological energy yield of ATP hydrolytic cleavage to ADP and phosphate is about 50 kJ/mol, or Gibbs-free energy of −7.3 Cal/mol [178].

Most anabolic pathways, movements, and transport processes are energy-dependent. ATP transfers phosphate residues, providing the nucleotide components for the activation reaction of proteins in metabolic and signaling pathways common to most cells and organisms. These pathways serve for the synthesis, degradation, and interconversion of essential metabolites and energy conservation.

##### ATP in Neurotransmission

Maintaining ion concentrations for neuronal signaling and synaptic transmission requires much energy. At the presynaptic terminal, ATP is needed to establish ion gradients that shuttle neurotransmitters into vesicles and prime the vesicles for release through exocytosis [173].

Neuronal signaling depends on the action potential reaching the presynaptic terminal, signaling the release of the loaded vehicles. This process depends on ATP restoring the ion concentration in the axon after each action potential, allowing another signal to occur. Active transport resets concentrations of the sodium and potassium ions to baseline after an action potential occurs by the Na^+^/K^+^ ATPase. During this process, ATP is hydrolyzed, releasing energy required for moving sodium and potassium ions against their concentration gradients, transporting sodium ions out of the cell, and two potassium ions, which are transported back into the cell. When reaching the presynaptic terminal, action potentials traveling down the axon initiate vesicular release. After establishing the ion gradients, the action potentials propagate down the axon through the depolarization of the axon, sending a signal toward the terminal. Approximately one billion sodium ions are necessary to propagate a single action potential. Neurons must hydrolyze nearly one billion ATP molecules to restore the sodium/potassium ion concentration after each cell depolarization [179]. Excitatory synapses largely dominate the gray matter of the brain. Vesicles containing glutamate will be released into the synaptic cleft to activate postsynaptic excitatory glutaminergic receptors. Loading these molecules requires large amounts of ATP because nearly four thousand glutamate molecules are stored in a single vesicle [179]. Significant energy stores are necessary to initiate the vesicle release, drive the glutamatergic postsynaptic processes, and recycle the vesicle and the left-over glutamate [179]. Therefore, due to the large amounts of energy required for glutamate packing, mitochondria are close to glutamatergic vesicles [177].

##### ATP in Intracellular Signaling

Signal transduction heavily relies on ATP. ATP can serve as a substrate for the most numerous ATP-binding protein kinases. When a kinase phosphorylates a protein, a signaling cascade can be activated, leading to the modulation of diverse intracellular signaling pathways [180].

In addition to kinase activity, ATP can be a ubiquitous trigger of intracellular messenger release [6]. These messengers include hormones, enzymes, lipid mediators, neurotransmitters, nitric oxide, growth factors, and reactive oxygen species [181].

An example of ATP utilization in intracellular signaling can be observed in ATP acting as a substrate for adenylate cyclase. This process mainly occurs in G-protein-coupled receptor signaling pathways. Upon binding to adenylate cyclase, ATP converts to cyclic AMP, which assists in signaling the release of calcium from intracellular stores [182]. cAMP has other roles, including secondary messengers in hormone signaling cascades, activation of protein kinases, and regulation of the function of ion channels.

#### 3.3.3. Brain Energy Supply and Metabolism in Stress, Stroke, Obesity, Diabetes, and Aging Disorders

The brain requires a continuous supply of energy in the form of ATP, most of which is produced from glucose by oxidative phosphorylation in mitochondria, complemented by aerobic glycolysis in the cytoplasm. When glucose levels are limited, ketone bodies generated in the liver and lactate derived from exercising skeletal muscle can also become important energy substrates for the brain. In neurodegenerative disorders and aging, brain glucose metabolism deteriorates in a progressive, region-specific, and disease-specific manner [183]. Brain energy rescue therapy aims to restore oxidative phosphorylation and glycolysis, increase insulin sensitivity, and correct mitochondrial dysfunction and ketone-based interventions, acting via hormones that modulate cerebral energetics, RNA therapeutics, and complementary multimodal lifestyle changes [183].

Brain energy metabolism and supply in demand concept [184] suggest the brain actively draws energy from the body when needed. Based on a selfish-brain theory, a model is used to understand the brain as passively receiving energy and actively procuring energy for itself on demand. This active model predicts and coherently explains all data examined, which included stress, sleep, caloric restriction, stroke, type 1 diabetes mellitus, obesity, and type 2 diabetes.

The “brain energy on demand” model aligns with the stress-system concept, assuming the central role of the neuroendocrine-immune system in adaptive stress response [79,80].

The brain-pull mechanisms, functions (a–h), neuroendocrine effectors within the stress system (SNS = sympathetic nervous system, HPA axis = hypothalamus–pituitary–adrenal axis), and energy carriers (glucose, lactate, ketones) that improve brain supply are summarized as follows [184]:(a)Suppresses ß-cell insulin secretion (‘cerebral insulin suppression’; CNS) and thereby reduces insulin-dependent glucose uptake via glucose transporter GLUT4 into muscle and subcutaneous fat tissue but increases insulin-independent glucose uptake via GLUT1 into the brain, SNS/HPA axis, and glucose.(b)Increases α-cell glucagon secretion, increasing hepatic glucose output, SNS, and glucose.(c)Reduces insulin-dependent glucose uptake into adipose tissue by inhibiting GLUT4 translocation, SNS, and glucose.(d)Increases muscular proteolysis and hepatic gluconeogenesis, SNS/HPA axis, and glucose.(e)Increases subcutaneous lipolysis, providing free fatty acids for cardiac and skeletal muscles, making more glucose available for the brain, SNS/HPA axis, and glucose.(f)Increases visceral lipolysis and hepatic ketogenesis, SNS/HPA axis, and ketones.(g)Increases muscular lactate release, SNS, and lactate.(h)Increases heart rate and thus enhances cardiac output, SNS, glucose, lactate, and ketones.

The essential role in the brain energy supply on-demand model is ATP-mediated distorted regulation associated with cerebral artery occlusion, which can lead to strokes [184] and brain fatigue. Energy sensors in brain regions (amygdala and ventromedial hypothalamus, VMH) detect the lowest drop in ATP and activate the brain-pull mechanisms: decrease plasma insulin levels and increase blood glucose concentrations. A minor ATP decline leads to GABAergic disinhibition of neurons. In turn, VMH neurons release glutamate, activating SNS and HPA. As a result, the SNS and HPA axis then suppress insulin secretion from pancreatic ß-cells. Furthermore, cerebral artery occlusion increases the systemic blood glucose level, as predicted by the brain energy on the model [184].

Stress and distress significantly affect ATP production in the brain and body, which vary depending on the duration and intensity of the stress and the organism’s resilience. Here is a breakdown of how stress influences ATP production and the role of increased ATP in the adaptive responses to stress. Acute stress typically triggers an increase in ATP demand to support the “fight-or-flight” response. This response is regulated by releasing stress hormones such as adrenaline and cortisol, which elevate heart rate, respiration, and blood flow to muscles and the brain. The brain requires more ATP during acute stress to support heightened alertness, faster processing, and other adaptive responses. In response to acute stress, the body mobilizes glucose (via glycogenolysis and gluconeogenesis) and fatty acids for rapid ATP production. This quick energy mobilization boosts ATP generation to meet immediate demands. The increase in ATP demand during acute stress is usually met by the upregulation of mitochondrial activity in cells, especially in neurons and muscle cells, leading to enhanced oxidative phosphorylation.

In acute stress, the increase in ATP production helps maintain optimal neuron function, allowing the brain to stay alert, focused, and responsive. Elevated ATP production in muscles supports physical actions that may be needed to respond to threats. This increase in ATP production serves as a short-term adaptive mechanism that prepares the body to face or escape danger. Essentially, it is a defense response, enabling rapid action and decision-making.

During prolonged or chronic stress, the body initially tries to maintain increased ATP production to meet ongoing demands. However, persistent stress can eventually exhaust energy reserves and impair mitochondrial function. Chronic stress is associated with mitochondrial dysfunction, leading to decreased efficiency in ATP production over time. This is partly due to the increased production of reactive oxygen species (ROS), which damages mitochondrial components and reduces their ability to produce ATP efficiently. As chronic stress continues, ATP production may decrease, contributing to feelings of fatigue, reduced cognitive function, and weakened physical performance. In the brain, impaired ATP production can lead to memory, learning, and emotional regulation difficulties.

Chronic stress increases ROS and oxidative stress, damaging cellular components such as lipids, proteins, and DNA. This damage further compromises ATP production and energy metabolism in the brain and other organs. With persistent stress, the initial adaptive increase in ATP production becomes unsustainable, leading to a failure of the adaptive response. Mitochondrial biogenesis (the production of new mitochondria) may be hindered under chronic stress, compromising cellular energy reserves.

Physical fitness, good sleep, and dietary support can enhance mitochondrial health and support ATP production during stress. Exercise promotes mitochondrial biogenesis and can increase resilience to stress by maintaining a more robust ATP production capacity.

Mild, controlled stress (like exercise or intermittent fasting) can sometimes stimulate mitochondrial function and enhance ATP production, creating a form of resilience. This phenomenon, known as hormesis, can improve the body’s ability to handle future stressors.

In summary, acute stress initially increases ATP production as an adaptive defense response, helping the brain and body mobilize resources and respond effectively. Chronic stress ultimately reduces ATP production due to mitochondrial dysfunction, oxidative damage, and depletion of energy reserves, which impairs the body’s ability to cope with continued stress. In essence, while an increase in ATP production is an adaptive response to acute stress, prolonged or unmitigated stress leads to a decline in ATP production, compromising cellular function and increasing susceptibility to stress-related diseases.

## 4. Overview of Stress-Protective and Anti-Fatigue Effects of Adaptogens

### 4.1. Adaptogens and Adaptive Stress Response

The term adaptogen is primarily associated with the notion of adaptability introduced by George Canguilhem in 1943 [185], who defined adaptability as the ability of an organism to alter itself or its responses to the changed environment or circumstances and assumed that adaptability shows the ability to learn and improve from experience—repeated mild exposure or low doses of stress result in the increased resistance of cells and organisms to subsequent stress exposure, resulting in an adaptation that favors survival. The adaptation phenomenon to repetitive low-level stress was first described by Hans Selye in 1936 using rats exposed to low temperatures, low oxygen tension, muscular exercise, adrenaline, and morphine. Several nonspecific reactions were evoked (thymus atrophy, adrenal hyperplasia, stomach ulceration, increased secretion of cortisol and catecholamines, etc.), which Selye termed the general adaptation syndrome [85]. In a broad aspect, adaptation is an active process of responding to challenges, and adaptedness resulting from this process means achieving a positive outcome, i.e., survival and reproduction, in the face of adversity. In this context, the term adaptogen refers to a time-dependent physiological process of adaptation of the organism in response to repeated administration of a botanical that triggers intracellular and extracellular adaptive signaling pathways and adaptive stress response (Figure 4) [31,32,186,187].

In 1957, Lazarev introduced the term “adaptogen” for some chemicals, increasing the so-called “state of nonspecific resistance” of the organism [25,87,188] and implementing “Ginseng-like” [189] tonic, stress-protective, harmless botanicals normalizing body function [190], increasing the adaptability to damaging environmental factors [25,68]. The concept of adaptogens has been thoroughly reviewed regarding physiology, pharmacology, toxicology, and potential medical uses [25,26,31,33,34,66,67,87,88,89,186,190,191,192,193]. The definitions of adaptogens were upgraded and updated depending on the progress in neuroscience, molecular biology, pharmacology, and the knowledge in phytotherapy research [25]. Further studies suggest that adaptogens increase survival and resilience in stress and aging by triggering intracellular and extracellular adaptive signaling pathways of cellular and organismal defense systems (stress system) and the neuroendocrine-immune complex, where the hypothalamic–pituitary–adrenal (HPA) axis provides a rapid response, defense against stress, and adaptation to stress (Figure 4, Figure 5, Figure 6, Figure 7 and Figure 8). It was demonstrated that adaptogens trigger the generation of hormones (cortisol, corticotropin-releasing hormone (CRH), gonadotropin-releasing hormones, urocortin, neuropeptide Y), playing critical roles in metabolic regulation and homeostasis and protecting against stress-related damage without overstimulating or suppressing normal biological processes [25,27,28,29,30,67,87,194,195,196,197,198].

Figure 4 shows links between the central and peripheral nervous systems, the reciprocal interactions with the endocrine system within the stress system, extra- and intracellular singalongs with the immune network [79,81,199,200,201], and the effects of adaptogens on some vital molecular targets, including molecular chaperone Hsp70 [29,194,195,196,197,198], and neuropeptide NPY [194,196]. The key mediators and effectors of the adaptogenic activity of adaptogens are CRH [197], cortisol [27,198,202,203,204], neuropeptide Y, molecular chaperones Hsp70, stress-activated protein JNK [27], monoamines [204,205,206,207], and melatonin [67]. Figure 4 provides additional information on this. Adaptogens act as mild stressors, triggering adaptive stress response pathways of body cells regulated by the neuroendocrine-immune complex and activating the defense response of the organism after single or repeated administration in the appropriate dose range (Figure 4). Under the stressor, we imply negatively affecting environmental factors of psychological, physical, viral, bacterial, and chemical origins.

Resistance to stress and survival depends on adaptability and the thresholds determining an organism’s innate tolerance to a given stress level. Figure 5 shows a hypothetical model of regulation homeostasis response to adaptogens.

The adaptogenic effect covers various pharmacological activities (adaptogenic, stress-protective, stimulating, and pleiotropic pharmacological profile) in stress-induced and aging-related disorders. Potential indications for use and health claims of adaptogens are stress-induced fatigue, mental and behavioral disorders, infectious diseases, and aging-associated disorders (Figure 6). Figure 6 shows adaptive signaling pathways by which animals and human subjects respond to the challenges of food deprivation/fasting, running, and adaptogens.

The ability of some plant secondary metabolites to activate the adaptive stress response in the human body is one of the essential mechanisms of action of adaptogens. Figure 5 and Figure 6 illustrate the simple fundamental concept that mild stressors, including adaptogen supplementation, promote optimal health. Humans developed the ability to consume various bitter plant species, which exert mild noxious effects on cells, presumably by triggering cellular defense responses that protect cells against harmful stressors. It was hypothesized that some of the evolutionary metabolites of plants play a role in defense against various environmental stressors, including microorganisms, insects, UV, etc. At relatively small doses, they are not harmful in humans but induce mild adaptive stress responses in humans. Many natural compounds and botanicals exhibit dual (hormetic) dose-dependent reverse effects in low or high doses. For example, some plants are known as neurotoxic, e.g., *Aconitum napellus*, *Atropa belladonna*, *Conium maculatum*, *Oenanthe crocata*, and *Ricinus communis* [208], and are used in small doses in homeopathy. In contrast, other plants, including *Ginkgo biloba*, *Curcuma longa*, *Panax ginseng*, *Withania somnifera*, *Rhodiola rosea*, *Centella asiatica*, and other adaptogenic plants, have neuroprotective activity on the human brain, demonstrating antioxidant, anti-inflammatory, and cognitive-enhancing properties, making them promising candidates for combating neurodegenerative diseases and improving brain function. Meanwhile, the same plant, e.g., *Bryonia alba* L., can be toxic (freshly collected in summer) or harmless (collected in fall and dried), depending on the harvesting season and processing methods of herbal substances, due to the metabolism of toxic cucurbitacin I converted to nontoxic metabolites tetrahydro cucurbitacin glucosides [209].

One hypothesis of evolutionarily developed adaptive responses to stressors is that plants protect themselves against microorganisms, insects, pests, fungi, viruses, and hazardous environmental changes by biosynthesis of secondary metabolites in their most vulnerable parts [210,211,212,213,214,215]. Plant secondary metabolites play a role in defense and adaptive response against various environmental stressors. Herbivorous and omnivorous animals that rely on plants as a primary source of nutrients have evolved complex mechanisms to neutralize the potentially harmful effects of phytochemicals. These natural compounds are not toxic in people at relatively small doses but still induce mild cellular stress responses [215,216]. One primary mechanism of action of plant secondary metabolites is that they activate the adaptive cellular stress response pathways in humans [217]. This phenomenon has been commonly observed and has been described as an adaptive stress response, pre-conditioning, or ’hormesis’ [217,218,219,220]. Major components of the hormetic response include various stress resistance proteins, such as heat-shock proteins (HSPs), antioxidants, and growth factors [210,212,221].

Adaptive stress response is essential in cell maturation, with mild stress initiating repair and maintenance mechanisms to protect cells against subsequent stresses. In contrast, chronic stress induces progressive failure of these mechanisms, leading to cellular senescence, aging, and death. With cellular maintenance on overdrive, the organism can continue to protect itself from chronic inflammation, which causes a range of serious illnesses, particularly aging-related diseases. The adaptive stress response is a survival mechanism (Figure 6, Figure 7 and Figure 8).

In the 1960s–1970s, several botanicals, namely *Schisandra chinensis*, *Panax ginseng*, *Eleutherococcus senticosus*, and *Rhodiola rosea*, were extensively studied and were incorporated into official medical practice in the USSR as stimulating tonic, restorative, and antistress medications. For instance, *Rhodiola rosea* (Golden Root, Arctic Root) was used in a liquid dosage form (extract, DER 1:1, extraction solvent—40% ethanol) as conventional medicine in the USSR since 1974 as a CNS stimulant in asthenic conditions, increased fatigue, neurasthenic conditions, and somatic or infectious diseases in patients with functional disorders of the nervous system, as well as in healthy people with asthenia and decreased performance [222]. Despite significant differences between the effects of adaptogens and other CNS stimulants, in the USSR, adaptogens have been recognized as a distinct group of tonic substances in a group of CNS stimulants [33]. In self-care, *E. senticosus*, *R. rosea*, *S. chinensis*, and *B. alba* were used as stimulants or tonics in states of fatigue and stress in sports medicine to prevent and treat injuries and other somatic conditions. Another use of these four herbs is in occupational medicine, such as for protection against adverse environmental factors, including exposure to low temperature in polar regions and to high noise levels and mechanical vibration in heavy industrial work; in mining; and medicine for treating acute hepatic poisoning, ischemia from oxygen deprivation, and for accelerating recovery after surgery [223,224]. Adaptogens were also used as curative agents in treating some neurologic and psychiatric disorders, such as asthenia, neurosis, depression, and alcoholism, and in some other conditions, as well as being prescribed as adjuvants to other medicines in diseases such as tuberculosis and conventional cancer therapy. Overall, “one drug for one disease” does not apply to adaptogens [33].

In the USA, herbalists widely implemented adaptogens in alternative medicine [225,226,227,228,229,230] and included them in textbooks on Pharmacognosy [231,232,233]. Often, however, the term “adaptogen” was carelessly applied to some botanicals without sufficient experimental evidence to support the criteria for the formal definition of adaptogens as natural compounds or plant extracts that increase the adaptability, resilience, and survival of organisms to stress.

In Europe, the European Medical Agency adopted the EC monographs of *Rhodiola rosea* L. [234,235], Ginseng [236], and Eleutherococcus [237] as traditional herbal medicinal products (THMP) for temporary relief of symptoms of stress, such as fatigue, exhaustion, and sensation of weakness [234,235], and symptoms of asthenia, such as fatigue and weakness [236,237]. Many commercially available R. rosea mono-drug preparations are evidenced to benefit burnout patients or subjects with work- or stress-related fatigue after repeated doses or a single dose. However, in the Assessment Report on *Rhodiola rosea* L., rhizome et radix, the Committee on Herbal Medicinal Products (HMPC) concluded that “The traditional use as an adaptogen for the relief of symptoms of stress such as fatigue and exhaustion is appropriate for traditional herbal medicinal products”; however, ’well-established use’ cannot be accepted. The studies in models investigating antifatigue and stress-protective effects indicate a possible relation to traditional medicinal use. The clinical trials do not give reasons for special safety concerns but exhibit considerable deficiencies in their quality. Due to the missing published data on genotoxicity, the development of a European Union List Entry cannot be supported”. Further evidence is required to support the effectiveness of sufficiently characterized standardized herbal preparations of Rhodiola, ensuring reproducible quality in well-designed clinical trials to qualify them as herbal medicinal products with well-established use [238].

Considering the active components, adaptogenic preparations contain two major groups: (i) phenolic compounds such as phenylpropanoids, phenylethane derivatives, and lignans, which have chemical structures similar to catecholamines, suggesting an effect on the sympathoadrenal system and possibly imply an effect in the early stages of the stress response; (ii) tetracyclic and pentacyclic triterpenes, such as ginsenosides, withanolides, cucurbitacines, and andrographolides, which structurally resemble the catabolic hormones corticosteroids that inactivate the stress system to protect against overreaction to stressors; and anabolic-androgenic steroids structurally related to testosterone and estrogen androgenic steroid hormones which regulate lipids and sugar metabolism (Table 2). The first group of adaptogens includes *Eleutherococcus senticosus* (Rupr. & Maxim.) Maxim, *Rhodiola rosea* L., *Schisandra chinensis* (Turcz.) Baill., and *Sideritis scardica* Griseb. The second group contains the substances in extracts of *Panax ginseng* C.A. Mey., *Withania somnifera* (L.) Dunal, *Andrographis paniculata* (Burm.f.) Nees, *Bacopa monnieri* (L.) Wettst., *Bryonia alba* L., *Codonopsis pilosula* (Franch.) Nannf., and *Rhaponticum carthamoides* (Willd.) Iljin [25,33,87].

The chemical composition of the group of (i) phenolic-rich adaptogens is characterized by a high content of compounds containing four common types of structural fragments (pharmacophores) covalently incorporated into the chemical structures of some active principles of adaptogenic plants, which are structurally similar to catecholamines (epinephrine, dopamine, norepinephrine) and the tyrosine fragment of neuropeptide Y, suggesting that these biased ligands have high affinity and allosterically compete for receptor sites of proteins involved in signaling pathways and cellular responses [24,33,87,194].

Notably, the various botanical products have product-specific chemical “fingerprints” (qualitative and quantitative compositions) corresponding to various pharmacological profiles (“conditional signatures”) derived from the same medicinal plant and different characteristics of quality (based on product specifications), efficacy, and safety [69].

The pharmacological activity and therapeutic safety of adaptogens and purified compounds, predominantly tetracyclic triterpenes, phenethyl- and phenylpropanoids, lignans, etc., were studied in humans, animals, and isolated cells, revealing pleiotropic biological activities and multitarget effects on the neuroendocrine-immune complex, including stress modulatory, antioxidant, anti-fatigue, nootropic, immunomodulatory, cardiovascular, neuroprotective, radioprotective, and other activities [25,46,49,50,256].

### 4.2. Multitarget and Pleiotropic Effects of Adaptogens

Our knowledge of biochemical mechanisms underlying brain fatigue, discussed in Section 3.3, is based on the fundamentals of the physiology of human organisms, and molecular biology was revealed by conventional methods of pharmacology and biology. Meanwhile, novel molecular biology and computational informatics methods unveil the complexity of physiological processes and the plethora of new mediators of their intra- and inter-cellular communications within neuroendocrine-immune and cardiovascular systems. For instance, inflammation covers several levels of interactions of many mediators of the inflammatory response, systematized in the atlas of inflammation, where various bioactive molecules play a role and can be a pharmacological target for pharmacological intervention. One task was identifying and purifying pharmacologically active compounds targeting a selective receptor free of adverse effects. Uncovering the mechanism of actions of adaptogens comprising multicomponent plant extract is even more challenging. The one-drug-one-target paradigm, commonly explored for Galenic preparations by extracting one or more active constituents of a plant, was based on a reductionistic concept of dissecting biological systems into their constituent parts. However, even purified active compounds, e.g., ginsenosides, salidroside, rosavin, and withanolides (Table 2), are not selective in their action, interacting with other receptors and molecular networks associated with various physiological functions.

The classical reductionist model, which presumes a specific receptor/drug interaction, is unsuitable for understanding adaptogens’ molecular mechanisms of action associated with the physiological notion of “adaptability” [25]. On the contrary, systems biology and network pharmacology concepts provide ideal mechanistic tools for understanding and conceptualizing adaptogen modes and mechanisms of action.

On the contrary, the multi-target network pharmacology-systems biology concept and innovative drug discovery tools, such as transcriptome-wide microarray gene expression profiling in vitro testing and artificial intelligence in silico, provided theoretical models that propose using botanical adaptogens to treat stress-induced disorders, including brain fatigue and related neurological diseases.

The advantages of network pharmacology and the systems biology approach vs. the ligand–receptor-based reductionist concept were discussed recently in several reviews [66,69,257,258,259,260].

Several recent studies of gene expression in neuroglia and neuronal cells unveiled the polyvalent and synergistic actions of botanical hybrid preparations (BHP) in stress-induced disorders, suggesting that adaptogens exhibited multitarget and pleiotropic effects via HPA- and GPCR-mediated signaling pathways [66,67,69]. Multiple molecular targets of adaptogens, molecular networks, and adaptive stress response signaling pathways were identified [61,62,63,64,65,66,67,68,261]. They are associated with chronic inflammation, atherosclerosis, neurodegenerative cognitive impairment, metabolic disorders, and cancer, which are more common with age [25,66] (Figure 8).

Overall, the mechanism of adaptogenic activity of adaptogens is characterized by a multitarget effect on the neuroendocrine-immune complex (stress system) [25,34,66,67,87], including the following:Triggering of intracellular and extracellular adaptive signaling pathways that promote cell survival and organismal resilience in stress;Regulation of metabolism and homeostasis via effects on expression of stress hormones (corticotropin and gonadotropin-releasing hormones, urocortin, cortisol, neuropeptide Y, heat shock proteins Hsp70) and their receptors.

Opponents of the adaptogenic concept reduce and rectify the effects of adaptogens on selected molecular targets or specific modes of action, underestimating their interactions within the networks of mediators of the neuroendocrine-immune complex that, in turn, regulates other pharmacological systems (cardiovascular, gastrointestinal, reproductive systems) due to numerous intra- and extracellular communications and feedback regulations. These interactions result in polyvalent action and the pleiotropic pharmacological activity of adaptogens, which is essential for characterizing adaptogens as distinct types of botanicals. They trigger the activation of adaptive response signaling pathways and, consequently, defense response to various stressors, increasing the resilience and survival of organisms (Figure 4).

Adaptogens trigger pleiotropic genes, molecular mechanisms, and cellular signaling pathways that mediate adaptive and defense responses. This results in multitarget modes of action simultaneously and nonspecific pleiotropic pharmacological activity [66]. Pleiotropy results from the effect of adaptogens on a single gene that impacts multiple signaling pathways, biological processes, physiological functions, and phenotype characteristics. Various cells use the gene transcription mechanism to trigger numerous downstream signaling pathways and molecular networks that collectively affect multiple molecular targets, resulting in polyvalent pharmacological activities (nonspecific effect) [66].

The pharmacological activities of adaptogens (Table 3) depend on their molecular mechanisms of action, including specific effects on the expressions of genes (Appendix A in the Appendix A), encoding proteins of adaptive stress response signaling pathways (Appendix A in the Appendix A), and networks involved in the modes of the pharmacological action, including the regulation of biological processes and physiological and cellular functions (Table 4), which are associated with progression of stress-induced and aging-related diseases (Table 5) as depicted on the flowchart below [66] (Figure 8).

Table 3, Table 4, Table 5 and Appendix A show predicted and evidence-based health claims and indications for the therapeutic use of adaptogens in various diseases, their pharmacological activities, their effects on biological processes, physiological and cellular functions, canonical signaling pathways, and several essential genes triggering the effects of adaptogens.

#### 4.2.1. Effects of Adaptogens on Neuroinflammation Signaling Pathways

One of the animals’ most crucial defensive signaling molecules is the TLR family protein TLR9, which occurs upstream in 152 signaling pathways [67]. Excessive activation of TLR signaling could be harmful and lead to tissue injury, including chronic inflammation and autoimmune diseases [262]. However, it is now evident that mammalian TLRs play a prominent role in the direct activation of host defense mechanisms. Activation of TLRs stimulates an innate immune response, which involves the production of direct antimicrobial effector molecules, including NO. It increases an adaptive immune response by inducing the production of IL-1h, IL-6, TNFα, and IL-12, which augments cell-mediated and humoral immune responses. TLRs play crucial roles in the innate immune system by recognizing pathogen-associated molecular patterns derived from various microbes [262].

Perhaps most common for adaptogens are their effects on genes encoding protein kinases and phosphatases, which control the level of active forms of signaling molecules. Among these were the mitogen-activated protein kinase 10 (JNK-3), mitogen-activated protein kinase 13 (related to p-38 MAP kinase), protein kinase C η, tyrosine kinases FLT1, MERTK, and ROS1, and tyrosine phosphatases PTRRD and PTPRR [67].

Among transcription regulators significantly modulated by adaptogens were signal transducer and activator of transcription 5A (STAT5A), Fos proto-oncogene, AP-1 transcription factor subunit (FOS), forkhead box O6 (FOXO6), scleraxis bHLH transcription factor (SCX), and zinc finger proteins. The stress-activated protein kinase MAPK13 is responsive to various stress stimuli, such as cytokines, ultraviolet irradiation, heat shock, and osmotic shock, and is involved in cell differentiation, apoptosis, and autophagy [67].

Persistent aging-related activation of the p38 MAPK pathway in muscle satellite cells (muscle stem cells) impairs muscle regeneration [263]. Among the genes encoding metabolic enzymes and chaperone proteins were several involved in many signaling pathways and molecular networks with key roles in metabolic regulation and cellular repair functions, such as guanylate cyclase 1 soluble subunit α 2 (GUCY1A), heat shock protein family A (Hsp70) member 6 HSPA6, lactate dehydrogenase D (LDHD), lipase E, hormone-sensitive type (LIPE), and phosphodiesterase 3B (PDE3B).

Adaptogens significantly affect gene expression in neuronal cells, influencing pathways related to stress response, neuroprotection, neuroplasticity, and neurotransmitter regulation.

The molecular mechanisms behind these effects are increasingly studied, especially in the context of neurodegenerative diseases, cognitive function, and mental health. They include AKT protein kinase B (PKB), AP-1 activator protein-1, BDNF brain-derived neurotrophic factor, COX cyclooxygenase, ERK extracellular signal-regulated kinase, GDNF glial-derived neurotrophic factor, GFAP glial fibrillary acidic protein, GSK-3 glycogen synthase kinase-3, HO-1 hemoxygenase-1, IFN-γ interferon-γ, iNOS inducible NO synthase, JAK/STAT janus kinase/signal transducer and activator of transcription, JNK c-Jun N-terminal kinase, MAPK mitogen-activated protein kinase, MCP-1 monocyte chemoattractant protein-1, NF-κB nuclear factor kappa-light-chain-enhancer of activated B cells; NLRP nucleotide-binding oligomerization domain, leucine-rich repeat, and pyrin domain, NO nitric oxide, NOS NO synthase, NPY neuropeptide Y, Nrf-2 nuclear factor erythroid 2-related factor 2, PGE2 prostaglandin E2, PI3K phosphatidylinositol-3 kinase, PPAR-γ proliferator-activated receptor γ, PRR pattern recognition receptor, TGF-β transforming growth factor β, TLR toll-like receptor, TNF-α tumor necrosis factor-α, TNFR tumor necrosis factor receptor, TRADD TNFR1 signal transducer, TRAF TNFR-associated factor, VEGF vascular endothelial growth factor, Interleukins IL-1β, IL-4, and IL-6 are shown on the neuroinflammation signaling pathways in Appendix A (in the Appendix A), activating many cellular and physiological functions, including both harmful ones such as ROS production, oxidative stress, neuron damage, Aβ generation, BBB disruption, and beneficial defense processes including neuroprotection, microglia activation and survival, post-synaptic neuron apoptosis, neurogenesis, phagocytosis of damaged neurons, microglial proliferation, Aβ clearance, etc. [67].

In the brain cells, inhibition of leukotriene signaling is highly associated with inhibition of neuroinflammation and neurodegeneration, increased neuronal survival, activation of neurogenesis, synaptic integrity, and integrity of the blood–brain barrier, decreased tau phosphorylation, and amyloid β plaque load—all the pathological hallmarks of Alzheimer disease [264]. There is a strong rationale for targeting the leukotriene signaling system, which mediates various aspects of AD pathology (Figure 9) [264].

Adaptogenic herbs affecting all these issues simultaneously by a pleiotropic mechanism might be a much better strategy to treat dementia compared with the monospecific approaches of the past decades, most of which, if not all, have entirely failed. Notably, 5-LOC levels and the activity of the leukotriene pathway are elevated in the brain with age, likely contributing to age-related CNS diseases [265].

The adaptogens modulate the expression of genes in stress and aging-related disorders, characterized by an imbalance between pro- and anti-inflammatory eicosanoids in low-grade systemic inflammatory conditions. A study of transcriptome-wide RNA sequencing to profile gene expression alterations in T98G neuroglia cells upon treatment with plant extracts shows that all tested adaptogen extracts revealed a specific “signature” on eicosanoid signaling and shared features for adaptogens and other anti-inflammatory plants [68]. The common feature for all tested anti-inflammatory plant extracts was related to the downregulation of the ALOX12 gene, which is associated with the neuroprotective action of these medicinal plants and their potential benefits in neurodegenerative diseases. *Rhodiola rosea*, *Withania somnifera*, and *Eleutherococcus senticosus* extracts downregulate the expression of ALOX5AP, DPEP2, and LTC4S genes involved in the biosynthesis of leukotrienes A, B, C, D, and E, resulting in inhibition of the leukotriene signaling pathway and suggesting their potential benefits in Alzheimer disease (Figure 9) [68].

Adaptogens modulated the expression of 14 genes encoding the proteins playing key roles in eicosanoid signaling [68]. All of them downregulated the expression of arachidonate 12-lipoxygenase, 12S type (ALOX12). The most potent effect exhibited by Eleutherococcus is decreasing gene expression by 7.4-fold. A similar magnitude effect was observed for the BHP combination of Withania with melatonin. However, the downregulation of ALOX12 expression is not specific for adaptogens since other tested anti-inflammatory plant extracts (curcumin and Boswellia) also attenuated ALOX12 expression, suggesting that all these plants may inhibit 12-HETE-mediated neurotoxicity. Some synergistic and antagonistic interactions between Withania, Rhodiola, and melatonin have been documented [67,68], including the unexpected potential benefits of the fixed combination in Alzheimer’s disease. A promising direction of further studies is related to the use of the combination of Rhodiola with Withania (Adaptra^®^), which inhibits the LTC4 signaling pathway by downregulation of expression of the leukotriene C4 synthase (LTC4S) gene and upregulating the PTGER3 gene for the treatment of allergic asthma and Alzheimer disease. The correlation between the effects of anti-inflammatory or adaptogenic plants on the eicosanoid signaling pathway was related to the downregulation of ALOX12, suggesting that it is one possible mechanism of the neuroprotective action of these medicinal plants as well as their potential benefits in neurodegenerative diseases [68].

#### 4.2.2. The Dose Matters: Hormetic Dose-Dependent Reversal Effects of Adaptogens

The pharmacodynamics of adaptogens is characterized by their interaction with networks of endogenous regulators and mediators of adaptive stress response, a time-dependent process of maintaining homeostasis. They induce an adaptive response to stress by triggering adaptive signaling pathways in the neuroendocrine-immune complex.

The concentration of active constituents of adaptogens, e.g., *Rhodiola rosea* preparations, in the organisms varies over time after their intake [266,267,268], and their response is reversal depending on the dose [88,269], commonly known as the hormetic response [210,211,212,213,214,215,216,217,218,219,220,221,270,271]. For example, Rhodiola exhibits the biphasic shape of the dose–response (hormetic) action in many studies [63,88,269], but this dual action is insufficiently understood due to the complexity of the stress system, including multiple “players” involved in stress response on various levels of regulation. The typical pattern of dose–response effects of *Rhodiola rosea* extracts was observed on the duration of thiopental-induced sleep in mice, ranging from stimulation at low doses (10 mg/kg), where the sleep period was reduced by 12.5 times, to sedation at high doses (500 mg/kg), where the sleep period was increased 3-fold [88]. Calabrese et al. (2023) reviewed the dose–response relationship of *Rhodiola rosea* extracts and one of its major constituents, salidroside, and evaluated their capacity to induce hormesis/hormetic effects [269]. The authors conclude that the findings indicate that the *Rhodiola rosea* extracts and salidroside commonly induce hormetic dose responses within a broad range of biological models and cell types and across a wide range of endpoints, with particular emphasis on longevity and neuroprotective endpoints [269]. It should be emphasized that the results of available dose–response studies demonstrate that Rhodiola is not toxic in the highest tested doses despite the decrease or lack of positive effect in the highest dose. That is typical for adaptogens and different toxic poisons, which could have beneficial positive effects in low doses (Figure 10). Only in a few of the numerous in vitro tests did the Rhodiola extract and salidroside exhibit negative effects in concentrations that are too high, which have no therapeutic significance. It is generally accepted that in vitro studies with concentrations of active compounds beyond the IC50 concentration of 25 µM for pure compounds and the IC50 of 100 µg/mL for the extracts have no therapeutic significance because they could not be implemented in vivo and are not appropriate for further pharmaceutical development [272]. The results of the dose–response effect of Rhodiola studies in rats are confusing: the dose of the maximal positive effect differs 200-fold: 15 mg/kg BW [51,273,274,275,276] vs. 3000 mg/kg BW [277].

The term “hormesis”, derived from the Greek word for exciting, was initially defined as a biphasic dose–response curve [219]. Hormesis is a biological phenomenon where a low dose of a potentially harmful stressor, such as a toxin or environmental factor, stimulates a beneficial adaptive response in an organism. In other words, low doses of toxins that would be damaging in high doses can enhance resilience, promote growth, or improve health at lower levels [278]. The recent update [220] substantially revised and expanded the hormesis concept, which has long been framed within the context of stimulating adaptive cellular stress responses following exposure to low doses of conditioning/priming agents [210,211,212]. The updated so-called “catabolic–anabolic cycling” hormesis model consists of two complementary phases: catabolic (adaptive stress responses and conservation of resources) and anabolic (growth and plasticity), referring to the time-response instead of the dose–response relationship [220] that is somewhat confusing.

Adaptogens exhibit a dual action over time: catabolic in the alarm of stress response and anabolic phase in the resistance phase [31,32,88,89,186] (Figure 1). On the contrary, according to Calabrese and Mattson, 2024 [220], any drug exhibits a dual dose-dependent reversal effect—stimulating in low and inhibitory in high doses; however, in different conditions: a catabolic response on the background of stress exposure or disease (e.g., in so-called “preconditioning”) and an anabolic response without stressful exposure (Figure 10). In any case, that is still in line with the adaptogenic activity for adaptogens, which are not toxic in therapeutic doses and suggest that harmful chemicals can have health-beneficial effects only in low doses (Figure 10). Notably, stressors that trigger hormesis and stress-mediated responses increase an individual’s overall adaptive response to various stressful stimuli, including drugs, natural compounds, herbs, etc. [279].

This difference between adaptogens and hormetins, which both are coined to the phenomenon of reversal effect from positive (stimulating and protective) at low doses to negative (toxic/harmful) at low doses, is in their chemical structure, which has a different affinity to specific various target receptors (Figure 10).

In comparison, the word adaptogens is derived from adaptation, which is the body’s protective/defense response to the repeated stimulating impact of mild stressors or a small dose of an adaptogen. Adaptogens are natural compounds that induce adaptive responses to stress by triggering adaptive signaling pathways in the neuroendocrine-immune complex. A characteristic feature of adaptogens is the lack of toxic/harmful effects at therapeutically meaningful high doses despite decreased efficacy at high doses (Figure 10).

The theoretical background of hormesis is related to the hypothesis of interactions of biologically active compounds (ligand/intervention/drug) with two target proteins (receptors) that have functionally opposite responses at different concentrations. It is proposed that a drug acting as a competitive antagonist at either or both of the receptors changes the relationship between the two opposing concentration-effect curves, resulting in potentiation, antagonism, or reversal of the observed effect; the theoretical model suggested that the total impact on the system can be obtained by the algebraic summation of the two effects resulting from the activation of the two opposing receptor populations providing a classical hormesis biphasic curve [279].

However, in practice, dose–response patterns are significantly complicated due to many other interactions with (i) multiple targets (receptors) of different affinities to the active compound, (ii) other regulatory proteins or mediators in the networks involved in the adaptive stress response, (iii) feedback down regulations in the molecular signaling pathways, and/or (iv) metabolic transformation of active ligands into metabolites, a secondary ligand, which have different affinities to various receptors of the adaptive stress response [69].

In one study, we aimed to find a possible quantitative threshold of saturation of the ligand-receptor interaction, which possibly is associated with the toxic active concentration (Cmax) and the minimal active concentration (Cmin), where a selective ligand-receptor interaction can be observed in this in vitro model [63]. Figure 11 shows a striking observation of a dose-dependent reversal effect of purified Ginsenoside Rg5 from *Panax ginseng* on gene expression in murine hippocampal neuronal cell line HT22 incubated in a wide range of Rg5 concentrations from 10^−4^ to 10^−18^ M. This observation suggests that Rg5 was pharmacologically active in a wide range of concentrations of neuronal cells, from 10^−6^ M to 10^−18^ M (Figure 11c), and had a significant impact on the gene expression of hippocampal neurons.

Ginsenoside Rg5 exhibits soft-acting effects on gene expression of neuronal cells in a wide range of physiological concentrations and a strong reversal impact at high (toxic) concentrations: significant up- or downregulation of expression of about 300 genes at concentrations from 10^−6^ M to 10^−18^ M, and dramatically increased both the number of differentially expressed target genes (up to 1670) and the extent of their expression (fold changes compared to unexposed cells) at a toxic concentration of 10^−4^ M. Network pharmacology analyses of genes’ expression profiles using ingenuity pathway analysis (IPA) software (Summer release 2021 (QIAGEN Bioinformatics, Aarhus C, Denmark).) showed that at low physiological concentrations, ginsenoside Rg5 has the potential to activate the biosynthesis of cholesterol and to exhibit predictable effects in senescence, neuroinflammation, apoptosis, and immune response, suggesting a soft-acting, beneficial impact on organismal death, movement disorders, and cancer.

Notably, the number of deregulated genes by one compound (ginsenoside Rg5, one of the main active *Panax ginseng*) and the total extract of *Panax ginseng* containing 22 quantified ginsenosides tested at the same concentrations and in the same experiment were comparable [63]. This suggests that the number of deregulated genes does not proportionally increase, and the regulatory system has a limited functional capacity. This also implies that Rg5 can interact with many other receptors and indicates a somewhat saturated cell receptor population and limited capacity to maintain cellular homeostasis.

Considering that the blood level of ginsenoside Rg5 is not steady and varies over time during absorption and clearance for 24–72 h after repeated oral administration, the target cells of various tissues, including brain tissues, are continuously exposed to different doses/concentrations of ginsenoside Rg5. In this context, the difference in gene expression profiles to varying concentrations of Rg5 and predicted effects on signaling and metabolic pathways, molecular networks, and cellular physiological functions in various concentrations of Rg5 was noteworthy.

Figure 12, Figure 13 and Figure 14 show the dose reversal effect of Withania extracts on gene expression and canonical adaptive signaling pathways: corticotropin-releasing hormone (CRH), glucocorticoids, and eicosanoid signaling pathways.

The dose-effect relationship of adaptogens is typically characterized by a bell-shaped curve in many pharmacological models and, in some pharmacological models, a biphasic curve at doses that are too high. That aligns somewhat with the hormesis concept, distinguished as a biphasic reversal effect from positive (stimulating) at low doses to negative (toxic/harmful) at low doses. The drug-response interactions model has many other limitations regarding biphasic dose response. The primary mechanism in common between conditioning a hormetic and adaptive response is that it activates molecular signaling pathways that enhance the ability of the cell and organism to tolerate more severe stress. The optimal range of doses of adaptogens or conditioning treatments is like that of the hormetic zone dose–response pattern. The underlying molecular mechanisms of hormesis are not fully understood. The theoretical background of hormesis is related to the hypothesis of interactions of biologically active compounds (ligand/intervention/drug) with two target proteins (receptors) that have functionally opposite responses at different concentrations. It was proposed that a drug acting as a competitive antagonist at either or both of the receptors changes the relationship between the two opposing concentration-effect curves, resulting in potentiation, antagonism, or reversal of the observed effect; the theoretical model suggested that the total impact on the system can be obtained by the algebraic summation of the two effects resulting from the activation of the two opposing receptor populations, providing a classical hormesis biphasic curve. However, in practice, dose–response patterns are significantly complicated due to many other interactions, including the following:(i)Multiple targets (receptors) of different affinity to the active compound;(ii)Other regulatory proteins or mediators in the networks involved in the adaptive stress response;(iii)Feedback down regulations in the molecular signaling pathways and/or;(iv)The metabolic transformation of active ligands into metabolites, a secondary ligand, has different affinities to various receptors of the adaptive stress response [69].

These observations show that a particular plant extract with a product-specific quantitative and qualitative composition and HPLC fingerprint/pharmaceutical profile can exhibit different pharmacological activity, profile, and signature depending on the dose, exhibiting a dual response in the organism from positive to inactive or even negative.

### 4.3. Antifatigue Effects of Adaptogens

Many plant extracts exhibit adaptogenic, stimulating, tonic, or anti-fatigue effects on cognitive functions in experiments on experimental animals [17,18,21,22,23,25,31,32,37,39,41,43,44,46,49,50,52,53,54,55,56,57,58,59,60,280,281,282,283,284,285,286]. However, to date, only a few adaptogenic herbal preparations were studied in well-conducted randomized, placebo-controlled, double-blind clinical trials to assess anti-fatigue effects and cognitive functions [18,21,23,37,39,41,44,53,54,55,56,57,58,59,60,280,281,282,283,284,285,286]. They include *Rhodiola rosea* L. (Arctic root), *Eleutherococcus senticosus* (Rupr. and Maxim.) Maxim. (eleuthero), *Schisandra chinensis* Baill. (schisandra), *Withania somnifera* (L.) Dunal (ashwagandha), *Panax ginseng* C.A. Mey preparations, and botanical hybrid fixed-ratio combinations of Rhodiola with *Camelia sinensis* (L.) Kuntze (green tea), *Actea racemosa* L. (black cohosh), *Ginkgo biloba* L. (ginkgo), *Cordyceps militaris* L., *Ginkgo biloba* L. (ginkgo), *Crocus sativus* L. (saffron), and caffeine. These studies suggest that some preparations increase cognitive performance stress resistance and exhibit significant antifatigue effects on mental fatigue [17,18,21,22,23,54,55,280,281,282,283,284,285,286].

Referring to Rhodiola studies, numerous clinical trials were conducted in patients who were diagnosed with fatigue syndrome, aging cognitive deficiencies, mild/moderate depression, anxiety, and burnout symptoms, and in healthy subjects who experienced life stress-induced fatigue [17,18,19,42,287,288,289,290,291,292,293,294,295,296,297,298,299,300,301,302,303,304,305,306,307,308,309,310,311,312,313,314,315,316,317,318,319,320,321,322,323,324,325] and 84 reviews only on Rhodiola have been published since 2011 and confirm that Rhodiola preparations exhibit antifatigue, antidepressant, and ergogenic activity, shown by an improvement in both physical and mental performance.

The results of these studies suggest that adaptogens might be helpful in brain fatigue. Considering the concept of quality of life, it could be suggested that adaptogens might help improve patients’ quality of life with brain fatigue. Adaptogens are likely to impact many facets of physical and psychological health. However, there is not yet clinical evidence of the efficacy of adaptogens regarding the quality of life in brain fatigue that can provide evidence-based indications for adaptogens to improve the quality of life in brain fatigue.

## 5. Neuroprotective Activity of Adaptogens for Promoting Adult Neurogenesis in Aging Neurodegeneration, Post-Stroke, Traumatic Brain Injury, and Brain Fatigue

Neuroprotection, in the context of therapeutic intervention, refers to the ability of certain botanicals and phytochemicals to protect neurons from damage and neurodegeneration by sustaining their structure and function. Neuroprotection is essential in stroke, traumatic brain injuries, and various neurological conditions, including brain fatigue. Many botanicals exhibit neuroprotective effects through various mechanisms that involve combating neuroinflammation and oxidative stress and correcting neurotransmitter imbalances [208,326]. These mechanisms are crucial in maintaining neurological health and protecting against neurodegenerative disorders.

The anti-neuroinflammatory and neuroprotective effects of adaptogens in neurodegenerative diseases were extensively studied to prevent, cure, and promote recovery from inflammatory disorders, particularly in low-grade chronic neuroinflammatory disorders/diseases, including senile dementia, depression, anxiety, ischemic stroke, and viral infections, etc. [19,26,34,191,193,256]. The results of many preclinical studies of *Rhodiola rosea* and *Withania somnifera* (discussed explicitly in Section 5.2) revealed that they exhibit anti-neuroinflammatory and neuroprotective effects in vitro and in vivo experimental models of neuroinflammation, targeting numerous mediators of inflammation interacting on various levels of regulation of cellular and organismal homeostasis underlying acute inflammation initiation, transition, resolution, and repair.

### 5.1. Essential Role of Neurogenesis in Post-Stroke Recovery for Brain Fatigue

Ischemic stroke is the second leading cause of death worldwide, with a high rate of fatality and disability worldwide, which severely affects approximately 17 million individuals every year. The Global Burden of Diseases, Injuries, and Risk Factors Study 2010 (GBD 2010) estimates the global and regional burden of stroke during 1990–2010; if these trends in stroke incidence, mortality, and disability-adjusted life years continue by 2030, there will be almost 12 million stroke deaths, 70 million stroke survivors, and more than 200 million disability-adjusted life years lost globally [327]. According to the National Center for Health Statistics database [328], evidence that stroke is a leading cause of death for Americans is as follows:Every year, more than 795,000 people in the USA have a stroke.Every 40 s, someone in the United States has a stroke. Every 3 min and 11 s, someone dies of a stroke in the USA.About 87% of all strokes are ischemic strokes, in which blood flow to the brain is blocked.Stroke is a leading cause of severe long-term disability.Stroke reduces mobility in more than half of stroke survivors aged 65 and older.In the USA, in 2022, 1 in 6 deaths (17.5%) from cardiovascular disease was due to stroke.

Post-stroke brain fatigue (PSBF) is a ubiquitous and overwhelming symptom for most stroke survivors, with a prevalence ranging from 42 to 53% [329]. PSBF is a common subjective experience characterized by extreme and persistent feelings of fatigue, weakness, or exhaustion after a stroke, occurring mentally, physically, or both, which is not alleviated by general rest [70,330].

Post-ischemic stroke neuroinflammation, distinct from systemic inflammation, involves active resolution processes mediated by arachidonic acid derivatives [331] and has crucial impacts on neural stem cells for effective brain repair [332]. Acute inflammation is a protective reaction by the immune system in response to tissue damage or invading pathogens. When the acute inflammatory response is not resolved, it can contribute to organ pathology and amplify many widely occurring chronic inflammatory clinical phenotypes, including neurodegenerative diseases, metabolic syndrome, asthma, allergy, diabetes, inflammatory processes of aging, arthritis, cancers, organ fibrosis, and cardiovascular and periodontal diseases [331].

Neuroinflammation plays a role in both secondary brain injury and neurorepair post-stroke, affecting neurogenesis [333]. Inflammatory impacts on neurogenesis are complex, with positive and negative effects, influencing CNS injury and repair [334]. Acute neuroinflammation promotes neurogenesis and neuronal survival [335,336].

The recovery from a stroke is mediated primarily by angiogenesis and neurogenesis at the tissue and cellular levels and can be targeted for therapeutic intervention [12,337]. Utilizing neurogenesis as a therapeutic target for self-repair and recovery from stroke damage is challenging due to complex interactions with neuroinflammation and angiogenesis, particularly the time-dependent effects of immune cells, cytokines, and chemokines in the post-stroke environment [338].

Cell death induced by ischemic stroke is characterized by the dysregulation of metabolic processes and activating glutamate receptors, triggering an increase in intracellular calcium concentration, and ultimately leading to cellular death [339]. Neurogenesis, or the birth of new neurons, is a key process in post-stroke recovery and repair in the infarct and surrounding areas of the damaged brain region [340,341]. This involves migrating neural stem cells originating from the sub-ventricular zone (SVZ) and the sub-granular zone (SGZ) of the dentate gyrus [34,342,343] to the infarct and peri-infarct region, followed by their differentiation into functional neurons [344,345,346], as shown in Figure 15 [9,12].

Cerebral ischemia-induced neuroinflammation causing brain damage has dual beneficial and harmful effects, depending on the spatial-temporal context [12]. Microglia activation upon the onset of an ischemic attack contributes to ischemia–reperfusion injury by releasing both pro- (at the M1 physiological state) and anti-inflammatory (M2 state) factors. Pro-inflammatory cytokines, IL-1β, TNFα, and IL-6, contribute to tissue repair by enhancing phagocytic activity, excessive release leading to the upregulation of inducible nitric oxide synthase (iNOS), damaging neurons, and disrupting the blood–brain barrier—the anti-inflammatory response results in neuroprotection by tissue repair, regeneration, and promoting angiogenesis (Figure 15).

Extracellular communications between various cells and tissues within the body are essential in cross-talks for angiogenesis and neurogenesis [347]. Some signaling molecules, e.g., vascular endothelial growth factor (VEGF), bind to selected receptors on endothelial cells, promoting the formation and survival of new blood vessels. Endothelial cells activated in the ischemic infarct region secrete VEGF, triggering neurogenesis [348]. Subsequently, these newly generated neural stem cells migrate to and become integrated into the infarcted and peri-infarct areas, where they later mature into functional neurons [349]. Neurogenesis and angiogenesis are highly coordinated and mutually supportive processes that enhance brain repair after ischemic stroke injury [12,347].

Neurogenesis is considered to occur in two so-called neurogenic areas of the brain: the subgranular zone (SGZ) of the hippocampal dentate gyrus (DG) and the subventricular zone (SVZ) of the lateral ventricles [350]. The hippocampus is a major brain region involved in memory, emotional processing, and vulnerability to stress and is one of the most severely affected areas [351,352]. Neurogenesis consists of five sequential stages of adult hippocampal neurodevelopment, including (1) proliferation of primary neural stem/progenitor neural stem cells giving rise to transient amplifying cells, (2) migration, (3) morphogenesis, (4) differentiation, and (5) synaptic integration, which define the progress of newly born neurons from the adult neural stem cell to functional integration within the hippocampal circuitry [14]. The expression of stage-specific biochemical mediators that identify newly born neurons on each step of neurodevelopment can be an essential target for pharmacological correction in the therapy of stroke and post-stroke brain fatigue as follows:Proliferation: IGF-1, Shh, BMP-Noggin, FGF2, VEGF, CNTF, LIF, and Wnt;Differentiation: FGF2, CNTF, and Wnt;Migration: DISC1 and NDEL1;Morphogenesis: IGF1, DISC1, NDEL1, FEZ1, GABA, CREB, KIAA1212, and PTEN/AKT;Synaptic integration: BDNF, DISC1, FEZ1, Glutamate, and CREB.

Several neurotrophic factors (IGF1, BDNF, and Wnt3) exhibit the canonical role in multiple neurogenic processes as follows:Insulin-like growth factor 1 (IGF1) acts through at least two pathways to stimulate proliferation, neuronal differentiation, morphogenic development, and survival of newly born cells. Dose-dependent proliferative effects of IGF1 on cultured hippocampal neural progenitors require activation of the mitogen-activated protein kinase (MAPK) pathway or G-protein activation through nuclear translocation of phosphorylated extracellular signal-regulated kinase (ERK)-1. Downstream signaling through the phosphoinositide 3-kinase (PI3K)-AKT (also known as protein kinase B) pathway, IGF1 signaling mediates an increase in survival of newly born neurons via the promotion of anabolic activity and glucose utilization, and the inhibition of glycogen synthase kinase 3b (GSK3b).In the hippocampus, brain-derived neurotrophic factor (BDNF) activates synaptic plasticity and related processes, including learning and memory. BDNF signaling facilitates the proliferation and stimulation of dendritic development through its cognate receptor TrkB, which activates the MAPK/ERK and the PI3K pathways. Glutamatergic effects on neurogenesis depend critically on the expression of receptor subtypes during the various stages of neurogenesis and play a critical role in cell survival and synaptic development, which could be partly mediated through a glutamatergic-mediated enhancement of BDNF.Wingless-type MMTV integration site-3 (Wnt3) is expressed in the hippocampus and regulates proliferation and neuronal differentiation. The canonical Wnt pathway involves inhibition of GSK3b, though noncanonical signaling pathways directly activate Jun-N-terminal kinase (JNK), or protein kinase C (PKC) and calcium-calmodulin-dependent protein kinase (CaMK) II, via a rise in intracellular calcium.The transcription factor, cAMP response element-binding protein (CREB), is a downstream effector of depolarizing a classical inhibitory neurotransmitter γ-aminobutyric acid (GABA) that depolarizes newly born neurons, which is opposed to its hyperpolarizing role in mature neurons [14].

The local environment highly influences developing neurons, and many exogenous factors influence the rate of neurogenesis and the survival and integration of newly born neurons. Exercise, sleep, sex, environmental enrichment, and learning are all conducive to neurogenesis. On the contrary, stress, drug abuse, aging, and depression decrease neurogenesis in the adult brain [14].

In addition, age remains a prominent factor affecting neurogenesis. The rate of neurogenesis steadily declines with age, with stroke increasing the sharpness of that decline [9,340]. An age-associated loss in neural stem cell number and/or activity could cause this decline in brain function, so interventions that reverse aging in stem cells might increase the human cognitive health span [16]. Dysregulated neurogenesis contributes to neurodegenerative diseases, including Alzheimer’s disease (AD), Parkinson’s disease (PD), Huntington’s disease (HD), and others [353].

The functional relevance of newborn hippocampal neurons has been implicated in many processes, such as resilience to and remission from stress, pattern separation, memory formation, and learning, as well as in neurological disorders such as AD and PD [16,354,355]. Stress and exposure to stress hormones (glucocorticoids) decrease the generation of hippocampal neurons and increase cell death in rodents [356]. In animal studies, stress negatively affects hippocampal neurogenesis [357], learning, and memory [358,359], increasing cognitive impairments [360,361]. Stress attenuates the inhibitory hippocampal regulation of the HPA axis [362,363], and neurogenesis is required to maintain this regulation [364,365], which is associated with stress reactivity and mood disorders such as major depressive disorder.

### 5.2. Botanicals for Promoting Neurogenesis and Angiogenesis and Recovery of Ischemic Stroke

Long-lasting post-stroke repair mechanisms involving neurogenesis, angiogenesis, and synaptic plasticity initiate within several days and continue for weeks and months [366,367]. Since the formation of new neurons is a complex cascade of multistep processes (described above in Section 5.1), any multitarget pharmaceutical intervention that could augment this process would improve stroke recovery. In this context, two promising therapeutic approaches are promising.

One of them is the implementation of complex hybrid botanical preparations (BHP) used in traditional Chinese medicine (TCM) for stroke and adaptogens. Adaptogens exhibit multi-target action and the ability to modulate many molecular targets simultaneously, which is desirable in the complex pathophysiology of stroke, wherein multiple cell death pathways are activated simultaneously.

This review focuses on botanicals used in TCM, specifically the adaptogens *Rhodiola rosea* L. and *Withania somnifera* (L.) Dunal.

#### 5.2.1. Adaptogenic Botanicals Used in TCM for the Treatment of Stroke

A comprehensive literature review based on data from seven electronic databases was conducted to assess the efficacy and safety of traditional herbal medicine (THM) for ischemic stroke and to explore experimental studies regarding the potential mechanisms of their action [71]. THMs were prescribed for ischemic stroke as adjuvant therapy with conventional Western medicine (WM) and a sole intervention for over 3–12 months of daily administration within 72 h of stroke onset. Compared with active control of WM, THM combined with routine WM significantly improves neurological function defect scores, promotes clinical total effective rate, and accelerates stroke recovery time with fewer adverse effects. These effects can be qualified to many mechanisms, mainly anti-inflammation, antioxidative stress, anti-apoptosis, brain blood barrier modulation, inhibition of platelet activation and thrombus formation, and promotion of neurogenesis and angiogenesis, targeting multiple intracellular mediators and signaling pathways, including BDNF/PI3K/Akt/mTOR/CREB/GSK3/NF-κB/Nrf2/NOS, where the central role is phosphoryl inositol 3 kinase (PI3K)/protein kinase B (AKT) (Figure 8). In conclusion, the authors suggested that THM can effectively treat stroke [71]. A systematic review included 25 meta-analyses and 21 randomized clinical trials of 33 THMs assessing the total effective rate, death, or dependency, and the secondary outcome measures mainly included the neurological deficit index, mental state evaluation index, cerebral hemodynamic index, AE indicators, and other indicators. As revealed by a traditional and network meta-analysis of 28 clinical trials involving 2780 patients, TCPMs (including complex botanical hybrid preparations (BHP)) significantly improved neurological function defect scores, the Barthel index (BI), and the Fugl-Meyer assessment (FMA) scores for stroke recovery compared with placebo [368]. Another systematic review and network meta-analysis of 64 studies involving 6225 participants reported that combined therapy with different CHIs and conventional WM had the highest probability of being the best treatment regimen [369].

THMs in these studies include mainly complex botanical hybrid preparations consisting of several up to 18 ingredients, several mono-drug botanicals including Renshen (Ginseng Radix et Rhizoma), Danshen (Salvia miltiorrhizae Radix et Rhizoma, in 15,570 patients), Rhubarb root, Ginkgo leaf extracts, and purified compounds such as salvianolic acid, polyphenol puerarin, and tetramethylpyrazine.

For instance, in 10 randomized clinical trials, the active intervention was the complex BHP MLC601 injection (NeuroAiD, also known as the Danqipiantan capsule in China), comprising a fixed combination of 14 extracts: Huangqi (Radix Astragali), Danshen (Radix Salviae Miltiorrhizae), Chishao (Radix Paeoniae Rubra), Chuanxiong (Chuanxiong Rhizoma), Danggui (Radix Angelicae Sinensis), Honghua (Carthami Flos), Taoren (Persicae Semen), Yuanzhi (Polygalae Radix), Shichangpu (Acori Tatarinowii Rhizoma), Shuizhi (Hirudo), Tubiechong (Eupolyphaga Steleophaga), Niuhuang (Bovis Calculus), Quanxie (Cicadae Periostracum), Lingyangjiao (Saigae Tataricae Cornu).

Xuesaitong injection is a standardized Chinese materia medica product extracted from the root of Panax notoginseng (Burk.) F. H. Chen. P. notoginseng total saponins.

Xingnaojing includes Shexiang (Moschus), Zhizi (Gardeniae Fructus), Yujin (Curcumae Radix), and Bingpian (Borneolum Synthcticum).

The Naoxintong capsule is derived from a classic formula comprising 16 natural herbal materials with over 200 identified bioactive compounds.

The Buyang Huanwu decoction has been a well-known traditional Chinese herbal prescription for treating stroke-induced disability for over 200 years [370]. It comprises seven Chinese herbs: Astragali Radix, Angelicae sinensis Radix, Paeoniae Radix Rubra, Chuanxiong Rhizoma, Persicae Semen, Carthami Flos, and Pheretima. The active ingredients of the Buyang Huanwu decoction include calycosin-7-O-β-D-glucoside, 6-hydroxykaempferol-di-O-glucoside, and galloyl-paeoniflorin.

Seventy-seven in vitro and in vivo animal studies of THMs were conducted to assess their anti-inflammatory, antioxidative stress, anti-apoptosis, brain blood barrier modulation, inhibition of platelet activation and thrombus formation, and promotion of neurogenesis and angiogenesis properties in stroke models. These studies suggest the central role of the BDNF/PI3K/Akt/mTOR/CREB/GSK3/NF-kB/Nrf2/NOS-mediated signaling pathway is the potential key mechanism of TCMs for stroke [71,251,369,371,372,373].

Activating the intracellular PI3K/AKT-mediated signaling pathway triggers a cascade reaction downstream protein that mediates multiple cellular functions, including neurogenesis, angiogenesis, neuroinflammatory response, and other repair mechanisms in stroke (see Figure 8).

The mechanism of 57 purified compounds was reported in vitro and in vivo studies. The following results were shown:Salidroside from Rhodiola crenulata promotes neurogenesis targeting FGF2-mediated cAMP/PKA/CREB ↑ [374];Ginsenoside Rd from *Panax ginseng* promotes neurogenesis targeting VEGF, BDNF, PI3K/Akt, ERK1/2 ↑ [373];Salvianolic acid A from *Salvia miltiorrhizae* promotes neurogenesis targeting NF-κB, GSK3/Cdk5↓, β-catenin/DCX, and Bcl-2 ↑ [251];Tanshinone IIA from *Salvia miltiorrhizae* promotes axonal regeneration targeting Nogo-A/NgR1/RhoA/ROCKII/MLC ↓ [375];Astragaloside IV from Astragali Radix promotes neurogenesis targeting BNDF/tropomycin receptor kinase B ↑ [376].

Salvianolic acid A alleviates ischemic brain injury by inhibiting inflammation and apoptosis and promoting neurogenesis in mice [251]. Ginsenoside Rd via the PI3K/AKT/GSK-3β axis could decrease the phosphorylation of tau protein after cerebral ischemia. Ginsenoside Rd substance could also improve neurogenesis after cerebral ischemia through the PI3K/AKT pathway [372,373]. Noteworthy is that all these plants exhibit the potential for effective adaptogenic remedies (Table 2).

#### 5.2.2. Rhodiola and Withania for Neuroprotection in Stroke and Their Synergistic Effect on Neurogenesis

##### Rhodiola in Experimental Models of Stroke

A systematic review and meta-analysis of 15 preclinical studies of *Rhodiola rosea* extract and isolated salidroside and tyrosol suggest they can effectively treat ischemic stroke [377]. They primarily exert potential neuroprotective effects in ischemic stroke through antioxidative, anti-inflammatory, astrogliosis, antiapoptotic, and neuroprotective mechanisms, alleviating the pathological BBB damage mechanisms [377]. The primary measured outcomes included the neural functional deficit score, infarct volume, brain water content, cell viability, apoptotic cells, terminal deoxynucleotidyl transferase (TdT)-mediated dUTP-biotin nick end labeling (TUNEL)-positive cells, B-cell lymphoma-2 (Bcl-2) level, and tumor necrosis factor-α (TNF-α) level. Pooled preclinical data showed that compared with the controls, Rhodiola preparations could significantly improve functional deficit score, modified neurological severity score, rotarod tests, infarct volume, and brain edema. It also can increase cell viability and Bcl-2 levels and reduce TNF-α levels, TUNEL-positive cells, and apoptotic cells [377].

The possible neuroprotective mechanisms of *Rhodiola rosea* preparations and salidroside for ischemic stroke are associated with the following:Alleviating the pathological BBB damage [378];Inhibition of lipid peroxidation [379];Antioxidant effect by increasing the activity of SOD, GSH-Px HO-1, Nrf2, and GST and decreasing the concentration of MDA and ROS [368,380];Inhibition of inflammation by decreasing the expression of pro-inflammatory cytokines such as TNF-α, IL-1β, IL-1, IL-2, and IL-6 [374,381,382,383];Antiapoptotic effects by increasing the levels of Bcl-2 [374,377,380,381,384];Decreasing the levels of Bax [374,380], caspase 3 [374], C-Fos [381], GFAP [382], p53 [385];Decreasing the activity of LDH [384] and reducing TUNEL positive cells [382,384];Neuroprotective effect via regulating BDNK-mediated PI3K/Akt pathway [378,384] and through modulating monoamine metabolism [386];Inhibiting reactive astrogliosis and glial scar formation, probably through the Akt/GSK-3β pathway [377];Direct shock cognate (HSC70) activation in BDNF signaling and neurogenesis after cerebral ischemia [387].

Noteworthy, a decrease in the infarct size effect of *Rhodiola rosea* was predicted in a gene expression profiling study using the T98G human neuroglia cell line after treatment with the *Rhodiola rosea* SHR-5 extract and several of its constituents, salidroside, triandrin, and tyrosol [62]. An interactive pathway analysis of the downstream effects was conducted using datasets containing significantly up- and downregulated genes, where the effects on cellular functions and diseases were predicted. Seven of nine deregulated genes exhibit a change in expression direction consistent with decreases in the infarct size (overlap *p*-value 1.72 × 10^−3^, activation z-score −2.157). The prediction was based on the change in the expression direction of genes CCR2 ↓, CD40LG ↓, CNR1 ↓, FGB ↓, GP6 ↓, HGF ↓, HLA-B ↑, NOX4 ↓, and PPP1R1A ↑ [62].

*Rhodiola rosea* extract has neuroprotective effects against L-glutamate-induced neurotoxicity in cortical neuronal cells. The treatment with *Rhodiola rosea* extract and rosin suppressed the L-glutamate-induced neurotoxicity but not by rosin. Rosin and salidroside attenuate the glutamate-induced increase in phosphorylated MAPK, pJNK, and p38 [388]. The pharmacological profile of rosavin, the major phenylpropanoid of *Rhodiola rosea*, was recently assessed for antioxidant, lipid-lowering, analgesic, antiradiation, antitumor, and immunomodulation activities in vitro and in vivo experiments in animals. Rosavin exhibited significant therapeutic efficacy in neurological, digestive, respiratory, and bone-related disorders [389].

*Rhodiola rosea* extracts and salidroside promote neurogenesis [374,387,389,390,391,392]. Salidroside, a principal bioactive component of the Rhodiola genus, is neuroprotective across a wide time window in vitro and in vivo stroke models due to its neuroprotection properties by significantly reducing infarct size, inhibiting cerebral edema, and improving neurological function [392]. The underlying mechanisms involve antioxidation, anti-inflammation, and anti-apoptosis by regulating multiple signaling pathways and key molecules, particularly NF-κB, TNF-α, and PI3K/Akt-mediated signaling pathways [392]. Salidroside is therapeutically effective against learning and memory decay by stimulating CREB-dependent functional neurogenesis in aging [393].

Salidroside protects against Aβ-induced neurotoxicity in four transgenic Drosophila AD models, increasing longevity and locomotor activity in salidroside-fed Drosophila. The neuroprotective effect of salidroside was associated with upregulated phosphatidylinositide 3-kinase (PI3K)/Akt signaling. Sal also decreased Aβ levels and Aβ deposition in the brain and ameliorated toxicity in Aβ-treated primary neuronal culture [204].

Several preclinical studies suggest that salidroside could effectively prevent neuronal injury after cerebral ischemia. In animal models of ischemic stroke, salidroside significantly ameliorated brain injury by reducing cerebral infarction, preventing cerebral edema, and improving neurological function [381,382,394]. Preventive effects of salidroside on memory and learning deficits caused by cerebral hypoxia or hypoperfusion were also found in some studies [395]. Salidroside exhibited neuroprotective actions against various cell injuries, such as apoptotic neuron death in vitro cell models [380,381,387]. More importantly, it has demonstrated excellent anti-stroke effects of salidroside across a broad therapeutic time window (>48 h) [387]. Nrf2 activated by Sal bind and upregulated the antioxidant response element such as heme oxygenase-1 (HO-1), SOD, and GSHPx [368].

Salidroside significantly reduced the level of these pro-inflammatory cytokines and chemokines, such as tumor necrosis factor-α (TNF-α), interleukine-2 (IL-2), interleukine-6 (IL-6), interleukine-8 (IL-8), interleukine-1β (IL-1β), MCP-1, and MIP1α, in the tissue or serum of the experimental model of IS [368,381,382,394,396].

A recent study reported that salidroside exerted protection against cerebral ischemia in MCAO mice, remarkably inhibiting the release of microglial-derived inflammatory factors and enhancing microglial phagocytosis through promoting microglia from M1 phenotype to M2 phenotype [382].

Fibroblast growth factor-2 (FGF2), which is involved in the cyclic adenosine monophosphate (cAMP)/protein kinase A (PKA)/CAMP response element (CRE)-binding protein (CREB) pathway, has been shown to facilitate dendritic and synaptic plasticity. Salidroside promotes dendritic and synaptic plasticity in the ischemic penumbra, significantly inhibiting inflammation and apoptosis and promoting dendritic and synaptic plasticity via the FGF2-mediated cAMP/PKA/CREB pathway in isolated PC12 cells under oxygen-glucose deprivation/reoxygenation conditions and in vivo in rats with middle cerebral artery occlusion/reperfusion [374]. Overall, that study suggests that salidroside is an effective treatment for ischemic stroke that functions via the FGF2-mediated cAMP/PKA/CREB pathway to promote dendritic and synaptic plasticity [374].

Salidroside enhances endogenous neural regeneration after cerebral ischemia/reperfusion in rats. It reduces infarct volume, ameliorates ischemia/reperfusion-induced neurobehavioral impairment, and restores neuronal nuclei-positive cell loss after ischemia/reperfusion injury. Salidroside treatment elevates the mRNA expression and protein concentration of BDNF and NGF in the ischemic periphery area. The mechanism of the effect may involve the regulation of BDNF/NGF and the Notch signaling pathway [391].

Salidroside induces neurogenesis after cerebral ischemia in rats of middle cerebral artery occlusion (MCAO). Salidroside dose-dependently decreased cerebral infarct volumes and neurological deficits and improved cognitive performance in beam balance and Morris water maze tests with maximal effects by 50 mg/kg/day. Intraperitoneal administration of Salidroside for 7 days significantly increased BrdU+/nestin+, BrdU+/DCX+, BrdU+/NeuN+, BrdU−/NeuN+, and BDNF+ cells in the peri-infarct cortex. It dose-dependently increases heat shock cognate HSC70 ATPase and HSC70-dependent luciferase activities, but it did not activate heat shock protein HSP70.

Salidroside also increased the ischemic brain’s BDNF protein and p-TrkB/TrkB ratio. Additionally, ANA-12 blocked salidroside-dependent neurogenesis and increased BrdU-/NeuN+ cells in the peri-infarct cortex. It was concluded that salidroside directly activates HSC70, thereby stimulating neurogenesis and neuroprotection via BDNF/TrkB signaling after MCAO. Salidroside and similar activators of HSC70 might provide clinical therapies for ischemic stroke [387].

Salidroside is also involved in inhibiting apoptosis of endothelial cells by specifically activating PI3K/Akt signaling via phosphorylating Akt on Ser473. This results in a reduction in the protein expression ratio of Bax/Bcl-2, the activation of caspase-3, and the decrease in the release of cytochrome c in human brain microvascular endothelial cells (HBMECs) and brain microvessels [378].

##### Withania in Experimental Models of Stroke

*Withania somnifera* (WS), also commonly known as Ashwagandha or Winter cherry, has been used for centuries to treat ailments in the Ayurvedic as well as indigenous systems of medicine as an aphrodisiac, nerve tonic, anti-inflammatory, and anti-cancer agent, and for other a plethora of human medical conditions [242,243,244,397,398,399,400,401,402]. Many studies have documented the neuroprotective activities of WS. Reports obtained from preclinical research and clinical trials have substantiated the neuroprotective role of WS. It was found to be active against many neurological and psychological conditions like Parkinson’s disease, Alzheimer’s disease, Huntington’s disease, ischemic stroke, sleep deprivation, amyotrophic lateral sclerosis, attention deficit hyperactivity disorder, bipolar disorder, anxiety, depression, schizophrenia, and obsessive–compulsive disorder [243].

In early studies, the neuroprotective activity of an aqueous extract of WS was tested in both pre- and post-stroke treatment regimens in a mouse model of permanent distal middle cerebral artery occlusion, MCAO [136,403,404,405,406,407,408,409,410]. In the dose of 200 mg/kg, WS extract improved functional recovery and significantly reduced the infarct volume in mice when compared to those treated with a vehicle in both treatment regimens. WS upregulated the expression of heme oxygenase 1 (HO1) and attenuated the expression of the proapoptotic protein poly (ADP-ribose) polymerase-1 (PARP1) via the PARP1-AIF signaling pathway, thus preventing the nuclear translocation of apoptosis-inducing factor and subsequent apoptosis. Semaphorin-3A (Sema3A) expression was reduced in the WS-treated group, whereas Wnt, pGSK3β, and pCRMP2 expression levels were virtually unaltered. The authors suggest that the interplay of antioxidant-antiapoptotic pathways and the possible involvement of angiogenesis in the protective mechanism of WS could be a potential prophylactic and therapeutic option in aiding stroke repair [409]. The dose of 300 mg/kg body weight and pre-supplementation with WS for 30 days to MCAO animals effectively restored the acetylcholinesterase activity, lipid peroxidation, and thiols and attenuated MCAO-induced behavioral deficits. WS significantly reduced the cerebral infarct volume and ameliorated histopathological alterations. Improved blood flow was observed in the single-photon emission computerized tomography images from the brain regions of ischemic rats pre-treated with WS. The results of the study showed a protective effect of WS supplementation in ischemic stroke and possibly in stroke management [407]. WS pre-supplementation also ameliorated MCAO-induced oxidative stress, mitochondrial dysfunctions, apoptosis, and cognitive impairments [410].

Withaferin A (WA), a steroidal lactone obtained from *Withania somnifera* extract, inhibits neuro-apoptosis, modulates vascular smooth muscle cell (VSMC) migration, and activates PI3K/Akt signaling. WA treatment (in doses of 25, 50, and 100 mg/kg body weight) significantly reduced the infarct area in a carotid ligation model; WA reduced intimal hyperplasia and proliferating cell nuclear antigen (PCNA)-positive cell counts and suppressed PI3K/Akt signaling following cerebral ischemia/reperfusion injury. WA supplementation was found to downregulate apoptotic pathway proteins. WA suppressed PTEN and enhanced p-Akt and GSK-3b levels and elevated mTORc1, cyclinD1, and NF-κB p65 expression, suggesting activation of the PI3K/Akt pathway. In vitro studies, WA exposure severely downregulated matrix metalloproteinases (MMP)-2 and -9 and inhibited migration of A7r5 cells. Additionally, WA reduced the proliferation of A7r5 cells significantly. Overall, WA exhibits neuroprotective effects by activating the PI3K/Akt pathway, modulating the expression of MMPs, and inhibiting the migration of VSMCs [136].

##### Botanical Hybrid Preparation of Rhodiola and Withania for Promotion of Neurogenesis

Considering botanicals comprise multicomponent active compounds, their interactions result in novel, unexpected pharmacological activity due to their synergistic and antagonistic effects.

Our review identified several new molecular targets related to brain fatigue and neurogenesis and a pharmaceutical intervention that may ameliorate brain fatigue—the BHP of *Withania somnifera* root extract (KSM-66^®^ standardized to contain ≥5% withanolides) and *Rhodiola rosea* rhizome root extract (EPR-7™ standardized to contain ≥3% rosavins and ≥1% salidroside), commercially available dietary supplements Adaptra^®^. Adaptra^®^ is declared as an adaptogen “to help mind & body thrive and prevent that “depleted” feeling you can get from a busy lifestyle and to manage stress, feel in control, maximize energy and stamina, support vital adrenal function, sharpen focus and concentration” (https://www.europharmausa.com/adaptra, accessed on 10 February 2025).

Both two ingredients, separately and in their fixed combination Adaptra^®^ were studied in T98G neuroglia cell culture in vitro conducting RNA sequencing to profile gene expression alterations and analyze the relevance of deregulated genes to adaptive stress-response signaling pathways using in silico pathway analysis software [67].

The choice of T98G neuroglia cells was justified in previous publications [61,411,412,413]. Glia, comprising approximately 90% of the CNS, is a transportation link between the bloodstream and neurons. It contributes to the neuroprotection of the brain through the expression of the innate immune response, promoting the clearance of neurotoxic proteins and apoptotic cells from the CNS as well as regulating the entry of inflammatory systemic cells into the brain at the blood–brain barrier, reducing axonal loss and gliosis [411,414]. This activates both tissue repair and the rapid restoration of tissue homeostasis. Glial cells express hormonal receptors, which play key roles in stress-induced disorders. The physiological function of neuroglial cells includes the uptake of neurotransmitters, biosynthesis, and release of neurotrophic factors, immune regulation, modulation of synaptic activity, metabolic supply of energy, and other substances [415,416], maintaining brain homeostasis—a function supposed to be characteristic for adaptogens by definition.

Isolated brain cells were incubated independently with Adaptra^®^ or Rhodiola and Withania. Gene expression degree was measured and expressed in fold changes compared to control and analyzed using Ingenuity Pathway Analysis (IPA) software (QIAGEN Bioinformatics). IPA performs different calculations based on the Ingenuity Knowledge Base, a large gathering of observations with approximately 5 million findings manually curated from the biomedical literature or integrated from third-party databases [67].

Adaptra^®^ deregulated 22 genes involved in activating neuronal development (Figure 16). The predicted increase in neuronal development in response to Adaptra^®^ intervention was based on fold changes in gene expression consistent with literature findings; 22 of 57 genes had the same direction (up- or down-deregulation) of gene expression (Table 6 and Figure 16).

Notably, 9 of 22 genes activating the development of neurons were independently unaffected by Rhodiola and Withania. It means that these two ingredients of BHP Adaptra^®^ act synergistically in combination. In other words, BHP Adaptra^®^ is superior to Rhodiola or Withania in activating neurogenesis and has the potential effects mentioned above.

The roles, functions, and consequences of the downregulation of expressions of the PRKCZ gene encoding the protein kinases PKCζ and PKMζ, which regulate glucose transport and long-term memories, and the GRIN3A gene encoding the glutamate ionotropic receptor gene subunit Grin3a of the glutamate ionotropic receptor NMDAR with central roles in brain plasticity, development, learning, and memory in the brain are summarized in Table 7 below. Both genes are critical in maintaining brain health, plasticity, and systemic glucose homeostasis. Their downregulation can contribute to neurological diseases and metabolic dysfunctions, but such changes could confer adaptive benefits in specific pathological contexts. Depending on the context, the downregulation of the PRKCZ and GRIN3A genes could have negative and positive consequences, including health conditions, brain function, and adaptation to specific environments.

Downregulation of *PRKCZ* and *GRIN3A* can have both positive and negative consequences in long-lasting post-stroke brain fatigue. Post-stroke brain fatigue involves a complex interplay of neuroinflammation, energy deficits, disrupted neuroplasticity, and impaired cognitive recovery, all of which may intersect with the functions of these genes. The predicted beneficial effects of Adaptra^®^ due to the downregulation of *PRKCZ* include lowering excitotoxic risk and energy conservation by reducing the overactivation of memory pathways. Downregulation of *GRIN3* suggests enhanced LTP for recovery and improved synaptic transmission efficiency.

The upregulation of *ADGRL1*, *CDK5R1/CDKL3*, *CHRNB2*, and *ROR2* gene expression in the context of long-lasting post-stroke brain fatigue can have a mix of positive and negative effects, depending on how these proteins influence brain function, recovery, and energy regulation. Table 7 shows a summary of predicted beneficial impacts on post-stroke brain fatigue.

Upregulation of *ADGRL1*, *CDK5R1/CDKL3*, *CHRNB2*, and *ROR2* genes in post-stroke long-lasting brain fatigue generally supports neuroplasticity, cognitive recovery, and neurogenesis, which are beneficial for rehabilitation. However, the energy costs and risks of maladaptive signaling or neurodegeneration could exacerbate fatigue if the brain’s metabolic and vascular recovery is insufficient. Balancing these pathways is key to optimizing recovery while minimizing harm.

Several findings highlight the essential roles of Adhesion GPCR Nanoclusters (ADGRL1/latrophilin-1) in neurogenesis, synaptic function, and neural development [418,419,420,421].

A recent study highlights the importance of latrophilin-1 in directing inhibitory synaptic connections to neuronal cell bodies, emphasizing its role in synaptic organization. It also indicates that human loss-of-function mutations in latrophilin-1 (ADGRL1) cause significant neurodevelopmental impairments, underscoring its critical role in neural development [418]. C-terminal phosphorylation of latrophilin-1/ADGRL1 affects the receptor’s function and its role in neurogenesis and synaptic activity [419]. That is related to the energy balance since latrophilin-1 (ADGRL1/LPHN1) controls energy balance and food intake, suggesting broader physiological roles beyond the nervous system [420].

Latrophilin-1 may function as a glucose receptor, indicating a potential link between metabolic processes and neural functions. This also suggests that latrophilin-1 is a glucose receptor that mediates energy and glucose homeostasis [421].

These studies give insights into the multifaceted roles of ADGRL1 (latrophilin-1) in neural development, synaptic function, neurogenesis, brain energy, and glucose homeostasis. Furthermore, these findings align with the glutamate neurotransmission imbalance, glucose depletion, and energy shortage hypothesis [149], Section 3.3.1.

The regulation of *NTF4*, *NEFH*, *PTPRD*, *ADGRF1*, *CHRNB2*, *GHSR*, *ITGB2*, *LRRK2*, *MAGI2*, *MBP*, *mir-10*, *MYH7B*, and *PPP1R9A* genes in the post-stroke brain can have complex effects. Due to their roles in neural repair, neuroinflammation, and neuronal function, these genes can potentially contribute to long-lasting mental or brain fatigue.

The interplay of these genes highlights the complexity of post-stroke recovery. While some upregulated genes may support repair, others may exacerbate inflammation or dysregulate neural pathways. Similarly, downregulated genes critical for neuronal survival and function could hinder recovery, collectively contributing to long-lasting mental or brain fatigue. The integrated effect of Adaptra^®^ on 72 deregulated genes predicts an increase in neurogenesis, suggesting a beneficial recovery in post-stroke brain fatigue (Table 6 and Table 7 and Figure 16).

## 6. Discussion

### 6.1. The Review’s Highlights

The term “brain fatigue”, introduced in medicine by [5,6,7], refers to long-lasting mental fatigue after stroke, traumatic or infectious brain injury, or aging low-grade neuroinflammation of the central nervous system.

Brain fatigue occurs when the brain becomes overworked, tired, and damaged. Rest and sleep help but do not cure it. There is no cure for brain fatigue; however, it seems to be alleviated. One approach to alleviating brain fatigue is possibly using adaptogens promoting neurogenesis in post-stroke rehabilitation and preventing stress-induced brain fatigue and other disorders.

The anti-neuroinflammatory and neuroprotective effects of adaptogens in neurodegenerative diseases were extensively studied to prevent, cure, and promote recovery from inflammatory disorders, particularly in low-grade chronic neuroinflammatory disorders/diseases, including senile dementia, depression, anxiety, ischemic stroke, and viral infections.

Recovery is a long-lasting process that requires restoring homeostasis, neurogenesis, angiogenesis, metabolic balance (optimal metabolic index, testosterone/cortisol ratio), SAS/HPA misbalance, and stress/distress disruption.

The biochemical and physiological bases of brain or mental fatigue are not sufficiently understood, and pharmacological perspectives of therapy are a challenge. There is no robust evidence-based intervention to improve post-stroke fatigue. However, some evidence suggests that pharmaceutical interventions may become promising treatments.

Some botanicals, particularly adaptogens, including *Rhodiola rosea* L. (Arctic Root) and *Withania somnifera* Burge roots (Ashwagandha) preparations, improve cognitive functions and prevent fatigue in human subjects under stressful conditions.

Meanwhile, multi-target network pharmacology systems biology concepts and innovative drug discovery tools, such as transcriptome-wide microarray gene expression profiling in vitro testing and artificial intelligence in silico, provided theoretical models that propose using botanical adaptogens to treat stress-induced disorders, including brain fatigue and related neurological diseases.

Recent studies of gene expression in neuroglia and neuronal cells unveiled the polyvalent and synergistic actions of botanical hybrid preparations (BHP) in stress-induced disorders, suggesting that adaptogens exhibited multitarget and pleiotropic effects via HPA- and GPCR-mediated signaling pathways.

Specifically, the BHP of Arctic Root and Ashwagandha, Adaptra^®^, shows a potential activation of neuronal development by specifically deregulating 22 genes in neuroglia cell culture, including nine product-specific synergistically deregulated genes: *PRKCZ* protein kinase C, *GRIN3A* glutamate ionotropic receptor, *ADGRL1* adhesion G protein-coupled receptor, *CDK5R1* and *CDKL3* cyclin-dependent kinases, *CHRNB2* cholinergic receptor, and *ROR2* receptor tyrosine kinase-like orphan receptor, suggesting beneficial effects in neurogenesis, neuroplasticity, and cognitive recovery, which are beneficial for rehabilitation in post-stroke long-lasting brain fatigue. Remarkably, Adaptra^®^ downregulates the expression of the *GRIN3A* gene encoding subunit Grin3a of glutamate ionotropic receptor *NMDAR* with central roles in brain plasticity, development, learning, and memory in the brain. That aligns with Rönnbäck’s and Johansson’s hypotheses of the glutamate-glutamine cycle imbalance and glucose-mediated ATP energy supply in brain fatigue.

Some other BHPs used in traditional Chinese medicines can significantly improve neurological function defect scores, promote clinical total effective rates, and accelerate the recovery time of strokes due to multiple mechanisms, mainly anti-inflammation, antioxidative stress, anti-apoptosis, brain blood barrier modulation, inhibition of platelet activation and thrombus formation, and promotion of neurogenesis and angiogenesis.

### 6.2. Where Do We Go in Drug Discovery?

Herbs, used as food and a source of drugs for centuries, have been extensively studied chemically and pharmacologically during the last decades, with two main aims. One of them was to identify and purify pharmacologically active compounds following the development of new, more active compounds and analogs targeting a selective receptor free of adverse effects. Another aim was to combine several plants or their purified ingredients into new multitarget hybrid combinations with more effective pharmacological activity.

The first approach was successfully applied in orthodox Western medicine, resulting in the discovery of morphine, salicin, papaverine, paclitaxel, etc., and their synthetic analogs and the current drug development platform, which is based mainly on the pharmacodynamics and pharmacokinetics of botanicals and accrues knowledge of interactions of biologically active molecules of plant extracts with receptor molecules, primarily proteins, involved in cellular and organism responses to achieve a therapeutic effect and maintain organismal homeostasis. In the meantime, the stimulus-response coupling is often mediated by complex intra- and extracellular communications and feedback regulations within networks of signaling pathways and pharmacological systems, including neuroendocrine-immune complex, cardiovascular, gastrointestinal, and other systems.

The second approach was used mainly in traditional Chinese medicine based on empirical knowledge and undergoing modernization by applying the current understanding of pharmacological systems and network pharmacology of metabolomics, proteomics, transcriptomics, genomics, and microbiomes (collectively—OMICS).

In this review, we discussed to some extent the following:The trends and pitfalls in pharmacognosy from galenic formulations to evidence-based medicine and artificial intelligence in network analysis of the systems pharmacology;Multitarget and pleiotropic effects of adaptogens: unveiling of the polyvalent and synergistic actions of botanical hybrid preparations in stress-induced disorders;Effects of botanical hybrid preparations on crosstalk of neuroendocrine-immune, cardiovascular, and other regulatory systems and the integrative response of intracellular and extracellular communications in stroke.

Our knowledge of biochemical mechanisms underlying brain fatigue, discussed in Section 3, is based on the fundamentals of the physiology of organisms and molecular biology, which conventional methods of pharmacology and biology have accrued. Meanwhile, novel molecular biology and computational informatics methods reveal the complexity of physiological processes and the plethora of new mediators of their intra- and inter-cellular communications within neuroendocrine, immune, and cardiovascular systems [66,69,257,258,259,260,422,423,424,425,426,427,428,429,430,431,432,433,434,435,436,437,438]. For instance, inflammation covers several levels of interactions of many mediators of the inflammatory response, systematized in the atlas of inflammation, where various bioactive molecules play a role and can be a pharmacological target for pharmacological intervention [331].

The conventional treatment of cardiovascular diseases, cancer, and other diseases includes concomitant polyvalent application of complex medications targeting multiple primary receptors interacting within networks of neuroendocrine systems and other regulatory systems. As an example, simultaneous treatment of post-stroke disability, including hypotensive, anticoagulants, hypocholesterolemic, anti-atherosclerosis, and anti-inflammatory medications, such as (i) clopidogrel, platelets aggregation inhibitor, which reduces the risk of heart disease and stroke; (ii) lercanidipine, a calcium channel blocker relaxing and opening the blood vessels—a hypotensive drug; (iii) losartan, an angiotensin receptor blocker, to treat hypertension, diabetic nephropathy, and to reduce the risk of stroke; (iv) spironolakton, K+ ion channels activators treatment of high blood pressure, low blood potassium; (v) bisoprolol, cardioprotective selective and competitive betta 1 receptor blockers of adrenalin, second-line agent for hypertension, first-line agent for heart diseases; and (vi) rosuvastin, hypocholesterolemia that affects low-density lipoprotein cholesterol.

This multitarget approach aligns with the traditional use of botanicals and natural complex botanical hybrid combinations, which lack numerous adverse effects of synthetic pharmaceuticals. The matter is the dose, particularly of some neurotoxic botanicals [208] containing highly active alkaloids and other plant secondary metabolites, which developed to defend against pathogens due to adaptability to stressors.

An important implication of these findings is that any particular plant extract/botanical preparation with specific qualitative characteristics, quantitative composition, and the product-specific HPLC fingerprint can exhibit quite different pharmacological activity and pharmaceutical profile (signature) depending on the dose, often exhibiting a dual response in the organism from positive to inactive or even harmful. The dose is essential and has crucial significance [69].

### 6.3. Critical Appraisal, Limitations, and Challenges in Network Pharmacology Studies

The network analysis model often limits itself to intracellular communications and does not include extracellular communications and interactions between various physiological regulatory systems, such as the integrative nervous, endocrine, immune, cardiovascular, respiratory, and renal systems in various organs and tissues. There is no place and role left for physiology and anatomy in this mechanistic model of the integrative regulatory system of the organism to maintain homeostasis in response to pharmaceutical intervention. The main challenge in network pharmacology is the lack of sufficient and evidence-based knowledge in molecular biology, pharmacodynamics, and pharmacokinetics of all compounds comprising botanical preparations and interactions of mediators of their pharmacological activity. Bioinformatics involves building networks and analyzing them mechanistically using available in silico tools. Artificial intelligence is sometimes insufficient without valid experimental and well-designed preclinical and clinical studies of well-characterized botanical preparations, ensuring their well-established quality, efficacy, and safety.

The conclusions of many network analysis studies are going too far and are not supported by actual evidence in many aspects. In these cases, further studies are based on misleading backgrounds and can lead to useless results. Furthermore, referring only to the latest publications, particularly to review articles, and ignoring original studies distorts the scientific history of many inventions. Several limitations of network pharmacology predictions based exclusively on in silico and some in vitro studies exist, which must be further verified in animal experiments.

The second limitation is the limited scientific information on the direction of correlations between gene expression and physiological function or disease used in silico analysis for therapeutic efficacy or toxicity predictions. Additional studies where different experimental outcome measures will be applied are required. Furthermore, a strong resistance mechanism can override all other effects of a drug. For instance, the drug efflux transporter P-glycoprotein in the cell membrane can expel medications before reaching their actual intracellular targets, thereby preventing all or most downstream signaling and network effects. Lastly, clinical studies on predicted diseases in human subjects are essential.

Finally, the compatibility of pharmaceutical-grade herbal extracts of reproducible quality and pharmacological activity compared to purified active constituents is challenging since the content of a genuine extract is adjusted to the defined content or range of the active constituents with known therapeutic efficacy or the content of the analytical markers.

## 7. Conclusions

This narrative review suggests that adaptogens trigger the defense adaptive stress response, leading to the extension of the limits of resilience to overload, which induces brain fatigue and mental disorders. For the first time, the review justifies the neurogenesis potential of adaptogens, particularly BHP of Arctic Root and Ashwagandha, providing a rationale for potential use in individuals experiencing long-lasting brain fatigue. In this review, we hypothesized that adaptogenic BHP of Arctic Root and Ashwagandha might speed up the recovery of patients with long-lasting brain fatigue and be helpful in rehabilitating patients with post-stroke, traumatic, and post-viral brain injuries. It might also be effective in the therapy of ischemic stroke by resolving stroke-induced brain damage and normalizing cognitive functions. The review provided insight into future research on the network pharmacology of adaptogens in preventing and rehabilitating long-lasting brain fatigue following stroke, trauma, and viral infections.

Further research is warranted to obtain clinical evidence of the efficacy and safety of Adaptra^®^ in brain fatigue, including the following:Placebo-controlled, double-blind, randomized clinical studies with different treatment strategies, e.g., pre- and post-distress, low- and high-dose regimens, etc.Brain imaging on humans suffering from brain fatigue following extracellular concentrations of glutamate, ATP, and neurotropic/neuroactive substances (e.g., Adaptra^®^) over time.Animal experiments: the role of the astroglial network and defective glutamate uptake.In vitro experiments: effects of Adaptra^®^ on glial, glial/glial-neuronal signaling, expression of Adhesion GPCR Nanoclusters (ADGRL1/latrophilin-1) in neurogenesis, synaptic function, and neural development, etc.

## Figures and Tables

**Figure 2 pharmaceuticals-18-00261-f002:**
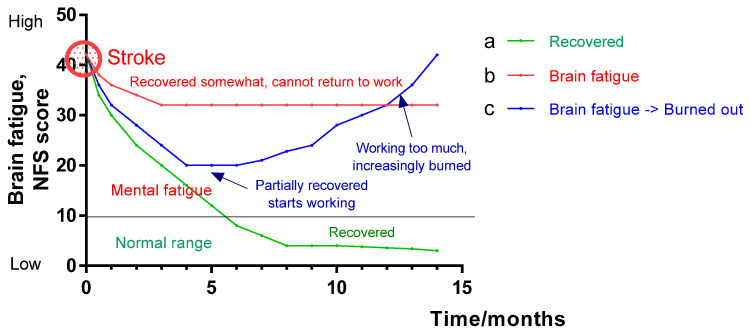
Symbolic representation of three typical scenarios of the degree of brain fatigue over time after a head injury, emotional strain-induced stroke, or viral infection: a—adaptive reversible recovery when the patient returns to work and social life (green line), b—for others, the mental energy does not return as expected, but the person gets on with the job, even if the energy is not quite enough. It works for a while, but after maybe a few months or half a year, the person has run out of energy, and brain fatigue increases (solid blue line); c—irreversible damage if some patients still have severe distressing and long-lasting brain fatigue can make work impossible and daily lifestyle at home be sufficiently exhausting (red line) [5].

**Figure 3 pharmaceuticals-18-00261-f003:**
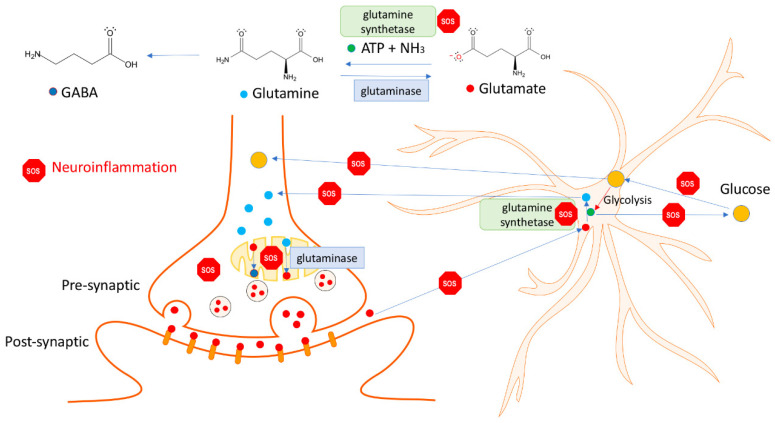
Glutamate–glutamine cycle in nerve cells [156,157,158,159,160]. Neurotransmission between two neurons’ synapses and interactions with a nearby astrocyte, where glutamate is the signal substance between nerve cells. Glutamate is released from the presynaptic terminal, and after exerting its effect on the recipient neuron, the postsynaptic membrane is taken up by the astrocytes’ discharge and converted to glutamine, which is then transported back to the nerve cells to form new glutamate. Glutamate inside the astrocytes also signals that glucose is taken from the blood into the astrocytes and onto the nerve cells as new energy. In distress or brain damage-induced neuroinflammation, see the SOS sign in red; the astrocytes’ capacity to convert glutamate into glutamine is reduced due to a lack of energy supply mediated by reduced ATP. More glutamate accumulates around the nerve cells, and increasing amounts accumulate in the synapse areas, making signaling less specific. If mental activity is high in this situation, there is a risk that nerve cell signaling will fail due to decreased amounts of glucose/energy. The astrocytes take up less glucose, and less energy and glutamate are available in the neurons. Modified from [5,6].

**Figure 4 pharmaceuticals-18-00261-f004:**
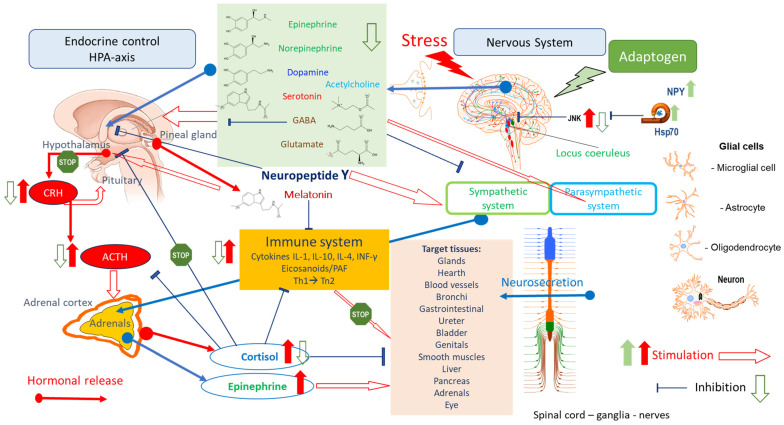
Simplified overview of the stress system (central nervous system, CNS, and peripheral tissues/organs in the periphery) and reciprocal connections of elements of the neuroendocrine-immune complex to mobilize an adaptive response against the stressor. The brain and spinal cord comprise the CNS, the cerebral cortex—glutamatergic pyramidal and GABA-ergic interneurons, and glial cells, including astrocytes, oligodendrocytes, and microglia. The forebrain includes dorsal glutamatergic neurons, ventral GABAergic interneurons, and locus coeruleus (LC) neurons. The peripheral components of the stress system include the hypothalamic–pituitary–adrenal axis (HPA), the autonomic nervous system (ANS) comprising the sympathetic nervous system (SNS) secreting mainly norepinephrine (NE) and acetylcholine (AcCh), and the sympathy–adrenomedullary (SAM) system, and (ii) the parasympathetic nervous system (PNS) secreting AcCh. Two key end hormones, cortisol and epinephrine, regulate metabolism, circulation, and blood homeostasis. The abbreviations of hormones and neurotransmitters are as follows: Hypothalamic hormones: CRH, corticotropin-releasing hormone; GnRH, gonadotropin-releasing hormone; and dopamine. Pituitary hormones: ACTH, adrenocorticotropic hormone; AVP, arginine vasopressin; FSH, follicle-stimulating hormone; GH, growth hormone; LH, luteinizing hormone; Oxt, oxytocin; PRL, prolactin; and TSH, thyroid-stimulating hormone. Adrenal cortex hormones: steroid hormones—corticosteroids (cortisol), mineralocorticoids, and androgens. Adrenal hormone: E, epinephrine. Pineal gland hormone: melatonin. Other peripheral hormones: testosterone, T; estrogens, Es; thyroxin, T4; triiodothyronine, T3; somatomedins, IGF; angiotensin II; erythropoietin; calcitriol; somatostatin; glucagon; insulin; parathyroid hormone; and calcitonin. Neurotransmitters: Neuropeptide Y; substance P; GABA; serotonin; dopamine; acetylcholine; norepinephrine; and epinephrine.

**Figure 5 pharmaceuticals-18-00261-f005:**
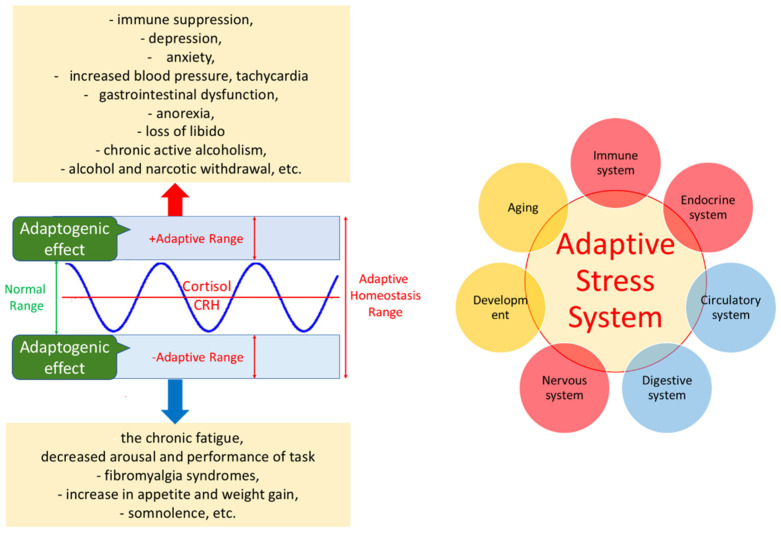
This figure presents a hypothetical representation of the adaptogenic effect of adaptogens on adaptive homeostasis. Adaptive homeostasis refers to the transient reversible adjustments of the homeostatic range in response to exposure to challenging signaling molecules or events. Any biological function or measurement oscillates around a mean or median within a homeostatic range considered ’normal’ or physiological. Adaptogens, as shown in the figure, increase the normal homeostatic thresholds (adaptive homeostasis) to a pathological state, thereby enhancing resilience to stress within the adaptive stress system. This system regulates various bodily functions, including the neuroendocrine-immune complex, blood circulatory and digestive systems, organismal development, and aging.

**Figure 6 pharmaceuticals-18-00261-f006:**
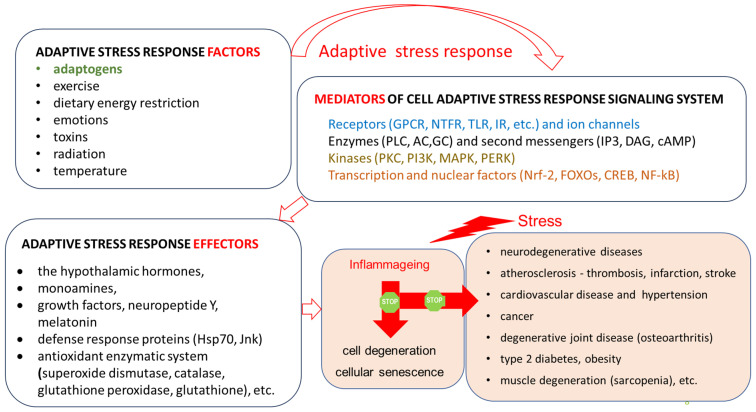
Adaptive stress response factors, mediators, and effectors and the effect of adaptogens in stress and inflammaging-induced aging-related diseases. The adaptive stress response involves the activation of intracellular and extracellular signaling pathways and increased expression of antiapoptotic proteins, neuropeptides, antioxidant enzymes, and the defense response of an organism, resulting in increased survival. One primary mechanism of adaptogens’ action is that they trigger adaptive cellular stress response pathways in human brain cells, similar to exercise, dietary restriction, and cognitive stimulation, which may exert their health benefits. Each of these environmental factors induces a mild stress response in nerve cells in the brain, increasing the expression of stress resistance proteins such as heat-shock protein 70 (HSP-70), Hsp32, and nerve cell growth factors, preventing the degeneration of neurons during aging, enhancing learning and memory, and exerting beneficial effects on many different organ systems, including the cardiovascular and glucose-regulating systems. These endogenous cellular defense pathways, including NRF2 signaling pathways, integrate adaptive stress responses to prevent neurodegenerative disease.

**Figure 7 pharmaceuticals-18-00261-f007:**
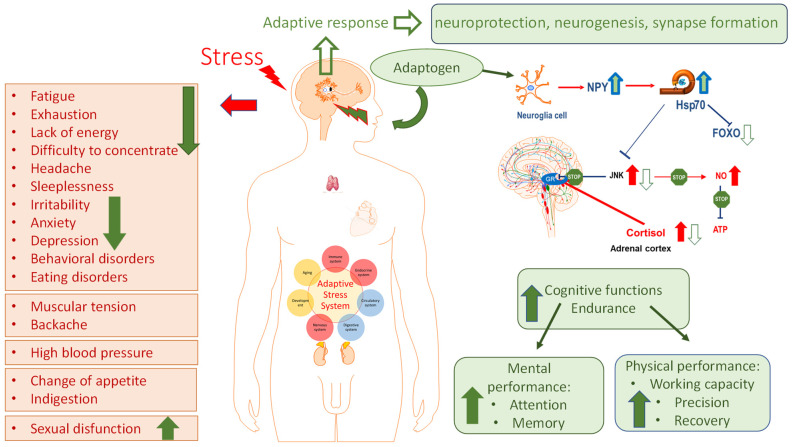
Chronic stress-induced symptoms and the effect of adaptogens on key mediators and effectors of adaptive stress response and effectors induce neuroprotection, resulting in increased cognitive function and mental and physical performance. Brain cells respond adaptively by enhancing their ability to function and resist stress, as shown by an update from the authors’ free access publication [25] and authors’ drawings.

**Figure 8 pharmaceuticals-18-00261-f008:**
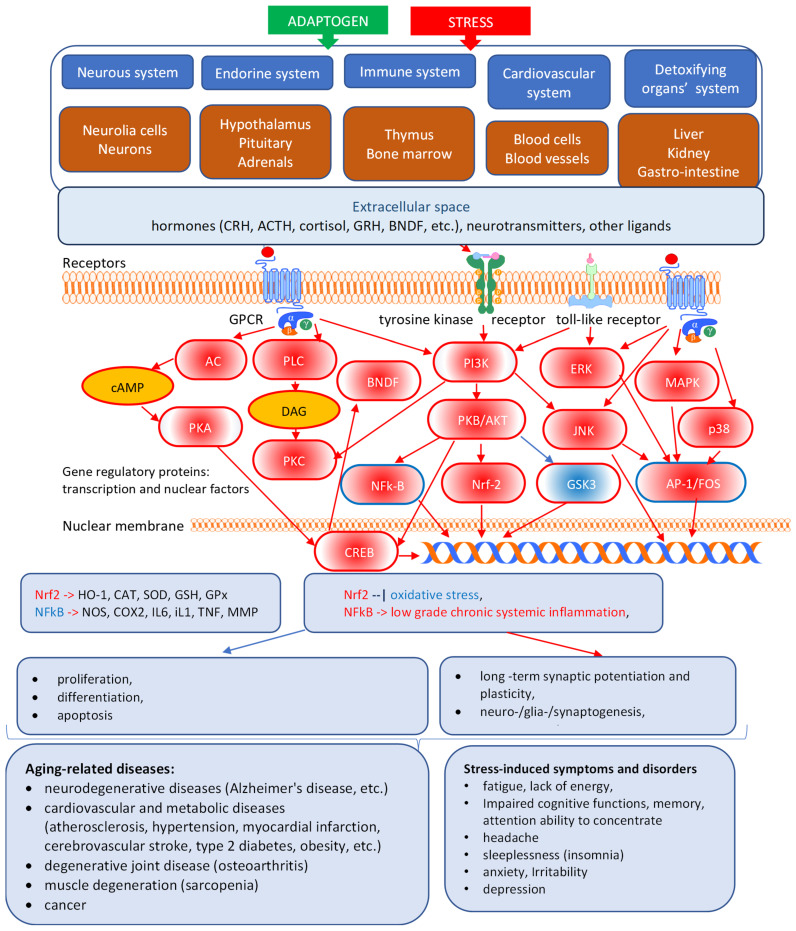
The hypothetic molecular mechanisms and modes of the pharmacological action of adaptogen are updated from the authors’ free access publication [66] and authors’ drawings. Effects of adaptogens on key mediators of neuroendocrine-immune complex, cardiovascular, and detoxifying systems that regulate adaptive stress response to stressors/pathogens in stress and aging-induced diseases and disorders. CRH- and ACTH-induced stimulation of GPCR receptors activates the cAMP-dependent protein kinase (PKA) signaling pathway in the regulation of energy balance and metabolism across multiple systems, including adipose tissue (lipolysis), liver (gluconeogenesis, glucose tolerance), pancreas, gut (insulin exocytosis and sensitivity), etc. The key molecules involved in the PI3K-Akt signaling pathway are receptor tyrosine kinases (RTKs). Activating the PI3K-Akt signaling pathway promotes cell proliferation and growth, stimulates cell cycle vascular remodeling and cell survival, and inhibits cell apoptosis in response to extracellular signals. The nonspecific antiviral action of ginseng is associated with the activation of innate immunity by upregulation of the expression of the pathogen’s pattern recognition receptors, specifically toll-like receptors and TLR-mediated signaling pathways. The protein kinase C (PKC) family of enzymes with isoforms plays an essential cell-type-specific role, particularly in the immune system, through phosphorylation of CARD-CC family proteins and subsequent NF-κB activation. Three stress-activated MAPK signaling pathways playing crucial roles in cell proliferation, differentiation, survival, and death have been implicated in the pathogenesis of many human diseases, including Alzheimer’s disease, Parkinson’s disease, and cancer. (1) The stress factors inducing the activation of the c-Jun N-terminal kinase (JNK)/stress-activated protein kinase (SAPK)-mediated adaptive signaling pathway are heat shock, irradiation, reactive oxygen species, cytotoxic drugs, inflammatory cytokines, hormones, growth factors, and other stresses. Activating the JNK/MAPK10 signaling pathway promotes cell death and apoptosis via the upregulation of proapoptotic genes. (2) The activation of the extracellular-signal-regulated kinase (ERK) pathway is initiated by hormones and stresses to trigger endothelial cell proliferation during angiogenesis, T cell activation, long-term potentiation in hippocampal neurons, phosphorylation of the transcription factor p53, activation of phospholipase A2 in mast cells, followed by activation of biosynthesis leukotrienes and inflammation/allergy, etc. (3) The third major stress-activated p38 signaling pathway contributes to the control of inflammation, the release of cytokines by macrophages and neutrophils, apoptosis, cell differentiation, and cell cycle regulation. Activation is shown in red, while the inhibition is in blue color cycles/ellipses (effect of ginseng/ginsenosides), arrows, and clouds. BDNF, brain-derived neurotrophic factor; cAMP, cyclic adenosine monophosphate; CREB, cAMP-responsive element-binding protein; ERK, extracellular signal-regulated kinase; GSK-3, glycogen synthase kinase-3; JNK, the c-Jun N-terminal kinase (JNK)/stress-activated protein kinase (SAPK MAPK, mitogen-activated protein kinase); NF-κB, nuclear factor-kappa B; Nrf2, nuclear factor E2-related factor 2; PI3K, phosphatidylinositol 3-kinase; PKA, protein kinase A; PKB, protein kinase B; PLC, phospholipase C.

**Figure 9 pharmaceuticals-18-00261-f009:**
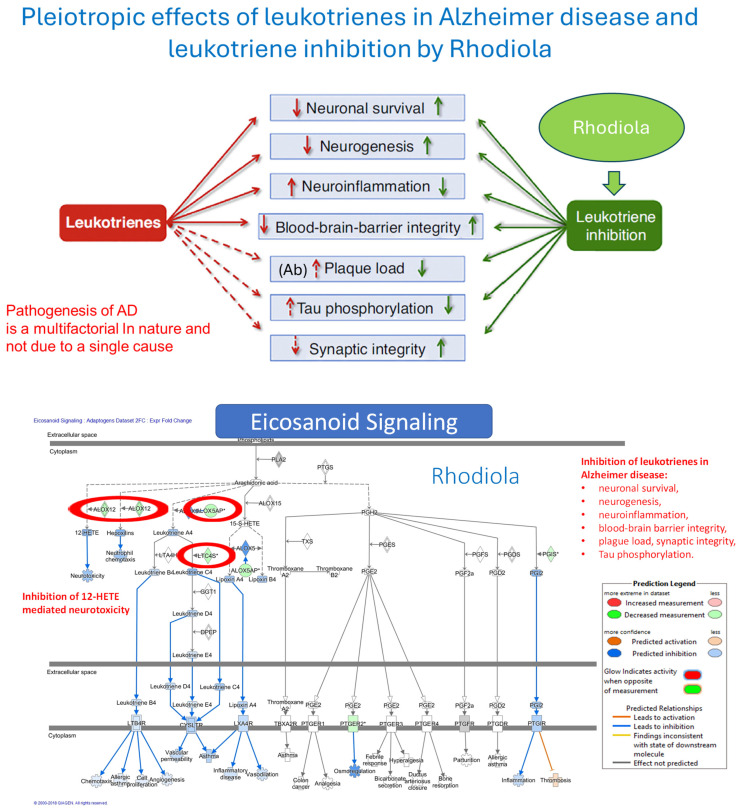
Effect of *Rhodiola* extracts on the eicosanoid signaling pathway. Arachidonic acid (AA) was released from membrane phospholipids by phospholipase A2 (PLA2) and then converted to eicosanoids by two types of enzymes: (i) prostaglandin endoperoxide synthases (PTGS), commonly referred to as cyclooxygenases (COX-1 and COX-2), catalyze the key step in the synthesis of prostaglandin H2 (PGH2), which is converted into pro-inflammatory thromboxanes (TXs), prostaglandins (PGE, PGF, and PGD), and prostacyclins (PGI). (ii) The lipoxygenases include ALOX5, ALOX12, and ALOX15. ALOX5 catalyzes the key step in the conversion of AA to pro-inflammatory leukotriene A4, B4, and C4. ALOX12 synthesizes pro-inflammatory 12(S)-HETE [12(S)-hydroxyeicosatetraenoic acid]. ALOX15, in concert with ALOX5, is involved in forming anti-inflammatory lipoxins A4 and B4. Eicosanoid receptors belong to the family of G-protein-coupled receptors. Some of these receptors include BLT-1,-2 CYSLTR1, and CYSLTR2 for pro-inflammatory leukotrienes; PTGERs for prostaglandin E2; PTGFR for pro-inflammatory prostaglandin F2; PTGDR for prostaglandin D2; and TBXA2R for pro-inflammatory thromboxane A2. Eicosanoids transduce signals via their membrane receptors and mediate complex biological processes like inflammation, vascular permeability, allergic reactions, labor induction, and carcinogenesis. Downstream effect analysis reveals predicted pharmacological effects of *Rhodiola* mediated by eicosanoid signaling pathways [68].

**Figure 10 pharmaceuticals-18-00261-f010:**
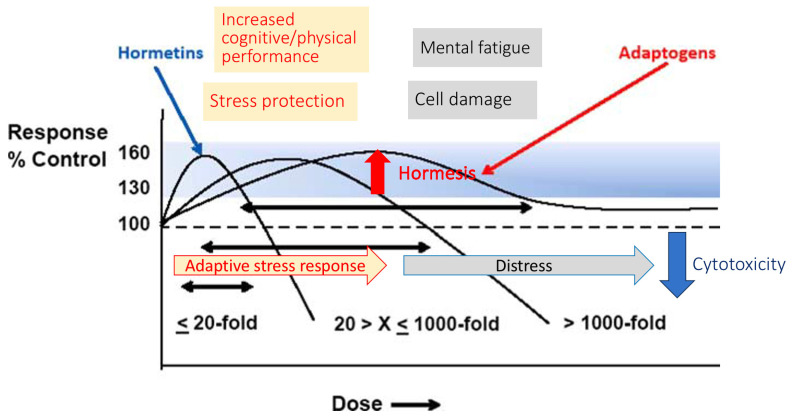
Simplified hypothetic representation of the hormetic dose–response relationship of toxic hormetins and adaptogens; figures adapted from free access publication [30,220,269].

**Figure 11 pharmaceuticals-18-00261-f011:**
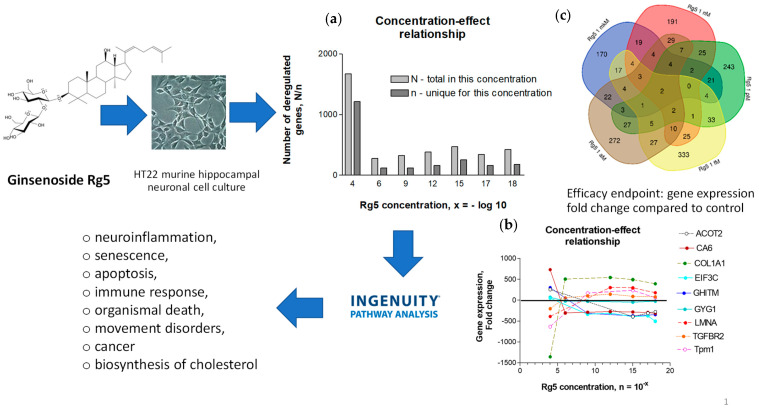
(**a**) The total number of genes deregulated by ginsenoside Rg5 in concentrations ranging from 1 μM to 1 aM; (**b**) ginsenoside Rg5 concentration-dependent fold change expression of selected differentially regulated genes in the hippocampal neuronal cell line HT22; (**c**) Venn diagram of genes deregulated by ginsenoside Rg5 at concentrations 1 μM, 1 nM, 1 pM, 1 fM, and 1 aM.

**Figure 12 pharmaceuticals-18-00261-f012:**
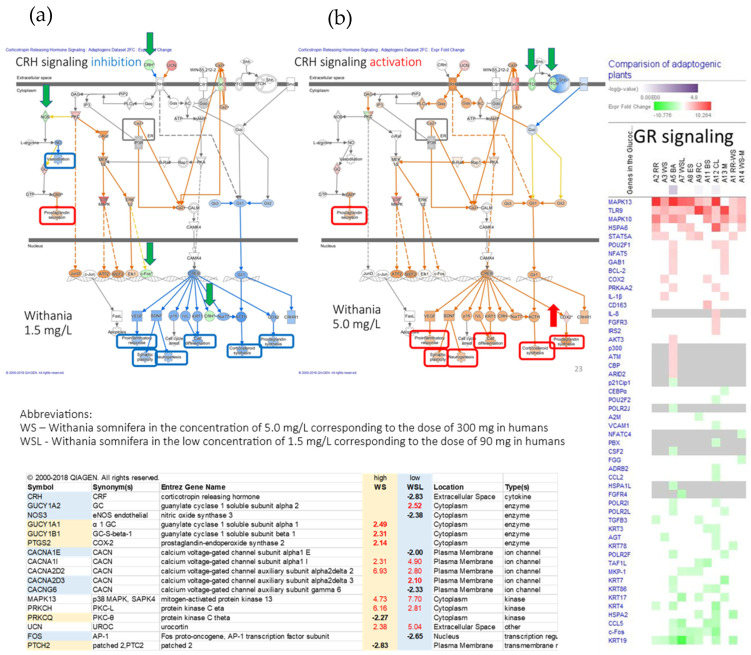
CRH signaling pathways are differently regulated by *Withania somnifera* extract at a concentration of 1.5 mg/L (corresponding to the dose of 90 mg in humans), WSL (**a**), and 5 mg/L (corresponding to the dose of 300 mg in humans), WS (**b**) in cultivated neuroglial cells. Figure 12a shows the inhibition of the CRH receptor-related intracellular signal transduction pathway, while Figure 12b shows the predicted activation of this pathway. At a concentration of 5 mg/L, corresponding to a human daily dose of 300 mg, WS extract did not affect the expression of CRH, AP-1 transcription factor subunit (FOS), CACNA1E, CACNG6, or CACNA2D3 encoding calcium voltage-gated channel auxiliary subunits as it did at a lower concentration of 1.6 mg/L. Protein kinase C η and ζ encoding genes. PRKCH and PRKCZ were downregulated, and guanylate cyclase 1 soluble subunit α and β (GUCY1A3, GUCY1B3) and prostaglandin-endoperoxide synthase 2/COX-2 (PTGS2) genes were upregulated [67].

**Figure 13 pharmaceuticals-18-00261-f013:**
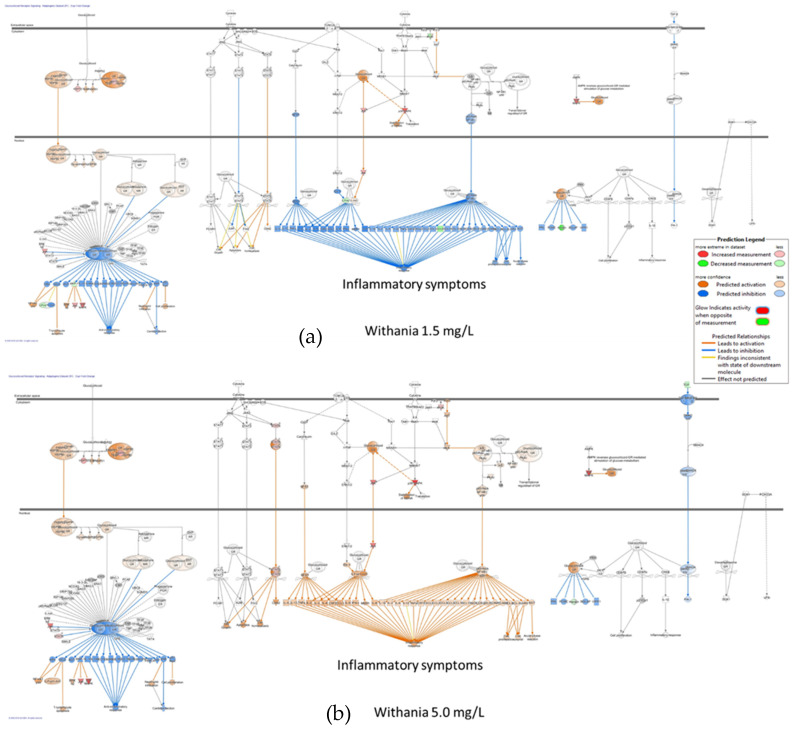
Glucocorticoid signaling pathways are differently regulated by *Withania somnifera* extract at a concentration of 1.5 mg/L (corresponding to the dose of 90 mg in humans), WSL (**a**), and 5 mg/L (corresponding to the dose of 300 mg in humans), WS (**b**) in cultivated neuroglial cells [67].

**Figure 14 pharmaceuticals-18-00261-f014:**
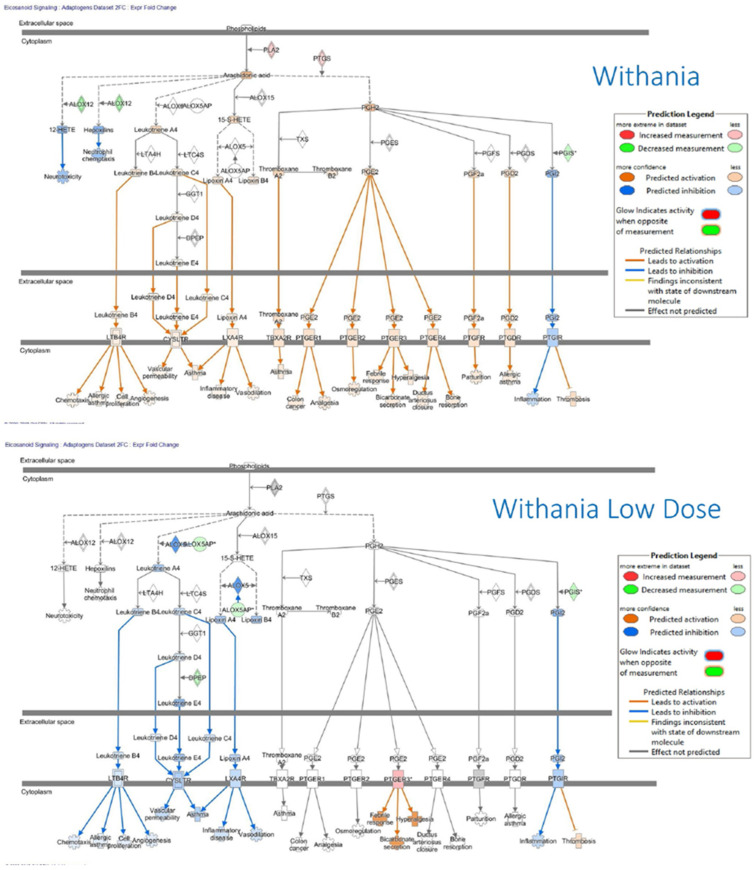
Effect of Withania extracts in two doses on the eicosanoid signaling pathway. The details are the legends of Figure 5 [67].

**Figure 15 pharmaceuticals-18-00261-f015:**
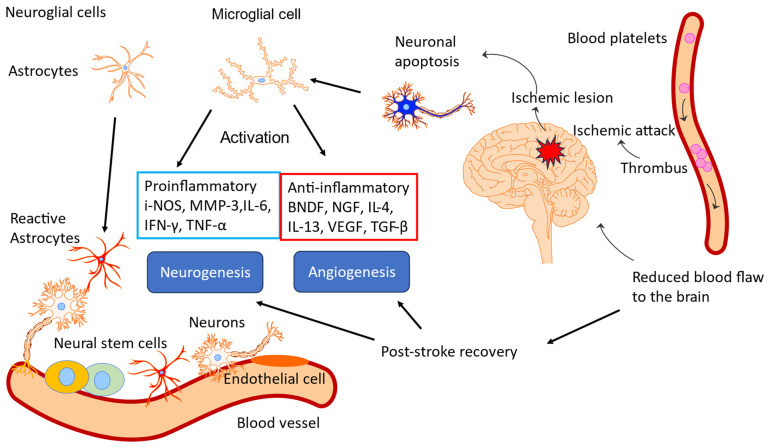
The mediators and interactions between vascular and brain cells in poststroke adult neurogenesis and angiogenesis: BDNF, brain-derived neurotrophic factor; IFN-γ, interferon-gamma; IL-4, interleukin-4; iNOS, inducible nitric oxide synthase; MMP-3, matrix metalloproteinase-3; NGF, nerve growth factor; TGF-β, transforming growth factor-β; TNFα, tumor necrosis factor-alpha; VEGF, vascular endothelial growth factor (Modified from [12]).

**Figure 16 pharmaceuticals-18-00261-f016:**
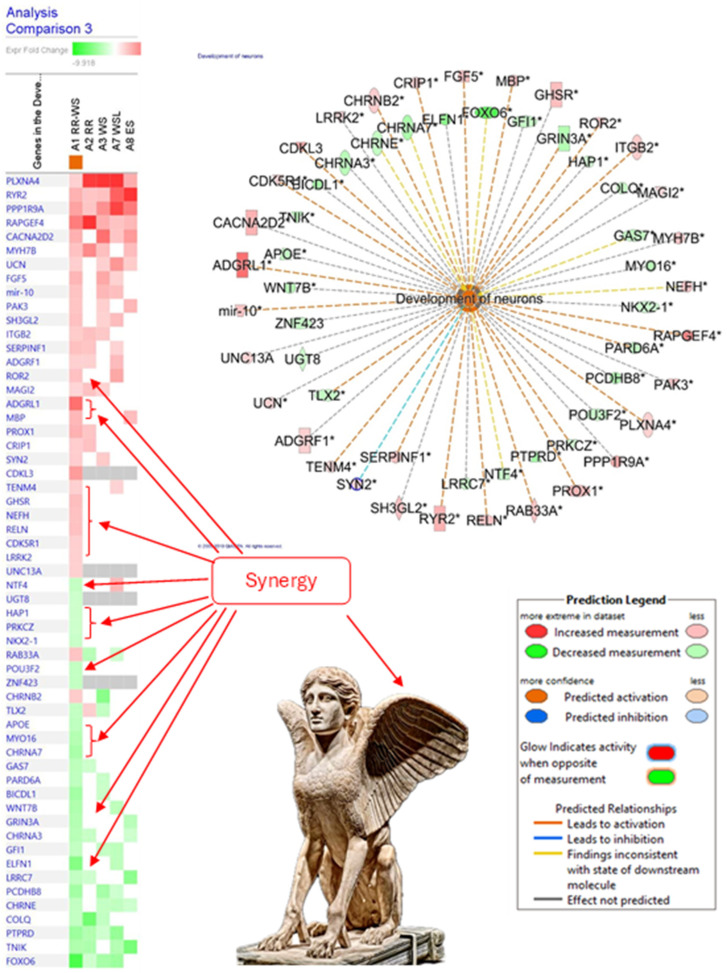
The effects of RR-WS (Adaptra^®^) on gene expression in human T98G neuroglia cells and the predicted activation of the development of neurons. The authors’ drawings were adapted from a free access publication [69]. The synergy effects (red arrows) of hybridization of a combination of Rhodiola with Withania on neurogenesis signaling pathways in isolated neuroglia cells. The intensity of green and red squares indicates fold changes compared to control, where green means down- and red means upregulation. Synergistic or antagonistic effects on gene expression were observed by comparison of the impact of the BHP Adaptra = combination of RR-WS (sample A1) with a lack of the impact of individual extracts RR (*R. rosea*), WS (*Withania somnifera*), and WSL *Withania somnifera* low dose, correspondingly samples A2, A3, and A7) at a significance level of *p* < 0.05 (log = 1.3) and a z-score > 2. The symbolic interpretation of synergy and antagonism by the image of a hybrid creature from Greek mythology, the Sphinx of Lanuvium, with a human head and a lion’s body-derived wing due to their synergistic and antagonistic (e.g., lack of human legs) interactions. The image of two kinds of hybrid creatures from ancient mythology is a visual analogy of botanical hybrid preparations (BHP) symbolizing synergy, e.g., wings of a sphinx, a mythical hybrid creature with the head of a human, the body of a lion, and the wings derived due to the synergy effect. Sphinx of Lanuvium. Near Rome. Roman, about AD 120–140. British Museum. It was found at Monte Cagnolo, outside Lanuvium, near Rome. Source: [417].

**Table 1 pharmaceuticals-18-00261-t001:** Typical symptoms of pathological brain/mental fatigue according to Johansson and Rönnbäck [5,114].

Abnormal loss of mental energy in everyday activities.
Disproportionately long recovery time and mental energy after activity and mental exhaustion.
Impaired ability to concentrate, decreased attention over time, and sensitivity to disturbances: trouble focusing on tasks or maintaining attention.
Subjective memory disturbance: forgetfulness or difficulty recalling recent information.
Mental fog: a feeling of mental cloudiness where thoughts feel slow or unclear.
Impaired simultaneous capacity, difficulty getting started with an activity.
Reduced cognitive performance: decline in problem-solving abilities or slower processing speeds.
Stress sensitivity.
Noise and light sensitivity.
Emotional lability: increased emotional sensitivity.
Increased Emotionality and Irritability: decreased tolerance for frustration.
Slurry speech.
Mood changes: increased anxiety, stress, or depressive feelings.
Sleep disturbance.
Lack of simultaneous capacity, i.e., difficulty performing several things at the same time.
Headache after mental overload.

**Table 2 pharmaceuticals-18-00261-t002:** Active compounds of adaptogens targeting the neuroendocrine system *.

Neuroendocrine System					
SNS = sympathetic nervous system			HPA-axis = hypothalamus pituitary adrenal axis		
	**Neurotransmitters and hormones**		**Hormones**		
	dopamine	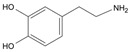		cortisol	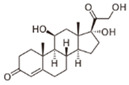
	norepinephrine	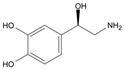		testosterone	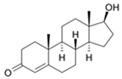
	epinephrine	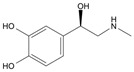		estradiol	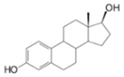
**Plant name,** **reference**	**Adaptogen name and phytochemical structure**		**Plant name**	**Adaptogen name and phytochemical structure**	
*Rhodiola* *rosea* [224,239]	thyrosol	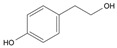	*Panax* *ginseng* [63,64]	ginsenoside Rg5 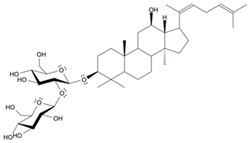	
	salidroside	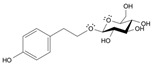			
	rosavin	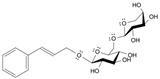			
*Eleutherococcus* *senticosus* [88]	eleutheroside B (syringin)	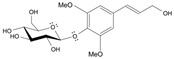	*Withania somnifera* [240,241,242,243,244]	withanolide A	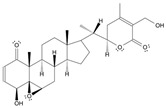
	eleutheroside B1	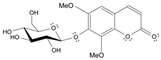			
	eleutheroside E	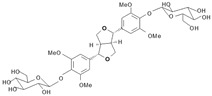		withaferin A	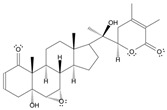
*Sideritis* *scardica* [24]	acteoside/verbascoside 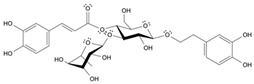		*Andrographis paniculata* [245]	andrographolide 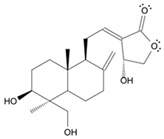	
*Schisandra* *chinensis* [223]	Schisandrin	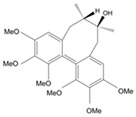	*Bryonia alba* [246]	cucurbitacin R diglucoside 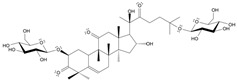	
	schisandrin B	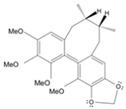			
*Rhaponticum cartamoides* [49]	*p*-coumaric acid	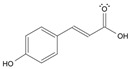	*Rhaponticum cartamoides* [247]	20-hydroxyecdysone 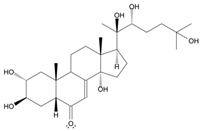	
	ferulic acid	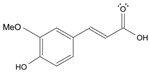			
	synapic acid	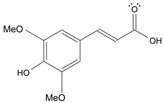			
*Bacopa monnieri* [248]	monnierasides I-IV		*Bacopa monnieri* [249,250]	bacoside A	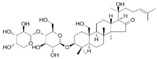
	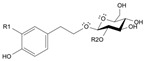	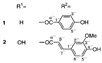 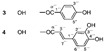		bacopaside I	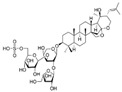
*Salvia* *miltiorrhiza* [251]	salvianolic acid A	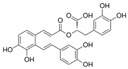	*Salvia* *miltiorrhiza* [252,253]	tanshinone *I*	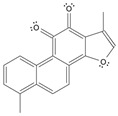
	salvianolic acid B	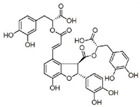		tanshinone IIA	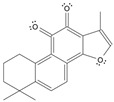
*Codonopsis pilosula* [254]	tangshenoside VIII	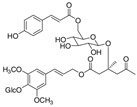	*Codonopsis pilosula* [89,255]	echinocystic acid glycosides	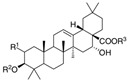
	codonoside B	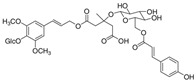		lancemaside A	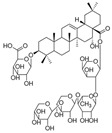

* Chemical structures of neuroendocrine hormones (cortisol, testosterone, estradiol, and epinephrine), neurotransmitters (dopamine, norepinephrine, epinephrine, and its precursor tyrosine), and plant metabolites of adaptogenic plants tyrosol, salidroside, rosavin, eleutherosides, acteoside, schizandrins, ginsenoside Rg5, withanolide A, withaferin A, andrographolide, acteoside, 20-hydroxyecdysone, tanshinones, bacosides, bacoposodes, lancemasides, tangshenoside VIII codonoside B, salvianolic acids, p-coumaric, ferulic, synapic acids, and cucurbitacin R diglucoside.

**Table 3 pharmaceuticals-18-00261-t003:** Pharmacological activities of adaptogens. Adapted from references [66,67].

Stimulating and Tonic Effects on the CNS System
Modulation of the stress response system, including the hypothalamus–hypophysis–adrenal (H.P.A.) axis
Modulation of the endocrine system and metabolic regulation
Regulation of cellular homeostasis and metabolism
Modulation of the immune response
Increased expression of defensin peptides Increased expression of pathogen pattern recognition proteins, TLRs Increased expression of interferons Inhibition of cytokine release Inhibition of NF-κB Activation of natural killer cells Activation of phagocytic cells Activation of T- and B-lymphocytes Activation of the melatonin signaling pathway
Anti-inflammatory activity
Inhibition of NF-κB-mediated signaling Inhibition of PLA2, arachidonic acid release, and metabolism Inhibition of nitric oxide generation
Detoxification and reparation of oxidative stress-induced damages in compromised cells
Activation of NRF2-signaling pathway proteins (KEAP1) Expression of phases I/II-metabolizing and antioxidant enzymes: glutathione S-transferase (G.S.T.), N.A.D. (P)H quinone oxidoreductase 1 (NQO1), superoxide dismutase (S.O.D.), and heme oxygenase 1 (HO1). Molecular chaperon (Hsp70)-mediated cytoprotection and repair processes Activation of melatonin signaling pathway
Direct antiviral activity
Inhibition of virus binding to host cell and its fission into the cytoplasm Termination of the viral life cycle in the infected host cell

**Table 4 pharmaceuticals-18-00261-t004:** Main cellular functions most influenced by adaptogens adapted from reference [62].

Cellular Function	Genes
**Cellular compromise** - Oxidative stress response of blood cells - Degranulation of β-islet cells - Damage to mitochondria - Degeneration of hepatocytes - Cytotoxicity of cytotoxic T cells - Fragmentation of photoreceptor outer segments - Degeneration of retinal cone cells	*AIPL1*, *ALOX12*, *CDHR1*, *NGB3*, *GNLY*, *HLA-B*, *NCAM1*, *SERPINA1*, *ULBP3*, *XRCC5*,
**Cell signaling**	*PDE3A*, *MUC20*, *PDE4D*, *PDE11A*, *ESR1*, *CCKBR*
**DNA replication, recombination, and repair**	*PARPBP*, *PDE3A*, *APLF*, *PDE4D*, *PDE11A*, *XRCC5*, *AICDA*
**Nucleic acid metabolism**	*PFKFB1*, *MTNR1A*, *PDE3A*, *APOBEC2*, *TAAR1*, *PDE4D*, *PDE11A*, *AIPL1*, *ESR1*, *AICDA*
**Lipid metabolism**	*NR4A3*, *RGS3*, *SLC27A2*, *AKR1D1*, *TNXB*, *SERPINA1*, *ALOX12*, *ESR1*, *CCKBR*, *CETP*, *NCAM1*

**Table 5 pharmaceuticals-18-00261-t005:** Age-associated diseases and genes involved in their pathogenesis and progression that are significantly regulated by adaptogens [62].

**Category**	**Diseases**	**Genes Affected by Adaptogens**
Organismal injury and abnormalities	Physical disability Degeneration of retinal cone cells Atrophy of gastric mucosa Hypoestrogenism Postmenopausal vulvar atrophy Nociception Cone dystrophy Pelvic organ prolapse	*PDE11A*, *PDE3A*, *PDE4D* *AIPL1*, *CNGB3* *CCKBR* *ESR1* *ESR1б*, *MTNR1A* *KCNK10*, *PDE11A*, *PDE3A*, *PDE4D*, *SCN2B* *CDHR1*, *CNGB3* *ESR1*, *SERPINA1*
Inflammatory and pulmonary diseases	Pulmonary emphysema Bronchiectasis Chronic bronchitis Chronic obstructive pulmonary disease	*PDE11A*, *PDE3A*, *PDE4D*, *SERPINA1* *PDE11A*, *PDE3A*, *PDE4D* *MMP8*, *MTNR1A* *PDE11A*, *PDE3A*, *PDE4D*, *SERPINA1*
Neurological and psychological diseases	Non-24 h sleep–wake disorder Sleep–wake schedule disorder	*MTNR1A* *PDE3A*
Cardiovascular diseases	Ischemic cardiomyopathy Cholesteryl ester transfer protein deficiency Angina pectoris Cerebral small vessel disease	*PDE11A*, *PDE3A*, *PDE4D*, *PPP1R1A* *CETP* *PDE11A*, *PDE3A*, *PDE4D*, *PDE3A*
Skeletal and connective tissues	Osteochondrodysplasia	*COL9A1*, *PDE4D*
Metabolic disease	Estrogen resistance	*ESR1*

**Table 6 pharmaceuticals-18-00261-t006:** Effect of RR, WS, and their combination RR-WS (Adaptra^®^) on genes involved in regulating neuronal development.

Gene Symbol	Gene Name	Literature Findings	Prediction *,**,***	Gene Expression, Fold Change		
				RR-WS	RR	WS
*ADGRF1*	Adhesion G protein-coupled receptor F1, Latrophilin-1,	Affects (4)	Affected	2.29	2.28	
*ADGRL1*	Adhesion G protein-coupled receptor L1	Increases (2)	Increased	6.93		
*APOE*	Apolipoprotein E	Affects (13)	Affected	−2.84		
*BICDL1*	BICD family like cargo adaptor 1	Affects (2)	Affected	−3.98		−2.34
*CACNA2D2*	Calcium voltage-gated channel auxiliary subunit α2 δ2	Affects (2)	Affected	3.76		6.93
*CDK5R1*	Cyclin-dependent kinase 5 regulatory subunit 1	Increases (4)	Increased	2.33		
*CDKL3*	Cyclin-dependent kinase like 3	Increases (3)	Increased	4.82		
*CHRNA3*	Cholinergic receptor nicotinic α3 subunit	Affects (2)	Affected	−3.09	−2.45	
*CHRNA7*	Cholinergic receptor nicotinic α 7 subunit	Increases (1)	Decreased	−3.74		
*CHRNB2*	Cholinergic receptor nicotinic β 2 subunit	Increases (8)	Increased	2.45		−5.20
*CHRNE*	Cholinergic receptor nicotinic ε subunit	Increases (1)	Decreased	−2.65		−2.59
*COLQ*	Collagen-like tail subunit of acetylcholinesterase	Affects (2)	Affected	−2.65	−6.30	−2.69
*CRIP1*	Cysteine-rich protein 1	Increases (1)	Increased	2.41	3.01	
*ELFN1*	Extracellular leucine-rich repeat and fibronectin type III domain containing 1	Affects (1)	Affected	−5.31		
*FGF5*	Fibroblast growth factor 5	Increases (1)	Increased	3.52		4.23
*FOXO6*	Forkhead box O6	Increases (3)	Decreased	−7.93	−2.10	−3.89
*GAS7*	Growth arrest specific 7	Increases (3)	Decreased	−2.85	−2.26	
*GFI1*	Growth factor-independent 1 transcriptional repressor	Affects (1)	Affected	−2.65		−2.59
*GHSR*	Growth hormone secretagogue receptor	Affects (3)	Affected	3.15		
*GRIN3A*	Glutamate ionotropic receptor NMDA type subunit 3A	Decreases (4)	Increased	−3.33		
*HAP1*	Huntingtin-associated protein 1	Affects (1)	Affected	−2.21		
*ITGB2*	Integrin subunit β2	Increases (1)	Increased	2.45	3.05	2.66
*LRRC7*	Leucine-rich repeat containing 7	Affects (1)	Affected	−2.66	−2.11	
*LRRK2*	Leucine-rich repeat kinase 2	Affects (4)	Affected	2.26		
*MAGI2*	Membrane-associated guanylate kinase,	Affects (10)	Affected	2.01	3.01	2.18
*MBP*	Myelin basic protein	Increases (1)	Increased	3.48		
*mir-10*	MicroRNA 100	Increases (1)	Increased	2.89		3.53
*MYH7B*	Myosin heavy chain 7B	Affects (1)	Affected	3.01	5.70	3.08
*MYO16*	Myosin XVI	Affects (1)	Affected	−3.32		
*NEFH*	Neurofilament heavy	Decreases (18)	Decreased	3.02		
*NKX2-1*	NK2 homeobox 1	Affects (4)	Affected	−2.38		
*NTF4*	Neurotrophin 4	Increases (5)	Decreased	−2.61		
*PAK3*	p21 (RAC1) activated kinase 3	Affects (4)	Affected	2.86		2.36
*PARD6A*	Par-6 family cell polarity regulator α	Decreases (2)	Increased	−2.84		−2.39
*PCDHB8*	Potocadherin β8	Affects (1)	Affected	−3.97		−3.89
*PLXNA4*	Plexin A4	Increases (5)	Increased	2.25	9.49	10.97
*POU3F2*	POU class 3 homeobox 2	Affects (4)	Affected	−2.65		
*PPP1R9A*	Protein phosphatase 1 regulatory subunit 9A	Affects (6)	Affected	4.51	2.85	5.38
*PRKCZ*	Protein kinase C ζ	Decreases (2)	Increased	−2.23		
*PROX1*	Prospero homeobox 1	Increases (1)	Increased	3.76	2.85	
*PTPRD*	Protein tyrosine phosphatase, receptor type D	Increases (3)	Decreased	−4.32	−3.42	−2.11
*RAB33A*	RAB33A, member RAS oncogene family	Increases (1)	Increased	2.81	−3.11	
*RAPGEF4*	Rap guanine nucleotide exchange factor 4	Increases (2)	Increased	6.31	11.27	3.92
*RELN*	Reelin	Increases (9)	Increased	3.01		
*ROR2*	Receptor tyrosine kinase-like orphan receptor 2	Increases (5)	Increased	3.01		
*RYR2*	Ryanodine receptor 2	Increases (2)	Increased	3.75	2.85	3.07
*SERPINF1*	Serpin family F member 1	Increases (1)	Increased	3.04	2.88	
*SH3GL2*	SH3 domain containing GRB2 like 2, endophilin A1	Affects (2)	Affected	3.02		2.16
*SYN2*	Synapsin II	Affects (3)	Affected	2.41		2.56
*TENM4*	Teneurin transmembrane protein 4	Increases (3)	Increased	2.25		
*TLX2*	T cell leukemia homeobox 2	Decreases (2)	Increased	−2.64	2.39	−2.59
*TNIK*	TRAF2 and NCK interacting kinase	Affects (1)	Affected	−3.53	−3.60	
*UCN*	Urocortin	Affects (1)	Affected	2.35		2.38
*UGT8*	UDP glycosyltransferase 8	Affects (2)	Affected	−2.21		
*UNC13A*	Unc-13 homolog A	Affects (2)	Affected	2.25		
*WNT7B*	Wnt family member 7B	Affects (2)	Affected	−3.54		
*ZNF423*	Zinc finger protein 423	Affects (4)	Affected	−2.66		

* Development of neurons predicted to be increased (z-score −2.87). Overlap *p*-value 7.29 × 10^−3^. ** Prediction is based on measurement direction and literature data: 22 of 57 genes deregulated by RR-WS have measurement direction consistent with an increase in the development of neurons. *** Twenty-five genes (in red color heatmap on the Figure 16) deregulated due to synergistic interactions of RR and WS in the fixed combination Adaptra^®^ are in red text, Figure 16.

**Table 7 pharmaceuticals-18-00261-t007:** Summary of positive impacts.

**Gene** **Expression**	**Role and Effect on Brain Fatigue**	**Potential Positive/Beneficial Consequences on Brain Fatigue Effects**
*PRKCZ* downregulation	Encode encoding the protein kinases PKCζ and PKMζ, which regulate glucose transport and long-term memories. Reduced overactivation of memory pathways. Protection against excitotoxicity.	Overexpression of PKMζ is linked to pathological memory processes, observed in post-traumatic stress disorder (PTSD) or chronic pain. Post-stroke fatigue can be worsened by mental overexertion, where the brain struggles to manage excessive cognitive demands. Reduced PKMζ might dampen overactive memory consolidation, allowing for better energy conservation in the brain. Excessive synaptic activity in damaged areas can cause excitotoxicity. Lower PKMζ activity could help reduce the metabolic burden on neurons and protect against additional neuronal damage in the post-stroke environment.
*GRIN3A*, downregulation	Encoding glutamate ionotropic receptor gene subunit Grin3a of glutamate ionotropic receptor NMDAR with central roles in brain plasticity, development, learning, and memory in the brain improved LTP and recovery of learning abilities.	Reduced GRIN3A expression can enhance calcium signaling through NMDARs, potentially improving synaptic plasticity, promoting calcium-dependent, and potentially aiding the recovery of learning and memory functions often impaired after a stroke.
	Reduced inhibition of NMDARs and. Enhanced synaptic efficiency.	GRIN3A subunits often dampen NMDAR activity, particularly in immature neurons. Their downregulation may improve synaptic transmission efficiency, which could benefit tasks requiring attention or learning during rehabilitation. Downregulation might promote more robust excitatory signaling and improved adaptability during tasks requiring high cognitive demand.
*ADGRL1*, upregulation	Regulates synaptic adhesion and communication between neurons. Enhanced synaptic connectivity, Facilitation of cognitive rehabilitation. Improved neural network stability.	Upregulation can increase neuronal adhesion, improve synaptic strength, and promote neuroplasticity, which is crucial for recovery after stroke. Better synaptic signaling may improve cognitive abilities, such as memory, attention, and learning, helping combat cognitive fatigue in post-stroke recovery. Strengthened synaptic connections could stabilize disrupted neural networks, supporting the brain’s ability to restore normal function.
*CDK5R1/* *CDKL3*, upregulation	Encode p35, an activator of CDK5, which regulates neuronal migration, cytoskeleton dynamics, and synaptic signaling. CDKL3 is another kinase implicated in similar pathways. Promotion of neuroplasticity: Support for cognitive recovery. Facilitation of axonal regrowth.	Upregulation may enhance dendritic spine remodeling, a key process in forming new connections to compensate for stroke-induced damage. Both CDK5R1 and CDKL3 are involved in learning and memory. Their upregulation could improve recovery of cognitive functions and counteract brain fatigue. Increased kinase activity may promote axonal regeneration, helping restore connectivity in damaged brain areas.
*CHRNB2*, upregulation	Encodes a subunit of nicotinic acetylcholine receptors (nAChRs), which mediate cholinergic signaling in the brain and play key roles in attention, memory, and arousal. Enhanced cognitive recovery. Increased neural excitability for rehabilitation. Reduction in inflammation.	Upregulation of CHRNB2 could improve cognitive function, such as attention and working memory, which are often impaired after a stroke. Enhanced cholinergic signaling can improve neural network responsiveness during rehabilitation, facilitating learning and neuroplasticity. Cholinergic signaling via nAChRs has anti-inflammatory effects, which might help reduce neuroinflammation in the post-stroke brain.
*ROR2* upregulation	ROR2 is involved in the Wnt signaling pathway, critical for cell proliferation, migration, and synaptic function. Support for neurogenesis and repair. Improved vascular recovery. Enhancement of neuroplasticity.	Upregulation of ROR2 may enhance neurogenesis and synaptic remodeling, aiding recovery from stroke-induced damage. ROR2-mediated Wnt signaling can promote angiogenesis, restoring blood flow and oxygen supply to damaged brain areas, which is vital for combating fatigue. ROR2 may facilitate synaptic plasticity, helping the brain adapt to stroke-induced deficits and improve cognitive function.
*ADGRF1* upregulation	Promotes cell adhesion and neuronal development.	Upregulation may attempt to counteract stroke damage, but excessive activity could disrupt normal signaling, contributing to dysregulated repair processes.
*CHRNB2* upregulation	Critical for synaptic transmission and cognitive function.	Increased CHRNB2 could enhance excitability but might also lead to excitotoxicity, exacerbating mental fatigue.
*GHSR* upregulation	Regulates energy homeostasis, appetite, and stress responses.	Overactivation may lead to metabolic imbalances and dysregulated stress responses, contributing to mental fatigue.
*ITGB2* upregulation	Modulates inflammation and leukocyte migration.	Upregulation could sustain inflammation in the brain, delaying recovery and worsening fatigue.
*LRRK2* upregulation	Linked to neuronal signaling and neuroinflammation.	Increased expression could drive chronic inflammation and impair neuronal recovery, intensifying cognitive fatigue.
*MAGI2* upregulation	Involved in synaptic scaffolding and plasticity.	Overexpression may lead to abnormal synaptic reorganization, contributing to cognitive and mental fatigue.
*MBP* upregulation	Essential for myelin sheath integrity and axonal insulation.	Increased MBP expression may reflect ongoing remyelination efforts, which could temporarily strain brain energy resources, contributing to fatigue.
*mir-10* upregulation	Regulates gene expression involved in neurogenesis and inflammation.	Dysregulation might affect neuronal repair and exacerbate inflammatory pathways, worsening brain fatigue.
*MYH7B* upregulation	Associated with cytoskeletal function and cellular transport.	Overexpression may lead to structural changes in neurons, which might either aid recovery or introduce metabolic strain.
*PPP1R9A* upregulation	Regulates synaptic signaling and plasticity.	Elevated levels may dysregulate synaptic activity, potentially interfering with cognitive recovery and contributing to fatigue.
*NTF4* downregulation	Supports survival and growth of neurons, synaptic plasticity, and repair mechanisms.	Reduced NEFH could weaken axonal stability and slow nerve conduction, contributing to persistent cognitive deficits and fatigue.
*NEFH* downregulation	Maintains the structural integrity of axons and supports neurotransmission.	Reduced NEFH could weaken axonal stability and slow nerve conduction, contributing to persistent cognitive deficits and fatigue.
*PTPRD* downregulation	Involved in synaptic organization and neuronal differentiation.	Upregulation may attempt to counteract stroke damage, but excessive activity could disrupt normal signaling, contributing to dysregulated repair processes.

## Data Availability

Data sharing is not applicable.

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
