# Peer review of "Adaptogens in Long-Lasting Brain Fatigue: An Insight from Systems Biology and Network Pharmacology"

_pharmaceuticals, 2025, doi:10.3390/ph18020261_

Round 1

Reviewer 1 Report

Comments and Suggestions for Authors

The manuscript titled “Adaptogens in long-lasting brain fatigue: An insight from systems biology and network pharmacology” by Alexander Panossian, Terrence Lemerond, and Thomas Efferth (Manuscript ID: pharmaceuticals-3470307) is an extended and multifaceted review, which aims to (i) present/summarize the main points of the authors’ previous works concerning the so-called adaptogens and their effects on stress-related fatigue along with relevant data of other researchers, (ii) evaluate/explain previous relevant findings from a broad and “holistic” point of view, and (iii) present the prospects of therapeutic application of (hybrid preparations of) adaptogens to brain fatigue, based on a well-established biological/pharmacological insight/comprehension. The review has been written in a meticulous way and provides a lot of information and interesting viewpoints, although some parts could have been shorter.

 Some minor comments are listed below:

 ---p. 2: Is the reference “(Billones et al., 2021)” identical with the reference [95] mentioned in the list of References?

 ---p. 6: Is the reference “(Kunasegaran et al., 2023)” identical with the reference [2] mentioned in the list of References?

 ---p. 6, two lines before the last line, “…or inflammation in the central nervous.”: this should change into “…or inflammation in the central nervous system.”

 ---p. 8, [111-11]: Do the authors mean [113-115]?

 ---p. 12, sixteen lines before the last line (also, p. 44, eight lines before the last line, and p. 61, thirteen lines before the last line), “law-grade”: Should it be “low-grade”?

 ---p. 14, legend of Figure 4, “Glutamate is released from the presynaptic terminal, and after exerting its effect on the recipient neuron, the postsynaptic membrane is taken up by the astrocytes' discharge and converted to glutamine, which is then transported back to the nerve cells to form new glutamate”. Is the phrase (including punctuation) correct? or is something missing??

 ---p. 14, eleven lines before the last line, “Glutamate is then released to the synapse and uptake by adjacent neurons, where it is…”: this should change into “Glutamate is then released to the synapse and taken up by adjacent neurons, where it is…”

 ---p. 16, five lines before the last line, “…by the Na/K ATPase”: This might be better to change into “…by the Na+/K+ ATPase”

 ---p. 19: The references “(Lazarev, 1957; Lazarev et al., 1958)” do not appear in the list of References.

 ---p. 20, legend of Figure 5, “locus caeruleus”: This should change into “locus coeruleus”.

 ---p. 24, seventeen lines after the first line: “(Panossian, 2003)” should be omitted.

 ---pp. 25-26, Table 2: The Table should be reproduced in a clearer way, especially the structural formulas of the active compounds of adaptogens.

 ---p. 26, “poumaric”: This should change into “coumaric”.

 ---p. 28, Figure 9, upper panel of parallelograms, “Detoxifying system”: This might change into “Detoxifying organs

 ---p. 29, Legend of Figure 9, 1st line, “NF-B”: this should change into “NF-κB”

 ---p. 29, Legend of Figure 9, 18th line, “GSK-3glycogen synthase kinase-3”: this should change into “GSK-3, glycogen synthase kinase-3”

 ---p. 36, Figure 11, “Ab”: This should change into “Aβ

 ---p. 38, Legend of Figure 12, “horemtins”: This should change into “hormetins”

 ---p. 40, Legend of Figure 13, “1 gM”: this should change into “1 fM”

 ---p. 42, Figure 15: This is difficult to be clearly seen, even at a high magnification scale.

 ---p. 42, lines 3-5, “That aligns with the hormesis concept, distinguished as a biphasic reversal effect from positive (stimulating) at low doses to negative (toxic/harmful) at low doses”: Is this phrase correct?

 ---p. 48, four lines before the last line, “phosphotyl inositol 3 kinase” should change to “phosphoryl inositol 3 kinase”

 ---p. 52, eight lines before the last line, “[242-406](Kulkarni and Dhir, 2008; Singh et al., 2011; Farooqui et al., 2018; Zahiruddin et al., 2020; Paul et al., 2021; Tewari et al., 2022; Mikulska et al., 2023; Lerose et al., 2024; WiciÅ„ski et al., 2024)”: “[242-406]” should change into “[242-244, 401-406]”, while “(Kulkarni and Dhir, 2008; Singh et al., 2011; Farooqui et al., 2018; Zahiruddin et al., 2020; Paul et al., 2021; Tewari et al., 2022; Mikulska et al., 2023; Lerose et al., 2024; WiciÅ„ski et al., 2024)” should be omitted. Please, cross-check.

 ---p. 55, Table 8, “Entrez gene name”: Is this correct?

Author Response

See Response to Reviewer 1

Reviewer 2 Report

Comments and Suggestions for Authors

In the current work, authors provided insight into future research on network pharmacology of adaptogens in preventing and rehabilitating long-lasting brain fatigue following stroke, trauma, and viral infections. This review is so comprehensive that can be accepted after minor revision.

Here are some points:

1. There is a redundancy, which makes the the manuscript out of the main subject. Authors are recommended to shoerten some parts appropriately. Alternatively, they can put some data to supplementary file to make the manuscript more brief and smoother.

2. Table 1 is not a table. It should be corrected in table format.

3. In general, figures are blurry.

4. In Figure 5, the text of some common molecules is not readable.

5. In Table 2, molecules are overlapped.

6. In Tables 4-7, blue and red colours are not necessary.

7. Authors could add future remarks to the Conclusion part.

8. Did authors use some methods to make some graphs in some figures? If yes, they could add a method part.

9. There are too many references. They should decrease appropriately.

Author Response

See reviewer 2 in attachment
